# Tailored ozone activation on geometrical-site-dependent cobalt with selective coordination

Shenning Liu[1], Yuxian Wang [1]✉, Ya Liu[2], Peihan Chen[1], Tao Kong[1], Xiaoguang Duan [2], Chunmao Chen[1]✉, Hongqi Sun [3]✉ & Shaobin Wang [2]

Cobalt-containing spinel oxides are promising platforms to fine-tune the intrinsic activity/selectivity of their geometric sites in catalysis. However, the role of tetrahedrally occupied $Co^{2+}$ ($Co^{2+}_{Td}$) and $Co^{3+}$ in an octahedral site ($Co^{3+}_{Oh}$) in controlling the catalytic activity remains controversial. Herein, we investigated a geometrical-site-dependent catalytic activation of ozone respectively on the $Co^{2+}_{Td}$ and $Co^{3+}_{Oh}$ sites. The same exposure of [111] crystal facet is achieved by substituting those undesired sites with catalytically inactive cations. The highly spin-polarized $Co^{2+}_{Td}$ sites invoke strong orbital interactions and intensive electron transfer with the adsorbed $O_3$ and become the active sites for selectively producing surface-bound hydroxyl radicals (˙OH) and avoiding the formation of unfavorable singlet oxygen ($^1O_2$), resulting in a 17.6-fold increase in turnover frequency (TOF). This work enlightens the spin-polarized electronic states into regulating the reaction thermodynamics in transition metal oxide-induced catalysis and envisages the practical application potentials of geometric site engineered spinel oxides.

Earth-abundant transition metal (TM) oxides have been demonstrated to be catalytically active in oxygen-involved catalysis, for example, oxygen reduction reaction and Fenton catalysis for chemical conversions and environmental remediations[1]. The multi-valence states of TM centers guarantee a swift redox cycling process for a durable catalytic activity and facilitate the electron transfer from their partially occupied $3d$ orbitals to the $2p$ orbitals of oxygen[2]. Among TM oxides, spinel $Co_3O_4$ and its substitutional variants have attracted intensive research interests because of their tunable crystal structures and active sites, in which the tetrahedrally occupied $Co^{2+}$ ($Co^{2+}_{Td}$) and octahedrally occupied $Co^{3+}$ ($Co^{3+}_{Oh}$) within the structure provide a versatile geometrical substitution platform[3,4]. Thus, a variety of substituted spinel cobaltite ($MCo_2O_4$) and $CoM_2O_4$ (M for substituted metals) have been synthesized with distinctive activity[5,6]. Previous studies usually ascribed the activity variations to the substitution-induced changes in the physicochemical properties, which resulted in the deviation of the

electron transfer ability[7,8]. Nevertheless, the intrinsic origin of the substitutional effects in cobalt catalysis remained ambiguous.

Recently, geometrical-site-dependent activity has been disclosed in cobalt-containing TM oxides[9,10]. Substituting the $Co^{2+}_{Td}$ or $Co^{3+}_{Oh}$ sites by cations with different reactivities rendered distinct activity deviations, which are intrinsically controlled by the altered local coordination environment[11]. The covalency competition between the $M_{Td}-O$ (metal–oxygen bond in a tetrahedral unit) and $M_{Oh}-O$ (metal–oxygen bond in an octahedral unit) can break the $M_{Td}-O-M_{Oh}$ backbones in spinels, resulting in the exposure of the non-oxygen-bonding metal[12]. And the high covalency competition enhances the electronic hybridization between the non-bonded metal $3d$ and O $2p$ orbitals, thereby promoting the catalytic activity. Additionally, the d-band positions of TM and the band overlap between the TM $3d$ orbitals and the O $2p$ orbitals[13,14], as well as the filling degree of the $e_g$ orbitals, have been proposed as descriptors to correlate the structure

[1]State Key Laboratory of Heavy Oil Processing, China University of Petroleum-Beijing, Beijing, China. [2]School of Chemical Engineering, The University of Adelaide, Adelaide, SA, Australia. [3]School of Molecular Sciences, The University of Western Australia, Perth, WA, Australia. ✉e-mail: yuxian.wang@cup.edu.cn; c.chen@cup.edu.cn; hongqi.sun@uwa.edu.au

and activity of Co-containing spinel oxides with tetrahedral and octahedral coordination[15].

These fundamental findings enlighten the excavation of the intricate mechanisms underlying cobalt catalysis, yet the dominant role of $Co^{2+}_{Td}$ and $Co^{3+}_{Oh}$ sites in modulating catalytic activity still remains controversial, even in the same catalytic system[2,16]. Most of the studies failed to regulate the exposed crystal facets of the spinel oxides by making them share the same facet, despite significant variations in reaction kinetics and thermodynamics across different exposed crystal facets[17]. Additionally, the spin polarization effects brought by spin-state variations in the substituted TM oxides are often overlooked in deciphering orbital hybridization states[18,19]. The asymmetric distributions of electrons in spin-up and spin-down channels demonstrate different electronic states, which significantly affect the reaction thermodynamics by modulating the positions of molecular orbitals and electronic communications with the reactants[20,21].

Heterogeneous catalytic ozonation, as one of the prevailing advanced oxidation processes, is appealing for its high efficiency in abating aqueous recalcitrant pollutants[22]. Herein, to well elucidate the origin of the activity of the cobalt-based spinels for selective ozone activation, we synthesize $Co_3O_4$ and its geometrical-site substitutes ($ZnCo_2O_4$ and $CoGa_2O_4$) with the same [111] crystal facet via a facile sol-gel method. X-ray absorption spectroscopy (XAS) validates the successful substitution of the $Co^{2+}_{Td}$ and $Co^{3+}_{Oh}$ sites in $Co_3O_4$ by catalytically inactive $Zn^{2+}$ and $Ga^{3+}$ with fully occupied $3d$ orbitals, respectively. By establishing the structural-activity relationship, the unsubstituted host Co cation sites are identified as the key factor governing $O_3$ activation, with $Co^{2+}_{Td}$ sites exhibiting much higher catalytic activity than $Co^{3+}_{Oh}$ sites. The improved $O_3$ utilization efficiency and enhanced selective ·OH generation by avoiding the formation of unfavorable $^1O_2$ account for the higher activity of $Co^{2+}_{Td}$ sites compared to $Co^{3+}_{Oh}$ sites. Theoretical investigations decipher that the highly spin-polarized $Co^{2+}_{Td}$ sites fine-tune the position of the antibonding molecular orbitals formed by the spin-down channels of the $Co^{2+}_{Td}$-$3d$ orbitals and $O_3$ $2p$ orbitals close to the Fermi level, favoring the $O_3$ adsorption and activation. The massively generated surface-bound ·OH at $Co^{2+}_{Td}$ sites results in a high chemical oxygen demand (COD) removal efficiency as well as the broadened band removal ability of the organics, as indicated by three-dimensional excitation-emission-matrix (3D-EEM) spectra for treatment of real refinery wastewater in a fixed bed continuous-flow reactor, envisaging the practical application potential of the cobalt-based spinels.

## Results

### Characterization of the catalysts

Substituting the host Co cations by $Zn^{2+}$ and $Ga^{3+}$ with fully occupied $3d$ orbitals makes octahedral $Co^{3+}$ ($Co^{3+}_{Oh}$) and tetrahedral $Co^{2+}$ ($Co^{2+}_{Td}$) as the exclusive Co sites in $ZnCo_2O_4$ and $CoGa_2O_4$, respectively (Fig. 1a). The synthesis routes of pristine $Co_3O_4$ and its geometrical-site substitutions are schematically illustrated in Supplementary Fig. 1. Powder X-ray diffraction (XRD) patterns suggest that replacing $Co^2$ by $Zn^{2+}$ and $Co^{3+}$ by $Ga^{3+}$ in $ZnCo_2O_4$ and $CoGa_2O_4$ accordingly successfully substitute the $Co^{2+}_{Td}$/$Co^{3+}_{Oh}$ sites in the crystal structure of pristine $Co_3O_4$ while preserving the normal spinel crystal phase (Fig. 1b; Supplementary Fig. 2a). The significantly larger octahedral site preference energy (OSPE) of $Co^{3+}$ compared to those of $Co^{2+}$ and $Zn^{2+}$ enables $Co^{3+}$ to preferentially occupy the octahedral sites while resides the $Co^{2+}$ and $Zn^{2+}$ in tetrahedral sites[23]. The fully occupied 3 d orbitals in $Ga^{3+}$ exclude the coordination geometry competition in $CoGa_2O_4$, making $Co^{2+}_{Td}$ as the dominant active sites. Additionally, the $A_{1g}$ symmetry and $F_{2g}^1$ symmetry in $CoGa_2O_4$ and $ZnCo_2O_4$ shifted slightly compared to the $Co_3O_4$, respectively. This observation reinforces the normal spinel structures of the as-synthesized Co-based spinels and the minor presence of inversed $Co^{2+}_{Oh}$/$Co^{3+}_{Td}$ sites[24] (Supplementary Fig. 3).

The atomic ratios of Co/Zn in $ZnCo_2O_4$ and Ga/Co in $CoGa_2O_4$ were determined using inductively coupled plasma optical emission spectrometry as 2.03 and 1.94, respectively, which are well matched to their theoretical stoichiometries. Moreover, the energy-dispersive X-ray spectroscopy (EDX) elemental mappings (Supplementary Figs. 4–6) indicate a uniform distribution of both the host cobalt and the substituted metal cations within these spinel oxides. $Zn^{2+}$/$Ga^{3+}$ with larger ionic radii than $Co^{2+}$/$Co^{3+}$ (0.58 vs. 0.60 Å, 0.61 vs. 0.62 Å) slightly altered the interplanar crystal spacing within the spinel structure (Supplementary Fig. 2b; Supplementary Table 1). This variation in interplanar crystal spacing induces covalency competition between $M_{Td}$–O (metal–oxygen bond in tetrahedral unit) and $M_{Oh}$–O (meta–-oxygen bond in octahedral unit) in the $M_{Td}$–O–$M_{Oh}$ backbone of the spinel crystal structure, affecting the exposed active sites and the electronic properties that govern the catalytic activity[25].

Scanning electron microscopy (SEM) observations reveal that these spinel oxides demonstrate a similar octahedral morphology and centered particle size distributions with average sizes ranging from 327 to 466 nm (Fig. 1b; Supplementary Fig. 7a–c). This centric particle size range offers a solid basis for evaluating the intrinsic reactivity differences arising from geometrical sites (Supplementary Fig. 8). We also prepared $MgCo_2O_4$ and $CoAl_2O_4$ via the same synthesis protocol. However, these two spinel oxides display different morphologies compared to the others (Supplementary Fig. 7d, e), which might be attributed to the large deviations in ionic radius between Co and the substituted Mg/Al cations[26].

The exposed crystal facets can act as the decisive factor in the activity of the metal-based catalysts by affecting both the reaction kinetics and pathways[27]. A rigorous investigation on the activity of the $Co^{2+}_{Td}$/$Co^{3+}_{Oh}$ geometrical sites requires pristine $Co_3O_4$ and its substitutes to have the same exposed crystal facet. In this study, the relative spatial positions of the lattice planes in reciprocal space were measured to determine their main exposed crystal facets by identifying the angles and spacings between the spots in selected area electron diffraction (SAED) images[28] (Supplementary Fig. 9). For the as-synthesized $Co_3O_4$, $ZnCo_2O_4$, and $CoGa_2O_4$, the fixed angle of 60° with the crystal spacing of 3.67 $nm^{-1}$ observed between the (220), (422), and (202) planes in their SAED patterns suggests that the [111] facet is the common exposed facet on their octahedrons[29] (Fig. 1d, e; Supplementary Fig. 10a, b). Additionally, atom-resolved surface structure characterizations and the lattice spacing measurements further confirm the common exposure of [111] facet for these spinel oxides[29] (Fig. 1c; Supplementary Fig. 10c, d).

Co sites symmetry and the valence states of pristine $Co_3O_4$ and its substitutes were investigated by X-ray absorption near-edge structure (XANES) (Fig. 1f). The intensity and position of the pre-edge peak (7709.1 eV) of Co K-edge spectrum provide clear insight into the symmetry of tetrahedral and octahedral sites within the spinel oxides because the highly non-centrosymmetry of the tetrahedral sites facilitates the $p$ to $d$ orbital transitions of the electrons[30]. Therefore, a narrower and more intense peak is expected in spinel oxides with a higher composition of tetrahedral sites than octahedral sites[31]. The inset in Fig. 1f shows that the pre-edge peak intensity for $CoGa_2O_4$ is greater than that for $ZnCo_2O_4$, suggesting that $Ga^{3+}$ substitution results in more Co occupation in tetrahedral sites, while replacing $Co^{2+}$ by $Zn^{2+}$ favors the exposure of the octahedral sites. Moreover, the transformation from tetrahedral to octahedral site symmetry also shifts the pre-edge peak to a slightly increased position (~0.3 eV).

Geometrical-site substitution also leads to variations of the main adsorption edges by affecting the valence states. $Ga^{3+}$ substitution shifts the main adsorption edge (7721.3 eV) towards a lower energy, while incorporating $Zn^{2+}$ raises the edge to a higher energy. This phenomenon suggests that the presence of Zn increases the oxidation state of Co while Ga drives Co to a reduced state[32]. The average oxidation states (AOS) of Co within $Co_3O_4$ and its substitutes were further

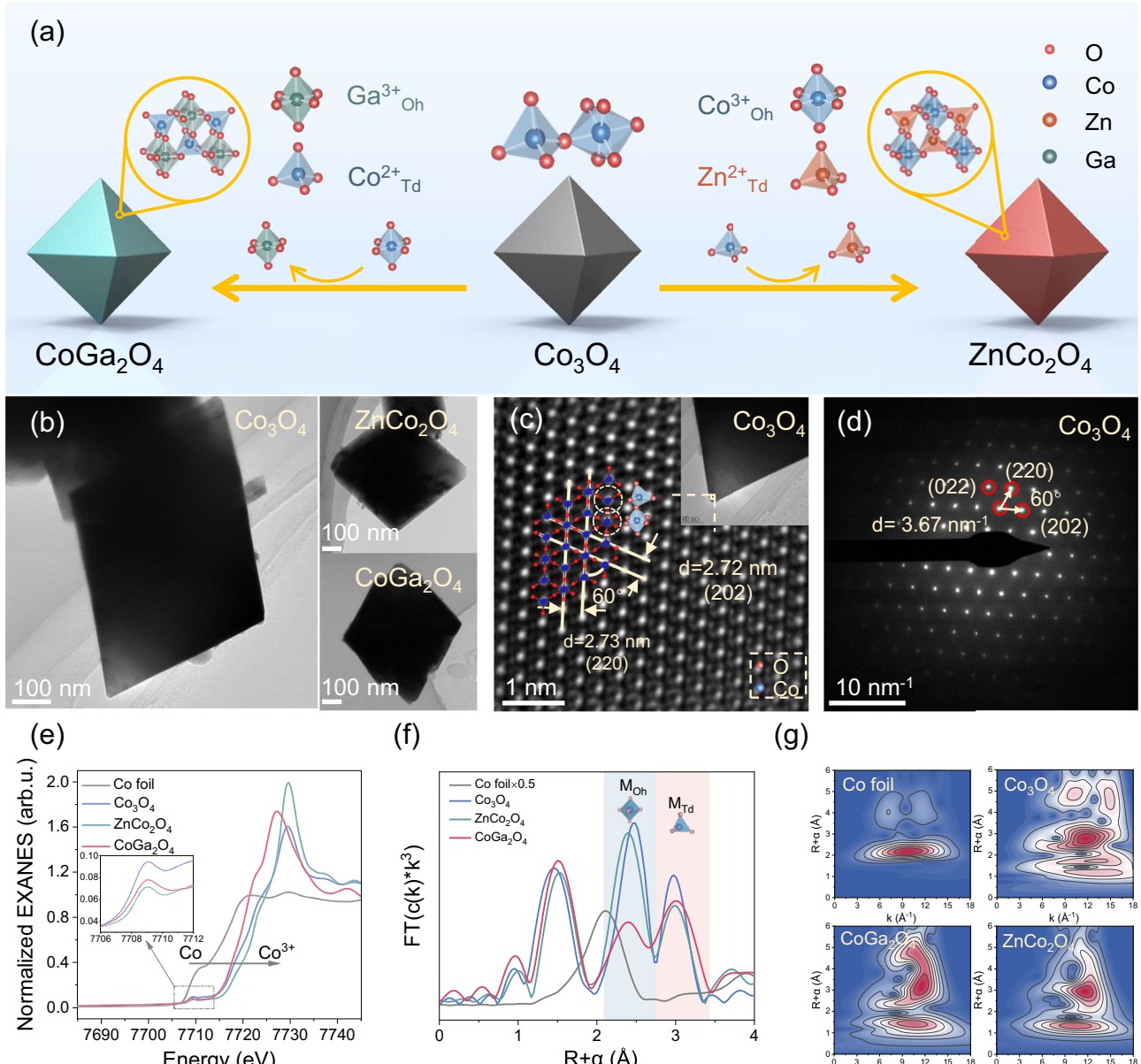

**Fig. 1 | Synthesis and structures of the catalysts. a** Schematic illustration of the synthetic process of the Co-based octahedral spinel oxides with $Co^{2+}$ and $Co^{3+}$ substitutions by $Zn^{2+}$ and $Ga^{3+}$, respectively. **b** Transmission electron microscopy (TEM) images of $Co_3O_4$, $ZnCo_2O_4$ and $CoGa_2O_4$ in [111] orientation. Atomic-resolved high-resolution TEM (HRTEM) image (**c**) and the corresponding selected area electron diffraction (SAED) image (**d**) of $Co_3O_4$. **e** Normalized Co K-edge X-ray absorption near-edge structure (XANES) spectra of the as-synthesized Co-based spinel oxides. Inset: enlarged features of the pre-edge peaks. **f** The corresponding Fourier transformed EXAFS (FT-EXAFS) spectra at R space. **g** EXAFS wavelet transforms of Co K-edge for Co foil and the as-synthesized Co-based spinels. Source data are provided as a Source Data file.

calculated by establishing a linear dependence between the half of the normalized absorbance level in the XANES spectrum ($E_{1/2}$) and Co oxidation states[25] (Supplementary Fig. 11). The calculated AOS values of Co species in $Co_3O_4$, $ZnCo_2O_4$, $CoGa_2O_4$ and $CoAl_2O_4$ are 2.50, 2.92, 2.07, and 2.11, respectively. These results verify the successful substitution of the $Co^{2+}_{Td}$ and $Co^{3+}_{Oh}$ in pristine $Co_3O_4$ by $Zn^{2+}$ and $Ga^{3+}/Al^{3+}$, respectively. X-ray photoelectron spectroscopy (XPS) analysis also corroborates these findings by examining the surface cobalt oxidation states (Supplementary Table 2; Supplementary Fig. 12).

Local geometric distributions of Co and the substitution cations in tetrahedral/octahedral sites of pristine $Co_3O_4$ and its substitutes were further deciphered by their Fourier-transform extended X-ray absorption fine structure (FT-EXAFS) (Fig. 1g) and wavelet transform

analyses (Fig. 1h). For pristine $Co_3O_4$ containing both $Co^{2+}_{Td}$ and $Co^{3+}_{Oh}$ sites, three main peaks located in-between 1 and 2, at ~2.5, and at ~3.0 Å are observed, corresponding to the distances of Co–O shells, octahedrally coordinated cations to its closest neighboring metal ions in octahedral sites ($Co_{Oh}$–$Co_{Oh}$), and the closest neighboring Co cations in octahedral and tetrahedral sites ($Co_{Oh}$–$Co_{Td}$), respectively. It is evident that the octahedrally coordinated cations exhibit two bonding modes with the surrounding octahedral (~2.5 Å) and tetrahedral sites (~3.0 Å). The tetrahedrally coordinated cations have only one bonding mode with the neighboring octahedral sites (~3.0 Å). Therefore, the identification of the peaks at ~2.5 and 3.0 Å in $ZnCo_2O_4$ suggests that the $Co^{3+}$ species are occupied in the octahedral sites. For $CoGa_2O_4$, despite that both peaks at ~2.5 and 3.0 Å are observed, the peak

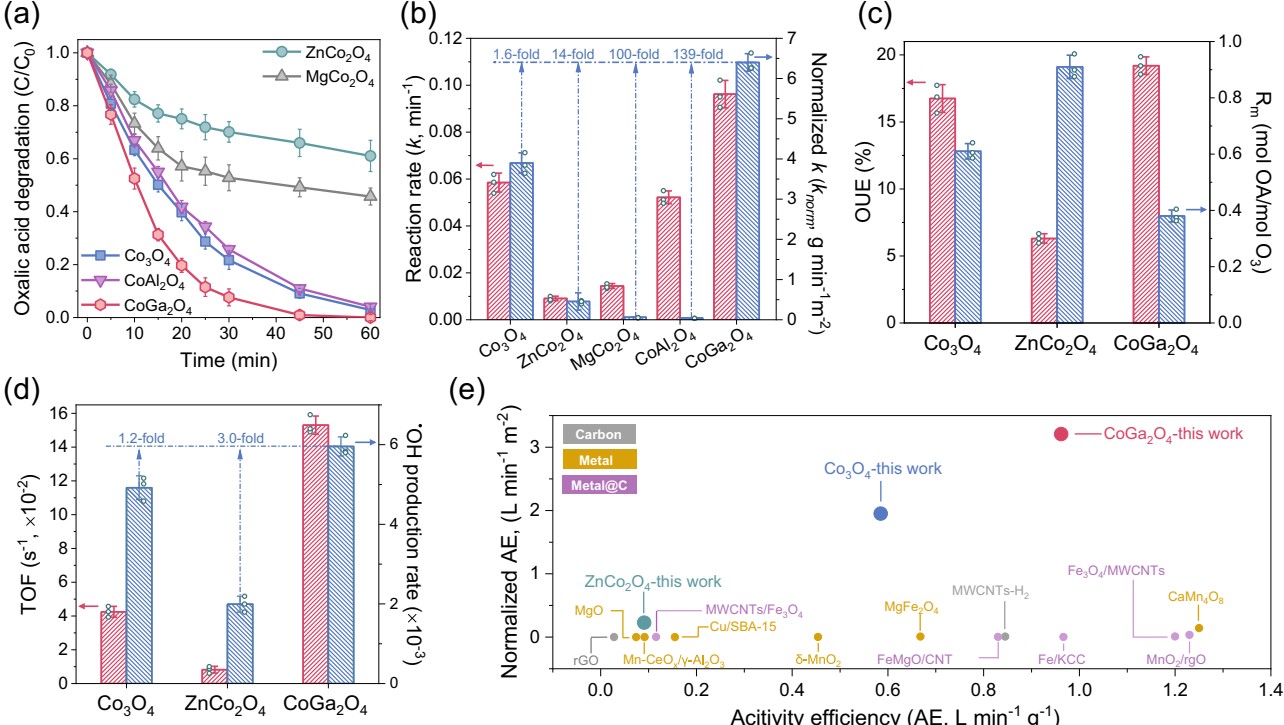

**Fig. 2 | Heterogeneous catalytic ozonation activities of the catalysts. a** Catalytic ozonation activities of the as-synthesized Co-based spinel oxides for oxalic acid (OA) degradation. Reaction condition: [catalyst] = 0.1 g L⁻¹; temperature: 25 °C; initial pH = 3, [OA]₀: 50 mg L⁻¹. **b** Comparison of the reaction rates (*k*s) and BET surface area normalized reaction rates (*k$_{norm}$*s) of different spinels/O₃ systems. **c** Comparison of ozone utilization efficiency (OUE) and R$_m$ values of different spinels/O₃ systems. **d** Comparison of turnover frequency (TOF) and •OH

production rates of different spinels/O₃ systems. **e** Comparison of the catalytic ozonation performance of our spinel oxide catalysts with values from previous reports. Catalysts include δ-MnO₂[62], CaMn₄O₈[63], MWCNTs[64], Fe₃O₄/MWCNTs[65], 2% Cu/SBA-15[66], Mn-CeO$_x$/γ-Al₂O₃[67], MgO[111][68], MgFe₂O₄[69], MnO₂/rGO[70], MWCNTs/Fe₃O₄[71], FeMgO/CNT[72], Fe/KCC[73], and rGO[74]. Error bars represent the standard deviations from three replicate measurements. Source data are provided as a Source Data file.

intensity at 2.5 Å is significantly lower than that in Co₃O₄ and ZnCo₂O₄, suggesting Co²⁺$_{Td}$ still exists as the governing sites. In addition, the significant variations in peak intensity ratios of Co$_{Oh}$–Co$_{Oh}$ path to Co$_{Oh}$–Co$_{Td}$ path for CoGa₂O₄ and ZnCo₂O₄ compared to that of Co₃O₄ suggested that the inversed Co²⁺$_{Oh}$/Co³⁺$_{Td}$ sites act as the minor coordination-geometry-competitors against the dominant Co²⁺$_{Td}$ and Co³⁺$_{Oh}$ sites (Supplementary Fig. 13). The acquired coordination numbers of Co cations by fitting EXAFS spectra (Supplementary Fig. 14; Supplementary Table 3) further reveal that Co³⁺$_{Oh}$ and Co²⁺$_{Td}$ are the prevailing geometric sites in ZnCo₂O₄ and CoGa₂O₄, respectively.

The coordination environments surrounding the substituted Zn and Ga were also investigated. The fully occupied *3d* orbitals of Zn and Ga restrict their valence states to +2 and +3 accordingly when incorporated into the Co₃O₄ crystal scaffolds, as revealed by their high-resolution XPS spectra (Supplementary Fig. 15). Additionally, the acquired Ga K-edge XANES spectrum and the corresponding fitted EXAFS results indicate that Ga³⁺ substituted the Co³⁺ position and exhibited both octahedral and tetrahedral bonding modes (Supplementary Fig. 16; Supplementary Table 4), similar to those of Co³⁺ in ZnCo₂O₄. These results suggest that Zn²⁺ and Ga³⁺ substitution did not alter the spinel scaffolds and that Co²⁺$_{Td}$ and Co³⁺$_{Oh}$ were the only forms of the Co²⁺ and Co³⁺ in CoGa₂O₄ and ZnCo₂O₄, respectively.

**Catalytic ozone activation is dependent on the physicochemical properties**

Catalytic ozonation efficiencies of pristine Co₃O₄ and its geometric site substitutes were evaluated using oxalic acid (OA) as the target probe (Supplementary Table 5). OA is highly recalcitrant to the attack by O₃ ($k_{O_3}$ = 0.04 M⁻¹ s⁻¹) and those ROSs with a weak oxidation capacity,

such as superoxide radical (O₂•⁻) and singlet oxygen (¹O₂), but can be readily destructed by •OH ($k_{•OH}$ = 1.4 × 10⁶ M⁻¹ s⁻¹)[33,34]. The adsorption abilities of OA by the as-synthesized spinel oxides were investigated prior to measuring their catalytic ozonation activity (Supplementary Fig. 17). Marginal OA adsorption (<3%) is observed for all the spinel oxides. Chemical drift measurements reveal that the spinel oxides exhibit a similar point of zero charges (pH$_{pzc}$) of 7 (Supplementary Fig. 18), suggesting that the catalysts are protonated at solution pH 3. Although an electrostatic attraction between the surface of spinel oxides and the negatively charged OA molecules (pK$_{a1}$ = 1.2) is expected, the low specific surface areas (SSAs) of the spinel oxides hinder further adsorption (Supplementary Table 6; Supplementary Fig. 19). This electrostatic attraction, however, facilitates the interactions between O₃ and OA molecules on the catalyst surface in catalytic reactions.

In the catalytic O₃ activation, geometrical-site-dependent activity is observed. CoGa₂O₄ with substituted Co³⁺$_{Oh}$ sites displays a higher performance to the pristine Co₃O₄ in OA destruction, achieving over 95% destruction of the initial OA (50 mg L⁻¹) within 45 min (Fig. 2a, details on optimizing the catalytic loading/O₃ dosage and influence of initial solution pH are provided in Supplementary Figs. 20 and 21, respectively). Contrarily, replacing the Co²⁺$_{Td}$ sites by Mg²⁺ and Zn²⁺ drastically decreases the OA degradation efficiency, with ~46 and 61% of OA remaining after 60 min. By normalizing the pseudo-first reaction kinetics (*k*, min⁻¹) of the spinel oxides with their SSAs, a tremendous increase (12.4−81.3-fold) in the apparent reaction kinetics (*k$_{norm}$*, g min⁻¹ m⁻²) is observed for Co²⁺$_{Td}$-dominated CoGa₂O₄ compared to those of ZnCo₂O₄ and MgCo₂O₄ with Co³⁺$_{Oh}$ as the prevailing sites (Fig. 2b; Supplementary Table 7). This suggests that Co²⁺$_{Td}$ sites exhibit a higher activity in O₃ activation than Co³⁺$_{Oh}$ (Supplementary Fig. 22).

Furthermore, the trivial metal leaching of these as-synthesized spinels from inductively coupled plasma mass spectrometry results indicates the robust crystal structures against the highly-oxidative reaction environment created by $O_3$ and the produced ROS (Supplementary Table 8). Noted that the high SSA by severe agglomerations of $CoAl_2O_4$ nanoparticles results in a much lower $k_{norm}$ than that of $CoGa_2O_4$. Additionally, the morphological differences between $CoAl_2O_4$/ $MgCo_2O_4$ and other spinels also lead to the exposure of facets with varying reactivities. Therefore, the subsequent investigations primarily focus on $Co_3O_4$ and its geometrical-site substitutes with comparable SSAs and identical exposure of [111] facet (i.e., $CoGa_2O_4$ and $ZnCo_2O_4$).

Considering the low SSAs of the as-synthesized catalysts (~0.3 $m^2 g^{-1}$) which might not facilitate the exposure of the active sites, we subsequently synthesized three-dimensional ordered macroporous $Co_3O_4$ (3DOM-$Co_3O_4$), plate $Co_3O_4$, and spherical $Co_3O_4$, achieving higher SSAs of 29.1, 17.5, and 78.4 $m^2 g^{-1}$ accordingly (Supplementary Fig. 23) to investigate the effect of SSAs on catalytic activity. Plate $Co_3O_4$ and spherical $Co_3O_4$ display lower activity than that of $CoGa_2O_4$ (Supplementary Fig. 24), alongside significantly higher cobalt leaching (0.58, 0.66, and 0.09 $mg L^{-1}$ for plate $Co_3O_4$, spherical $Co_3O_4$, and $CoGa_2O_4$, respectively). 3DOM-$Co_3O_4$ obtains a similar activity to $CoGa_2O_4$, yet simply increasing exposure amount of the active sites did not enhance their intrinsic activity. Both the $k_{norm}$ and the electrochemical surface area (ECSA) normalized reaction kinetics ($k_{ECSA}$) of 3DOM-$Co_3O_4$ are significantly lower than those of $CoGa_2O_4$ (59-fold and 8.7-fold, respectively) (Supplementary Fig. 25). Similarly, the intrinsic activity ($k_{ECSA}$s) of Co sites in plate $Co_3O_4$ and spherical $Co_3O_4$ are 13 and 39-fold less than that of $CoGa_2O_4$. These suggest the great significance of regulating coordination geometry in Co-based spinel outperforms SSA engineering in determining the intrinsic activity of the active sites.

The high activity of $CoGa_2O_4$ is accompanied by a 1.15-fold improvement in $O_3$ utilization efficiencies (OUE) compared to the pristine $Co_3O_4$ (Fig. 2c; Supplementary Fig. 26). In contrast, substituting the active $Co^{2+}$ with catalytically inactive $Zn^{2+}$ dramatically reduced the OUE. This trend is also discovered in the variations in molar ratios of the degraded OA to the consumed $O_3$ ($R_m$). The lowest $R_m$ value of $CoGa_2O_4$ among the as-synthesized spinel oxides indicates that accelerated conversion kinetics of $O_3$ into ROS on $Co^{2+}_{Td}$ sites. As a result, the calculated turnover frequency (TOF) (Fig. 2d; Supplementary Table 9) for $CoGa_2O_4$ with the high exposure of $Co^{2+}_{Td}$ sites is 3.5- and 17.6-fold greater than that of $Co_3O_4$ with mixed $Co^{2+}_{Td}$/ $Co^{3+}_{Oh}$ sites and $ZnCo_2O_4$ with $Co^{3+}_{Oh}$-dominant sites (0.15, 0.043, and 0.0082 $s^{-1}$), respectively. In addition, $CoGa_2O_4$ synthesized in this study, despite with small SSAs, are among the highest reported (Fig. 2e; Supplementary Data 1). This high intrinsic activity of $CoGa_2O_4$ can be attributed to the selective exposure of the active $Co^{2+}_{Td}$ sites via coordinating environment regulation. $Co^{2+}_{Td}$ sites with a greater tendency to donate the frontier electrons facilitate the electrophilic activation of $O_3$ molecules than $Co^{3+}_{Oh}$ sites[16,21].

Influence of the exposed crystal facet on catalytic activity was also investigated by synthesizing cubic-shaped $Co_3O_4$ exposing the [100] crystal facet as the reference catalyst (Supplementary Fig. 27). Results suggest that octahedral-shaped $Co_3O_4$ with the [111] crystal facet exhibits a higher intrinsic activity than the cubic-shaped $Co_3O_4$ with the [100] crystal facet. Although the sparse distribution of Co atoms in the [100] facet results in a greater surface defective structure than the [111] facet, the larger number of the active $Co^{2+}_{Td}$ sites on the [111] facet governs the catalytic activity (Supplementary Fig. 28; Supplementary Table 10)[35,36]. This further discloses the decisive role of geometric coordination sites in catalytic activity. Furthermore, the inversed $Co^{2+}_{Oh}$ sites as the minor coordination-geometry-competitors against the dominant $Co^{2+}_{Td}$ sites demonstrate a markedly lower intrinsic catalytic activity than the $Co^{2+}_{Td}$ sites, which can be ascribed to the inefficient electron transfer ability of octahedrally coordinated O atoms $Co^{2+}_{Oh}$ sites compared to tetrahedrally coordinated O atoms in $Co^{2+}_{Td}$ sites (Supplementary Fig. 29)[21].

Apart from the geometrical sites, surface chemical features and electronic properties of TM oxides might affect $O_3$ adsorption and its subsequent activation[37,38]. To disclose the effects of these properties on catalytic ozonation activity, structural-activity relationships are established based on the obtained physicochemical properties from the temperature programmed desorption (TPD), cryo-electron paramagnetic resonance (EPR), and electrochemical impedance spectroscopy (EIS) tests (Supplementary Fig. 30), which were summarized in a heatmap in Supplementary Fig. 31. Although geometric site substitutions alter the surface chemical properties compared to those of pristine $Co_3O_4$, these variations in physicochemical properties do not act as pivotal factors in determining the catalytic activities of the spinel oxides. These results further reinforce the fundamental roles of the unsubstituted host Co cation sites in determining the intrinsic activity for catalytic ozonation.

## Selective formation of ROSs in the catalytic ozonation processes by spinel oxides

The type of ROSs produced by $Co_3O_4$ and its geometrical-site substitutes were then distinguished by quenching tests and probe-based in situ spectroscopies. The high accuracy and selectivity of these techniques for ROS detection have been verified using the classic homogeneous systems designed to produce single type of ROS (Supplementary Fig. 32). Methanol (MeOH) and acetic acid (AA) which exhibit high reaction kinetics for ·OH ($k_{·OH} = 9.7 \times 10^8$ and $8.5 \times 10^7 M^{-1} s^{-1}$, respectively)[34] but are recalcitrant for $O_3$ oxidation were employed as the ·OH scavengers (Supplementary Figs. 33–35). The presence of these scavengers significantly hindered OA degradation kinetics for $Co_3O_4$, $CoGa_2O_4$, and $ZnCo_2O_4$ (Fig. 3a), suggesting that OA destruction was dominantly driven by the produced ·OH. In situ EPR tests proved the generation of ·OH by identifying the DMPO−·OH signals with an intensity ratio of 1:2:2:1 (Fig. 3b). Furthermore, the intensity order of the DMPO−·OH signal for pristine $Co_3O_4$ and its substitutes closely correlated with their activities, solidifying that $Co^{2+}_{Td}$ sites stimulated the $O_3$ activation to produce a higher amount of ·OH than $Co^{3+}_{Oh}$ sites. The production of ·OH on the surface of spinel oxides was then visualized by employing coumarin as the probe[39], characterized by fluorescent blue color accumulation on/near the catalysts on inverted fluorescence microscopy (IFM) images. Among the spinels, $CoGa_2O_4$ displayed the highest fluorescence intensity with faint blue halos around its surface (Fig. 3c)[40]. This suggests that the abundant ·OH on $CoGa_2O_4$ surface can activate the ambient water molecules into ·OH because of its high oxidation capability, which is confirmed by the higher open-circuit potential (OCP) than those of $Co_3O_4$ and $ZnCo_2O_4$ from in situ electrochemical analysis (Supplementary Fig. 36).

The surface interactions of $O_3$ with spinel oxides were examined by using dimethyl sulfoxide (DMSO) as the quenching agent. For $CoGa_2O_4$ with exposed $Co^{2+}_{Td}$ sites, DMSO addition results in a similar quenching effect to those of ·OH scavengers, suggesting that the surface-adsorbed $O_3$ on $CoGa_2O_4$ exhibits fast reaction kinetics for ·OH evolution and subsequent release into the bulk solution. Adding DMSO to the $ZnCo_2O_4$/$O_3$ system resulted in greater inhibition effect than those of ·OH scavengers. The weak interactions between $O_3$ and the $Co^{3+}_{Oh}$ sites on $ZnCo_2O_4$ hampered the swift $O_3$ conversion into ·OH and might initiate the production of singlet oxygen ($^1O_2$) as the byproducts. The inferior oxidation potential (0.81 $V_{NHE}$) of $^1O_2$ results in sluggish reactions with OA[41]. Singlet oxygen sensor green (SOSG) was employed as a selective probe for $^1O_2$ detection (Fig. 3d). For $ZnCo_2O_4$, a strong fluorescence signal at 525 nm was observed, suggesting the formation of a large amount of $^1O_2$[42]. This fluorescent intensity decreased for pristine $Co_3O_4$ and vanished for $CoGa_2O_4$ with exposed $Co^{2+}_{Td}$ sites, indicating that negligible or no $^1O_2$ was formed

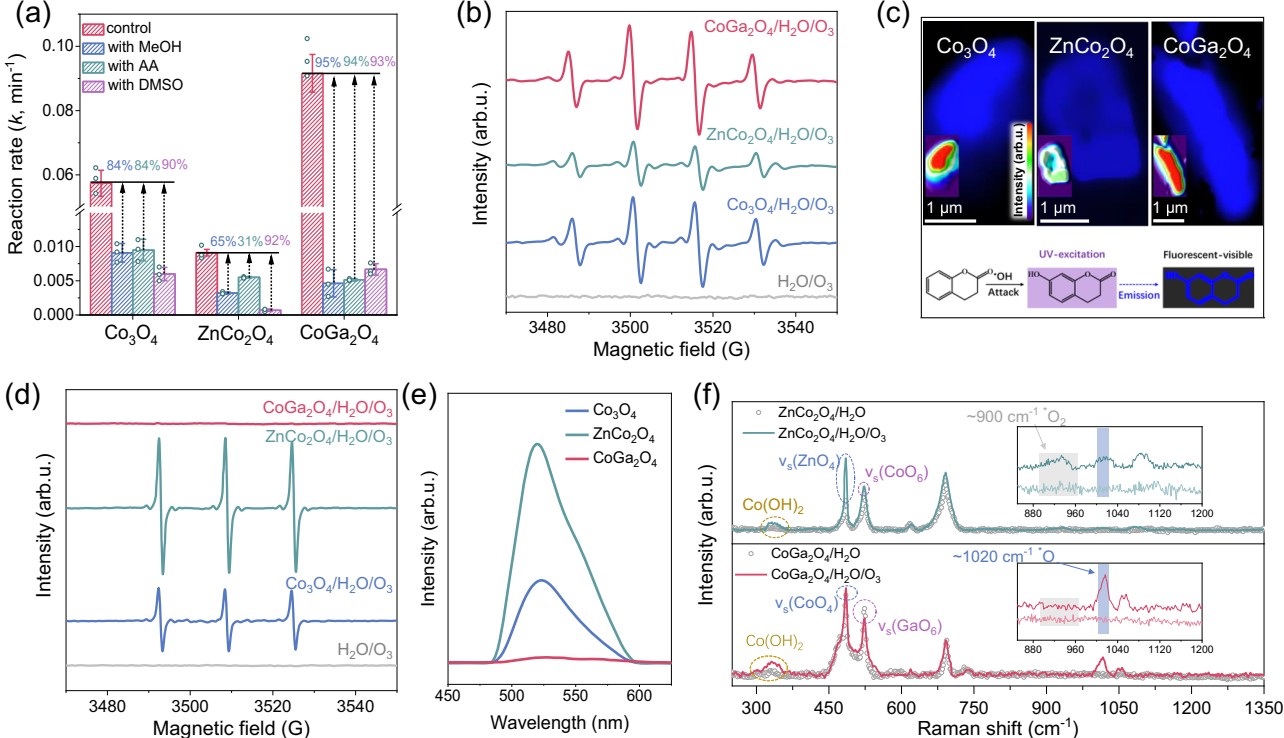

**Fig. 3 | Selective ROSs generation on different Co sites. a** Comparison of rate constants of quenching tests in different spinels/$O_3$ systems. Reaction conditions: [catalyst]: 0.1 g L$^{-1}$; temperature: 25 °C; initial pH = 3, [OA]$_0$: 50 mg L$^{-1}$, [methanol, MeOH]$_0$: 10 mM, [acetic acid, AA]$_0$: 10 mM, [dimethyl sulfone, DMSO]$_0$: 1 mM. **b** Electron paramagnetic resonance (EPR) spectra with 5,5-dimethyl-1-pyrroline (DMPO) as the spin trapping agent for ·OH detection. **c** Visualization of the generated ·OH at the surface of spinel oxides using coumarin as the ·OH

fluorescence probe by inverted fluorescence microscopy (IFM). **d** EPR spectra with 2,2,6,6-tetramethyl-4-piperidinol (TEMP) as the spin trapping agent for $^1O_2$ detection. **e** Photoluminescence spectra of singlet oxygen sensor green (SOSG) as the $^1O_2$ probe. **f** In situ Raman spectra of $ZnCo_2O_4$ and $CoGa_2O_4$. Error bars represent the standard deviations from three replicate measurements. Source data are provided as a Source Data file.

on the spinels with exposed $Co^{2+}_{Td}$ sites. Results of the in situ EPR tests with 2,2,6,6-tetramethyl-4-piperidinol (TEMP) as the spin trapping agent displayed a similar trend to that of the SOSG tests (Fig. 3e), with $ZnCo_2O_4$ displaying a high signal intensity compared to the negligible intensity observed for $CoGa_2O_4$. The above results suggested that $Co^{2+}_{Td}$ sites exhibit a strong selectivity for ·OH generation while $Co^{3+}_{Oh}$ sites can induce both ·OH and $^1O_2$ production.

In situ Raman tests reveal a strong signal at ~1020 cm$^{-1}$ in the $CoGa_2O_4/O_3$ system, which can be ascribed to the produced surface-adsorbed single O atoms (*O) stemming from the catalytic dissociation of the $O_3$ on the $Co^{2+}_{Td}$ sites (Fig. 3f). Moreover, the $CoGa_2O_4/O_3$ system obtains a higher signal intensity at ~300 cm$^{-1}$ than the pristine $CoGa_2O_4$ without a reaction[43], suggesting that a large amount of surface-bound ·OH were formed during the catalytic ozonation process. In contrast, the lower Raman signals of 1020 and 300 cm$^{-1}$ observed in the $ZnCo_2O_4/O_3$ system than those in the $CoGa_2O_4/O_3$ system reveal that less *O and surface-bound ·OH are produced because of the weak interaction between the $Co^{3+}_{Oh}$ sites and $O_3$. Additionally, a peak corresponding to the surface-adsorbed $O_2$ species (*$O_2$) at ~900 cm$^{-1}$ emerges in the $ZnCo_2O_4/O_3$ system, indicating that side reactions occur on the $Co^{3+}_{Oh}$ sites. We further evaluate the variations of Raman peaks for the stretching vibrations of the $Co^{2+}_{Td}$ (~486 cm$^{-1}$) and $Co^{3+}_{Oh}$ (~525 cm$^{-1}$) sites after ozone introduction, which were originated from the stretching of the Co atoms by complexing with $O_3$ and the subsequent geometrical-site-involved lattice interactions. Clearly, introducing $O_3$ to $CoGa_2O_4$ induced less variations in the geometrical sites-involved stretching vibrations than those of $ZnCo_2O_4$. This suggests a robust bonding behavior within the lattice frameworks and a minimal change in the surface chemical environment after Ga substitution.

The selectivity of the $Co^{2+}_{Td}$ and $Co^{3+}_{Oh}$ sites on ·OH production was assessed by the ·OH/$O_3$ conversion rates. Compared to the catalytically inactive $ZnCo_2O_4$, the highly active $CoGa_2O_4$ obtained a 3.0-fold increase in ·OH/$O_3$ conversion rate (Fig. 2d), confirming the high selectivity of $Co^{2+}_{Td}$ sites towards ·OH production. The production of $^1O_2$ as the byproduct ROS might stem from the distinct $O_3$ adsorption and intermediate evolution routes on $Co^{3+}_{Oh}$ sites, which require higher electron consumption and result in a decreased $O_3$ utilization efficiency.

## Origin of the ozone-metal interaction and the pathways of ROSs formation and evolution

To illustrate the intrinsic electronic states of the spinel oxides and hybridization strength between Co $3d$ orbitals and O $2p$ orbitals from $O_3$ molecules and their influences on activity, we constructed $Co_3O_4$, $CoGa_2O_4$, and $ZnCo_2O_4$ models with the same exposure of [111] facet, based on the deciphered results from XAS and superconducting quantum interference device magnetization analysis (Supplementary Figs. 37 and 38; Supplementary Table 11), for use in the density functional theory (DFT) calculations. As revealed in the crystal orbital Hamilton population (COHP) analysis, the strong covalency competition arising from the substitution of $Ga^{3+}$ or $Zn^{2+}$ into crystal structure crystal favors the exposure of the unsubstituted cations $Co^{2+}_{Td}$ and $Co^{3+}_{Oh}$ as the preferentially exposed active sites in $CoGa_2O_4$ and $ZnCo_2O_4$[12] (Supplementary Fig. 39). The calculation outcomes well match the XAS analysis results, supporting the validity of the constructed models. Given the electron-rich nature of $O_3$ molecules, all molecular orbitals (MOs) and partial of anti-bonding molecular orbitals (MO*s) can be occupied when $O_3$ hybridizes on either $Co^{2+}_{Td}$ or $Co^{3+}_{Oh}$ sites (Supplementary Fig. 40), making the occupancy of MO*s

crucial to determine the electron transfer resistance and hybridization strength[44]. Projected electronic density of states (PDOS) profiles on Co $3d$ orbitals and the corresponding $d$-band center analysis reveal that the high spin polarization resulting from the covalency competition positions the band centers of spin-down channels ($\varepsilon_{d\downarrow}$) of the spinel oxides at higher energy levels than those of spin-up channels ($\varepsilon_{d\uparrow}$), endowing spin-down channels a decisive role in communicating with the $O_3$ $2p$ orbitals for MO*s formation (Supplementary Fig. 41). Based on the orbital analysis results, orbital overlapping diagrams are plotted (Fig. 4a, b). MO*s formed by hybridization of spin-down channel of $Co^{2+}_{Td}$ sites and $O_3$ $2p$ orbitals (MO*s- $Co^{2+}_{Td}$) are situated closer to the Fermi level than those formed by $Co^{3+}_{Oh}$ sites. This facilitates the electron transition at $Co^{2+}_{Td}$ sites and decreases the energy barriers for $O_3$ dissociation, which promotes the production of reactive intermediates[45].

The interaction of $O_3$ molecules with the exposed $Co^{2+}_{Td}$ or $Co^{3+}_{Oh}$ sites during the ozone activation process results in distinctly different electronic states. $O_3$ adsorption (transition state, TS) and dissociation (final state, FS) on $Co^{2+}_{Td}$ sites result in the movement a larger portion of electrons across the Fermi level ($\varepsilon_f$) than those on $Co^{3+}_{Oh}$ sites (Fig. 4c, d), confirming the higher occupancy of MO*s- $Co^{2+}_{Td}$ than those formed by $Co^{3+}_{Oh}$ orbitals. Charge density difference (CDD) calculations reveal that the terminal O atom in $O_3$ is more favorable to abstracting electrons from the $Co^{2+}_{Td}$ site than the $Co^{3+}_{Oh}$ site (Fig. 4e), which aligns well with the results of Bader charge analyses (Supplementary Figs. 42–45). Consequently, a stronger chemisorption between $Co^{2+}_{Td}$ and $O_3$ than that of $Co^{3+}_{Oh}$ is achieved, providing the adsorption energies of 0.41 and 0.32 eV, respectively. The higher bond order between $Co^{2+}_{Td}$ $3d$ orbital and O $2p$ orbital than that of $Co^{3+}_{Oh}$ (2 vs. 1.5) (Supplementary Fig. 46) further verifies the greater hybridization strength at the $Co^{2+}_{Td}$ sites, which promotes the electronic communications. The magnetic moments for $O_{2\ free}$ generated on $Co^{2+}_{Td}$ and $Co^{3+}_{Oh}$ sites are calculated as 2 and 0 $\mu_B$ accordingly, corresponding to the normal $O_2$ molecule and the $^1O_2$ molecule, respectively. This agrees with the results of ROS identification, suggesting the selectivity of the geometrical sites in ROS production. Bader charge analysis reveals that the $O_3$ dissociation into $^1O_2$ on $Co^{3+}_{Oh}$ sites requires a greater number of electrons than on $Co^{2+}_{Td}$ sites, which explains the low OUE and ·OH production rate on $Co^{3+}_{Oh}$ sites.

$O_3$ activation for ·OH evolution on spinel oxides mainly involves four elementary reactions[46,47]. (1) $O_3$ adsorption on the active site to form surface-adsorbed $O_3$ (*$O_3$); (2) catalytic dissociation of *$O_3$ to form a surface-adsorbed O atom (*$O_{ad}$) and a free $O_2$ molecule ($O_{2\ free}$); (3) interaction of *$O_{ad}$ with the ambient $H_2O$ molecule (*$O_{ad}$···$H_2O$); and (4) cleavage of the adsorbed $H_2O$ molecule for ·OH production (*$OH$··· ·OH). In both TSs for *$O_{ad}$ formation (TS$_1$) and surface-confined ·OH production (TS$_2$), which govern the tendency for $O_3$ dissociation and ·OH evolution accordingly, $Co^{2+}_{Td}$ sites obtain lower energy barriers than $Co^{3+}_{Oh}$ sites (Fig. 4f, g). Additionally, the energy released during both $O_3$ and $H_2O$ dissociation steps is greater on $Co^{2+}_{Td}$ sites than $Co^{3+}_{Oh}$ sites. These further solidify that MO*s-$Co^{2+}_{Td}$ sites near the Fermi level facilitate the electron injection, enhancing the reaction thermodynamics for catalytic $O_3$ dissociation and ·OH formation. Previous studies have reported that the *$O_{ad}$ attached to active sites after $O_3$ dissociation can also bond with another $O_3$ for ·OH generation[48] (Fig. 4h). However, this ·OH evolution route is complicated and requires higher $O_3$ and electron consumptions, making it thermodynamically unfavorable compared to the direct $H_2O$ oxidation (Supplementary Fig. 47).

## Practical application potentials of the spinel-induced catalytic ozonation process

$CoGa_2O_4$ with the highest catalytic ozonation activity owing to the maximized exposure of $Co^{2+}_{Td}$ was employed to evaluate the application potentials. Cyclic experiments reveal that $CoGa_2O_4$ exhibits a high

reusability, maintaining over 95% of its initial catalytic activity after 5 cycles (Fig. 5a; Supplementary Fig. 48). In contrast, both $Co_3O_4$ and $ZnCo_2O_4$ were partially deactivated after multiple runs. XPS deconvolution results on O 1s spectra suggest that $ZnCo_2O_4$ demonstrated a more significant change in the peaks related to -OH/OV for the fresh/ used samples than that of the $CoGa_2O_4$ (Supplementary Fig. 49). In addition, $O_2$-TPD profiles further reveal that the area of the OV peak at 200–300 °C decreases remarkably for the used $ZnCo_2O_4$, yet minor changes are observed for the fresh/used $CoGa_2O_4$. Therefore, the high surface rigidity with marginal metal leaching (0.3 wt.%) accounts for the superior stability of $CoGa_2O_4$.

Influences of inorganic anions on catalytic performance were then evaluated. The presence of inorganic anions with the concentrations between 10 and 100 mM induces minor inhibitions in activities (Fig. 5b; Supplementary Fig. 50). It is suggested that surface-bounded ·OH produced by the strong hybridization between $Co^{2+}_{Td}$ and $O_3$ via the inner-sphere complexation accounts for the high adaptation towards inorganic anions. In contrast, the weak interaction between $Co^{3+}_{Oh}$ sites and $O_3$ resulted in the outer-sphere complexation, which can be heavily interfered with by ionic strength (Supplementary Fig. 51). We also investigated the catalytic ozonation performance of $CoGa_2O_4$ to degrade various recalcitrant phenolic contaminants (Fig. 5c; Supplementary Figs. 52 and 53). Compared to single ozonation process, the $CoGa_2O_4/O_3$ system achieved both high reaction rates and total organic carbon (TOC) mineralization rates for the treatment of phenolics with different ionization potentials[49]. This can be ascribed by the high OUE and ·OH production efficiency $CoGa_2O_4$ via the efficient electron transfer between $Co^{2+}_{Td}$ and $O_3$.

To further evaluate the practical application potential of $CoGa_2O_4/O_3$ system, $CoGa_2O_4$ was immobilized on cleaned, defatted cotton in a continuous-flow fixed-bed column reactor (0.25 L working volume, with residence time of 5 min) to treat the air flotation effluent wastewater of a petrochemical plant in southern China with initial COD of 108 mg L$^{-1}$ (Fig. 5d; Supplementary Fig. 54). The detailed properties of the real wastewater are listed in Supplementary Table 12. The effluent COD decreases to lower than 50 mg L$^{-1}$ (with a removal efficiency of 46%) within 90 min treatment and reaches 41 mg L$^{-1}$ after 6 h treatment with a corresponding TOC value of 20 mg L$^{-1}$ (Fig. 5e). These meet the Discharge Standard of Pollutants for Petroleum Refining Industry in China (GB 31570-2015, COD < 60 mg L$^{-1}$, TOC < 20 mg L$^{-1}$). Additionally, the time-dependent variation profile of UV$_{254}$ indicates that the $CoGa_2O_4/O_3$ system achieves an effective removal of aromatic contaminants within 90 min of the treatment (Fig. 5f). Furthermore, 3D-EEM fluorescence spectra suggests that two major fluorescence peaks centered at A (Ex/Em maxima at 275–300/325–375 nm) and B (Ex/Em maxima 225–250/352–400 nm) are discerned in the untreated petrochemical wastewater (Fig. 5g), attributed to aromatic N-containing compounds and petroleum-derived dissolved organic matters (DOM), respectively[50]. After treatment, the peaks in the 3D-EEM spectra nearly disappeared, indicating the exceptional ability of the $CoGa_2O_4/O_3$ system to destroy a broad band refractory organic contaminants within the real petrochemical wastewater.

Economic evaluations on the catalyst synthesis costs and the EE/O (electrical energy per order) suggest that the cobalt-based spinels synthesized in this study have a great advantage on technological economy than the other types of $Co_3O_4$-based catalysts (Fig. 5h; Supplementary Fig. 55; Supplementary Data 2; and Supplementary Table 13). For future applications, $Ga^{3+}$ can be replaced by more economical metals. Facile treatments, such as defect engineering and heteroatom doping, can also be applied to the cobalt-containing spinels. The optimized ambient microenvironments surrounding the active $Co^{2+}_{Td}$ sites can enhance the activity by inducing more intense electronic communications and stronger complexation with the $O_3$ molecules, which thereby trigger the increased production of surface-bound ·OH that are recalcitrant to inorganic anions in wastewater.

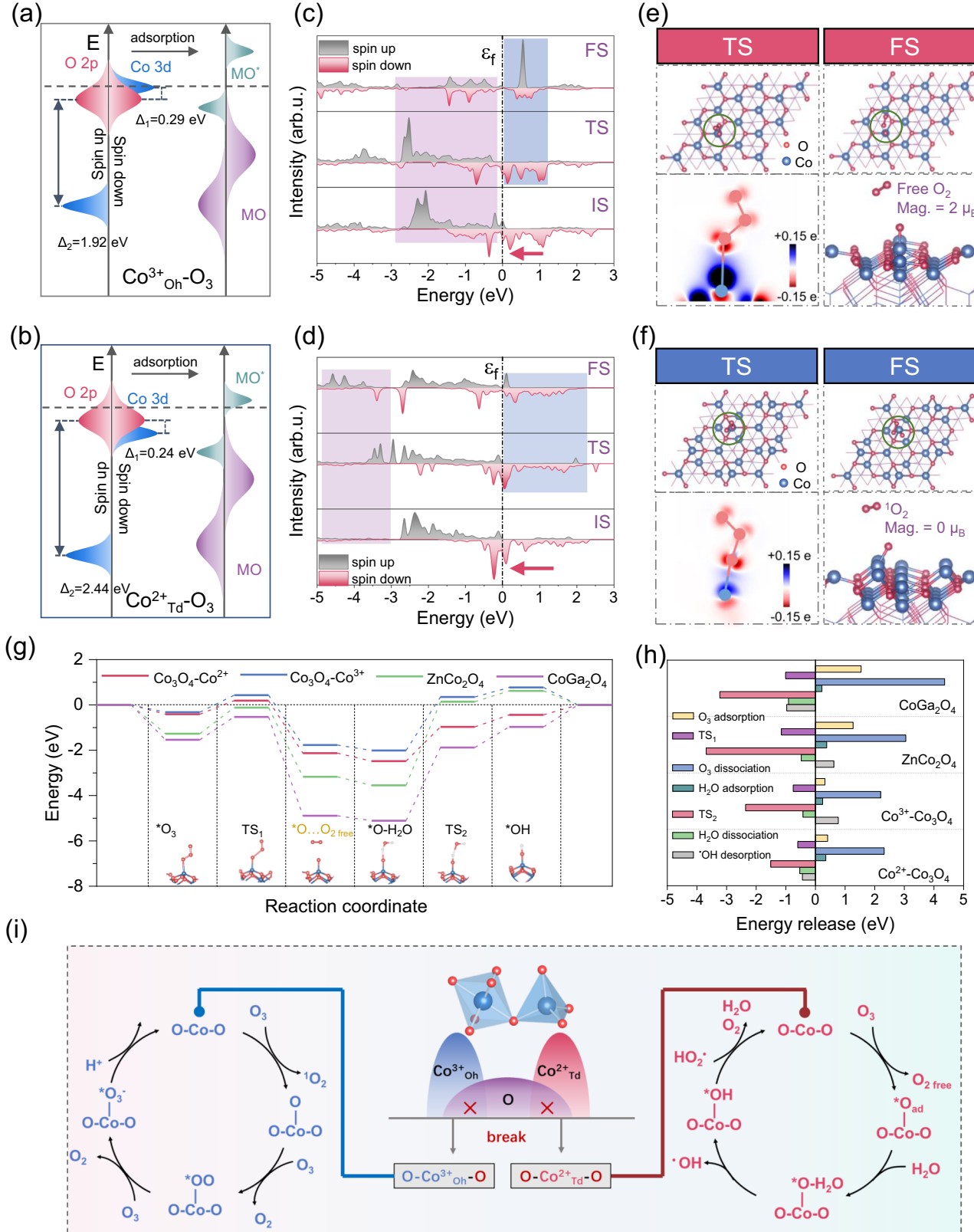

**Fig. 4 | Mechanism investigation for activity and selectivity differences.** Schematic illustration of molecular orbital formation between $Co^{2+}_{Td}$ (**a**)/$Co^{3+}_{Oh}$ (**b**) sites and $O_3$ molecules based on the projected electronic density of states (PDOS) profiles on Co *3d* orbitals and the corresponding *d*-band center analysis. PDOS profiles of $O_3$ adsorption at the $Co_3O_4$-$Co^{2+}_{Td}$ site (**c**) and the $Co_3O_4$-$Co^{3+}_{Oh}$ site (**d**) of initial state (IS), transition state (TS), final state (FS). $O_3$ adsorption configurations at TS and FS and the corresponding charge density difference (CDD) plots at the $Co_3O_4$-$Co^{2+}_{Td}$ site (**e**) and the $Co_3O_4$-$Co^{3+}_{Oh}$ site (**f**). Relative energy profiles and

the simplified surface structures of the various reaction species for reaction pathway over spinel oxides (**g**) and the corresponding computed energy releases between different transition states (**h**). **i** Proposed reaction process for ˙OH generation on different Co sites. At $Co^{2+}_{Td}$ sites, decomposition of $O_3$ generates $O_2$ and surface-adsorbed O atom (˙$O_{ad}$) with a high oxidation potential, which oxidizes water to generate ˙OH. In contrast, $Co^{3+}_{Oh}$ sites preferentially produce $^1O_2$, while the remaining ˙$O_{ad}$ with a low oxidation potential fails to further react with water to generate ˙OH. Source data are provided as a Source Data file.

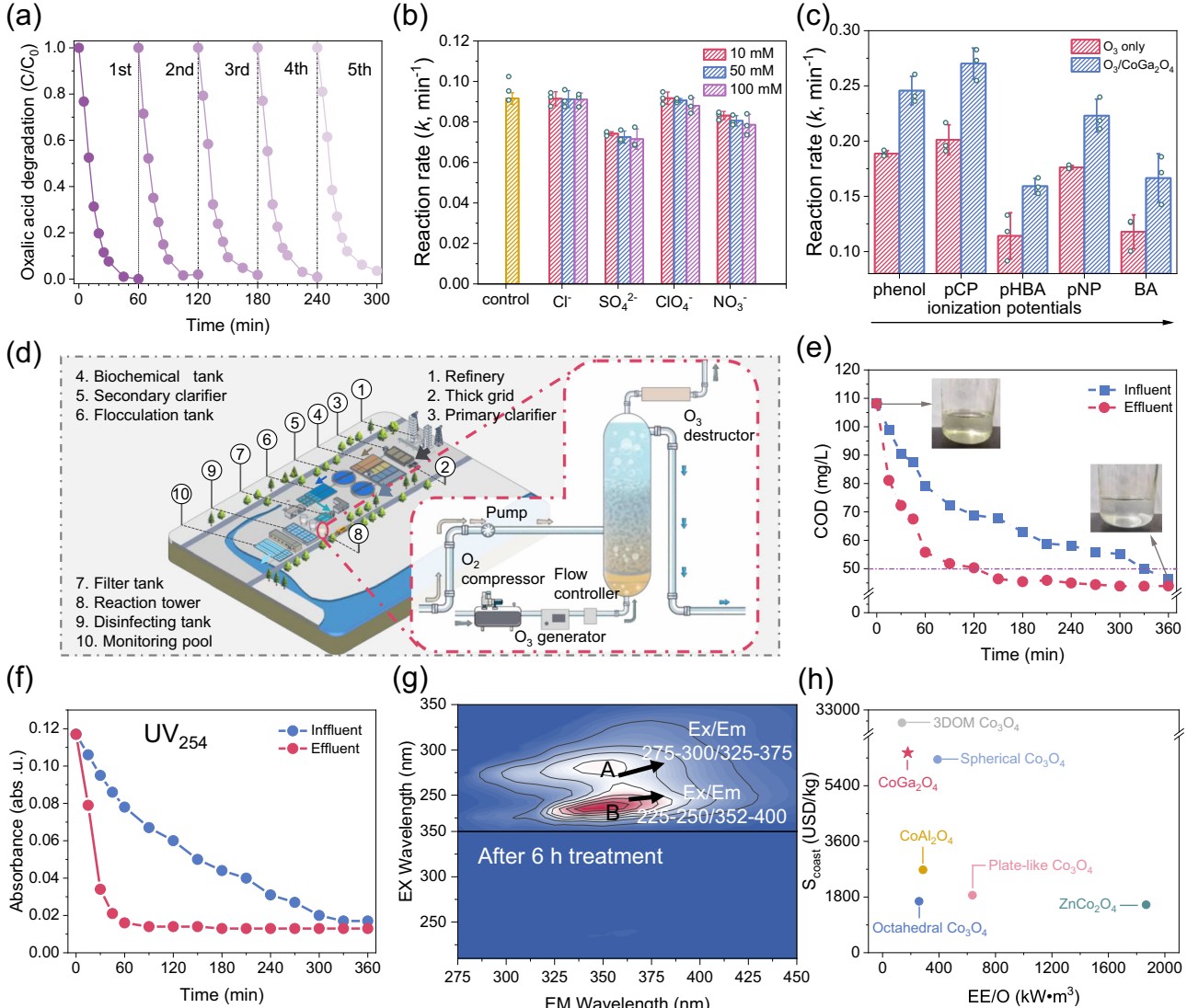

**Fig. 5 | Practical application potentials. a** Cyclic experiments for the CoGa$_2$O$_4$/O$_3$ system. **b** Influences of inorganic anions on the catalytic performance for the CoGa$_2$O$_4$/O$_3$ system. **c** Comparisons of reaction rate constants for different phenolics. Pollutants include phenol, para-chlorophenol (pCP), para-nitrophenol (pNP), para-hydroxyl benzoic acid (pHBA), benzoic acid (BA). **d** Schematic diagram of the proposed catalytic ozonation system using the continuous-flow reactor. O$_3$ generated by the pressurized O$_2$ was introduced into the fixed-bed reactor (with catalyst immobilized on the polyurethane foam) from the bottom diffusor with its flow rate regulated by a mass-flow rate controller. Simultaneously, the wastewater to be treated was fed above the O$_3$ diffuser. After the reaction, the remaining O$_3$ gas exiting from the top of the reaction column went through an ozone decomposer

before being discharged, while the treated reaction solution was collected directly. **e** Chemical oxygen demand (COD) removal profiles for treatment of real refinery wastewater in the continuous-flow reactor. **f** UV$_{254}$ variation profiles for the treatment of real refinery wastewater. **g** 3D-Excitation-emission-matrix (EEM) spectra raw and treated wastewater. Ex excitation wavelength, Em emission wavelength. Reaction condition: ozone flow rate 100 mL min$^{-1}$, ozone concentration: 25 mg L$^{-1}$, catalyst loading: 1 g L$^{-1}$, wastewater flow rate: 48 mL min$^{-1}$. **h** Comparison of electrical energy per order (EE/O) and synthesis costs for different catalytic ozonation systems. Error bars represent the standard deviations from three replicate measurements. Source data are provided as a Source Data file.

## Discussion

In summary, a geometrical-site-dependent activity was discovered in catalytic ozone activation on Co$_3$O$_4$ and its Co$^{2+}_{Td}$ and Co$^{3+}_{Oh}$ substitutes at the [111] crystal facet. CoGa$_2$O$_4$ with Co$^{2+}_{Td}$-dominated sites demonstrated a 17.6-fold higher TOF than that of ZnCo$_2$O$_4$ with high exposure of Co$^{3+}_{Oh}$ sites, which can be ascribed to the improved O$_3$ utilization efficiency and the enhanced ·OH generation selectivity by avoiding the formation of unfavorable $^1$O$_2$ accounted on Co$^{2+}_{Td}$ sites. Theoretical calculations suggested that the highly spin-polarized Co$^{2+}_{Td}$ sites fine-tuned the position of the MO*s-Co$^{2+}_{Td}$, formed by spin-down channels of Co $3d$ orbital, close to the Fermi level, decreasing the energy barriers for O$_3$ dissociation and accelerating the

electronic communication with the O $2p$ orbitals of the adsorbed O$_3$ molecules. The strong hybridization strength with the promoted electron transfer between Co$^{2+}_{Td}$ sites and O$_3$ molecules induced inner-sphere complexation and favored the formation of surface-bound ·OH. As a result, the CoGa$_2$O$_4$/O$_3$ system exhibited high adaptability towards inorganic anions and demonstrated high treatment efficiency for real petrochemical wastewater. Tailoring the spin-polarized electronic states from geometrical-site engineering offers intrinsic insights into regulating the reaction thermodynamics in TM oxide-induced catalysis. Moreover, the modulated ROS production by different geometrical sites envisaged the practical potential of spinel oxides in real wastewater treatment.

## Methods

### Chemicals

Sulfuric acid ($H_2SO_4$, 98%), cobalt chloride hexahydrate ($CoCl_2\bullet 6H_2O$, 99.99%), zinc chloride anhydrous ($ZnCl_2$, 99.95%), aluminum chloride hexahydrate ($AlCl_3\bullet 6H_2O$), magnesium chloride ($MgCl_2$, 99.95%), gallium chloride ($GaCl_3$, 99.99%), citric acid (CA, 99.5%), ethanol (99%), methanol (MeOH, 99%), para-chlorophenol (pCP, 99.5%), para-nitrophenol (pNP, 99.9%), phenol (99.9%), para-hydroxyl benzoic acid (pHBA, 99.5%), benzoic acid (BA, 99.9%), DMSO (99%), methylene blue (MB, 96%), rhodamine B (95%), OA (99.8%), sodium hydroxide (NaOH, 99%), and sodium sulfate ($Na_2SO_4$, 99%) were obtained from Aladdin Reagents. Horseradish peroxidase (95%) was procured from Sigma-Aldrich. 2,2,6,6-tetramethyl-4-piperidinol (TEMP, 99%) and 5,5-dimethyl-1-pyrroline (DMPO, 99%) were purchased from Dojindo Laboratories. SOSG was purchased from Invitrogen by Thermo Fisher Scientific. Ultra-pure water (18.2 MΩ•cm) was used in all the experiments. All the chemicals in this work were used for further purification.

### Synthesis of geometrical-substituted cobalt-based catalysts

A sol-gel method was used to synthesize cobalt-based spinel oxides[51]. For synthesis of $CoGa_2O_4$, $ZnCo_2O_4$, and $MgCo_2O_4$, stoichiometric amounts of $CoCl_2\bullet 6H_2O$, $GaCl_3$, $ZnCl_2$, and $MgCl_2$ (Co/Ga = 1/2 for $CoGa_2O_4$, Co/Zn = 2/1 for $ZnCo_2O_4$, and Co/Mg = 2/1 for $MgCo_2O_4$) were mixed in a citric acid solution (18 mL, 1.5 M) with a molar ratio of 1:1.5 under magnetic stirring at room temperature to form a stable homogeneous solution. The resulting mixed solution was vigorously stirred while heated in a water bath at 80 °C until a highly viscous gel was formed. The gel was subsequently dried overnight at 120 °C in an oven for foaming. The dried mixture was then milled to a powder and calcined at 700 °C in a muffle furnace for 3 h with a heat ramp of 5 °C/min. Pristine $Co_3O_4$ without geometrical-site substitution was prepared using the same synthesis protocol with $CoCl_2\bullet 6H_2O$ as the precursor. For $CoAl_2O_4$ synthesis, a similar synthesis protocol was adopted with the $CoCl_2\bullet 6H_2O$ and $AlCl_3\bullet 6H_2O$ as the metal precursors ($CoCl_2\bullet 6H_2O$/$AlCl_3\bullet 6H_2O$ molar ratio of 1/2), except for the final calcination temperature was changed to 1000 °C.

### Characterization

XRD patterns of the as-synthesized materials were recorded to investigate their crystallinity and phases (X'Pert-PROMPD, PAN analytical B.V. with Cu Kα radiation). Morphologies of the samples were observed using a ZEISS Gemini SEM 300 with an accelerating voltage of 5 kV. Moreover, transmission electron microscopy (TEM) images, high-angle annular dark field scanning TEM (HAADF-STEM) images, and corresponding energy-dispersive X-ray spectroscopy (EDX) elemental mappings were obtained from a Talos F200S TEM/STEM equipped with an energy-dispersive spectrometer (EDS, SUPERX). The X-ray absorption structure (XAS) spectra, including XANES and extended X-ray absorption fine structure (EXAFS) for the as-synthesized cobalt-based spinels at Co K-edge and Ga K-edge were collected at the Beamline 21A in National Synchrotron Radiation Research Center, Taiwan, China. The surface chemistry of the materials was investigated using the X-ray photoelectron spectroscopy (XPS) under ultra-high vacuum conditions with Al-Kα X-ray irradiation (Thermo Fisher Scientific ESCALAB 250Xi). $N_2$ sorption isotherms were measured using a Micrometrics Tristar 3000 to determine the SSAs and pore size distribution. Raman spectra were recorded using an ISA argon-laser Raman spectrometer (LabRAM HR Evolution) with a wavelength of 532 nm. Temperature-dependent magnetizations were conducted under a magnetic field (H = 1 kOe) by the following field-cooling procedures (400–5 K) by a vibrating sample magnetometer (Lakeshore 7404). The point of zero charge ($pH_{pzc}$) was measured by the chemical drift method[52]. Probe-based in situ electron paramagnetic resonance (EPR) tests for ROS detection were conducted by a Bruker EMX-PLUS EPR spectrometer. An additional detailed description of characterization methods is provided in Supplementary Note 1.

### Ozone activation and catalytic ozonation

Catalytic ozonation degradation of aqueous organics was performed in a semi-batch glass reactor at 25 °C. High-purity oxygen (99.999%) was a flow rate of 100 mL min$^{-1}$ was used for ozone generation (Anseros, Germany), and the concentration of the generated $O_3$ was monitored by an online ozone detector (Anseros Ozomat GM, Germany). The generated ozone was introduced into the reaction solution through the glass diffusor at the bottom of the semi-batch reactor. Unless specified, the reaction solution was prepared by adding 0.05 g of catalyst to a 500 mL solution containing 50 mg L$^{-1}$ of OA. The initial pH of reaction solution was adjusted to 3 by 0.01 M $H_2SO_4$/NaOH in OA solution. At certain time intervals, samples were withdrawn from the reaction solution, filtered by a 0.22 μm PES filter, and injected into sample bottles for the subsequent ultra-high-performance liquid chromatography (UPLC, Thermo Fisher U3000) and mineralization of TOC (total organic carbon, Shimadzu TOC-CPH) analysis. In the stability evaluation of the catalyst after multiple runs, the catalyst was recovered by vacuum filtration after each cycle, washed three times with ultra-pure water, and dried at 60 °C in the oven for the next use. The dissolved $O_3$ in the reaction solution was measured through the Indigo blue method[53]. $H_2O_2$ in solution was examined by the horseradish peroxidase method[54]. A detailed description of the analysis methods is provided in Supplementary Note 2.

### Ozone utilization efficiency calculations

The ozone utilization efficiency can be calculated by the following equation:

$$OUE = \frac{[O_3]_{used}}{[O_3]_{input}} = \frac{\int_0^t [O_3]_{in} dt - \int_0^t [O_3]_{offgas} dt - \int_0^t [O_3]_{solution} dt}{\int_0^t [O_3]_{in} dt} \quad (1)$$

where, $[O_3]_{used}$ is the total ozone utilization, $[O_3]_{input}$ is the input ozone concentration, $[O_3]_{solution}$ is the dissolved ozone concentration, $[O_3]_{offgas}$ is the exhaust gas concentration after the ozone reaction.

The $R_O$ factor (molar ratios of the degraded OA to the consumed $O_3$) can be calculated by the following equations.

$$n[OA]_{degraded} = n[OA]_0 - n[OA]_t \quad (2)$$

$$R_O = \frac{n[OA]_{degraded}}{n[O_3]_{used}} \quad (3)$$

where $n[OA]_{degraded}$ is the molar amount oxalic acid degraded, $n[OA]_0$ is the oxalic acid concentration at the initial moment, $n[OA]_t$ is the residual oxalic acid concentration after the reaction, and $n[O_3]_{consumed}$ represents the molar amount of consumed $O_3$.

### Calculation of $S_{BET}$ normalized activity efficiency

$S_{BET}$ normalized activity efficiency (AE), which can be derived from the following equations, was used to compare the catalytic performances of the as-synthesized Co-based spinels with the reported catalysts in literature.

$$AE = \frac{Reaction\ rate}{catalyst\ loading} \quad (4)$$

$$Normalized\ AE = \frac{AE}{S_{BET}} \quad (5)$$

## In situ Raman experiments

In situ Raman spectra were collected by a laser confocal Raman spectrometer (LabRAM HR Evolution HORIBA, Japan) equipped with a 532 nm laser. Typically, 10 mg of the sample was placed in an in situ reaction cell. Saturated $O_3$ solution with pH 3 was continuously pumped into the reaction cell with a flow rate of 15 mL L$^{-1}$ to ensure the catalyst was uniformly permeated.

## Calculation of $^•$OH concentration

The concentration of the produced $^•$OH during the reaction was quantified using OA as the molecular probe due to its sluggish reaction rate with the produced $^1O_2$ and ROS with low oxidation potentials.

$$-\frac{d[OA]}{dt} = k_{O_3}[OA][O_3] + k_{^•OH}[OA][^•OH] \tag{6}$$

$$\ln\left(\frac{[OA]}{[OA]_0}\right) = -(k_{O_3}\int_0^t [O_3]_{sol}dt + k_{^•OH}\int_0^t [^•OH]dt) \tag{7}$$

$$\int_0^t [^•OH]dt = -(\ln\left(\frac{[OA]}{[OA]_0}\right) + k_{O_3}\int_0^t [O_3]_{sol}dt)/k_{^•OH} \tag{8}$$

Where $k_{O_3}$ is the rate constant for OA with ozone, $k_{^•OH}$ is rate constant for OA with $^•$OH radicals, $[O_3]_{sol}$ is the dissolved $O_3$ concentration in the reaction solution, and $[^•OH]$ is the concentration of $^•$OH.

Correspondingly, the $^•$OH/$O_3$ conversion rate ($R_{^•OH}$) can be calculated by the following equation.

$$R = \frac{\int_0^t [^•OH]dt}{\int_0^t [O_3]_{sol}dt} \tag{9}$$

## In situ electrochemical measurements

In situ electrochemical measurements were conducted on an electrochemical workstation (CHI 760D, CH Instrument) with a graphite electrode, a saturated calomel electrode, and a glassy carbon electrode (GCE) as the counter electrode, reference electrode, and working electrode, respectively. The electrolyte was prepared by 0.1 M $Na_2SO_4$ solution. $H_2SO_4$ and NaOH solutions (both 0.1 M) were employed to adjust the electrolyte pH to 3. For the preparation of the working electrode, 20 mg of the catalysts were dispersed in 1 mL of isopropanol containing 20 μL of Nafion® solution. The mixed solution was ultrasonicated for 1 h to form a homogeneous ink. The GCE surface was coated by 10 μL ink drops, which were then air-dried and subsequently heated in an oven at 80 °C for 5 min. The working electrode was stabilized in the electrolyte overnight to ensure a stable potential prior to each electrochemical test. In situ OCP tests were performed after stabilization of spinel-catalysts-GCE in the electrolyte. $O_3$ was introduced to the reactor with a flow rate of 30 mL min$^{-1}$ and a concentration of 25 mg L$^{-1}$ and the change in potential was recorded. Electrochemical impedance spectroscopy (EIS) is measured in the frequency range of 0.01 Hz to 100 kHz with an amplitude of 10 mV at the open-circuit potential (OCP) of each material.

## Density functional theory calculations

DFT calculations were performed using the Vienna ab initio simulation package, version 5.4.1[55–57]. The exchange-correlation energy was calculated using the Perdew–Burke–Erzenhof (PBE) functional[58]. The Hubbard $U$ parameter (GGA +$U$) with $U$ = 4.0 eV was adopted to accurately calculate the electron correlation within the d states of cobalt ions. The total energy change during the structure optimization process ultimately converged to $5 \times 10^{-6}$ eV. In addition, the forces per atom were reduced to 0.01 eV Å$^{-1}$. The k-point mesh of the Brillouin zone was set at $2 \times 2 \times 1$ for geometry optimization. A cutoff energy of 450 eV was employed during the computations. The detailed information of the constructed DFT models is presented in Supplementary Table 14. In the calculations, the bottom atom layer was remained fixed, while the other atoms were allowed to relax[59,60]. The climbing image nudged elastic band (CI-NEB) method was used to determine the transition state of the reaction. The charge distribution within a specific structure was determined through static self-consistent calculations. Based on these results, local atomic charges were computed using the Bader charge analysis method. The crystal orbital Hamilton populations (COHPs) of spinel oxides were calculated using the computer program Local Orbital Basis Suite Toward Electronic-Structure Reconstruction (LOBSTER)[61]. And the computed energy releases in different transition states are provided in Supplementary Table 15.

The formation energy of the adsorbate is calculated using the following formula:

$$\Delta E_{ads} = E_{slab-adsorbate} - E_{slab} - E_{adsorbate} \tag{10}$$

where the $E_{surface-adsorbate}$, $E_{slab}$, and $E_{adsorbate}$ are the total energies of the adsorbed state, clean surface, and adsorbate, respectively.

## Data availability

The data supporting the findings of this study are included within the main text and the Supplementary Information file. Source data are provided with this paper as a Source Data file. All the raw data relevant to the study are available from the corresponding author upon request. Source data are provided with this paper.

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

## Acknowledgements
Y.W. and C.C. greatly appreciate the financial supports from the National Natural Science Foundation of China (No. 22478426 and 22278436). Y.W. acknowledges the Young Elite Scientists Sponsorship Program by BAST (No. 1101020370359).

## Author contributions
S.L., Y.W., S.W., and H.S. conceived the idea and designed the experiments. S.L., Y.L., P.C., and T.K. conducted the research. C.C. and X.D. contributed new reagents/analytic tools. S.L., Y.W., and H.S. wrote the paper. All authors contributed to the discussion of the manuscript.

## Competing interests
The authors declare no competing interests.
