## [Transparent Peer Review file · Nature Communications]

Tailored ozone activation on geometrical-site-dependent cobalt with selective coordination

Corresponding Author: Professor Hongqi Sun

Version 0:

Reviewer comments:

Reviewer #1

(Remarks to the Author)

This manuscript synthesized octahedral Co_3O_4 and doped Ga or Zn to prepare tetrahedral Co or octahedral Co sites only. The authors used these materials for oxalic acid decomposition using O_3 oxidant. Although authors provided massive characterization data, basically CoGa_2O_4 presented slightly better activity than Co_3O_4 (Fig 2a and Fig S18a). Considering that their BET surface areas are similar as $0.3 \text{ m}^2/\text{g}$, I doubt if the difference in activity actually results from Co coordination difference. If Co_3O_4 with larger surface area, which can be synthesized easily, is tested for OA decomposition, better activity would be easily obtained. I can't recommend this work for Nat Commun.

Reviewer #2

(Remarks to the Author)

The manuscript titled "Tailored ozone activation on geometrical-site-dependent cobalt with selective coordination" presents a compelling study on the role of geometrical site-specific cobalt activation in catalytic ozone decomposition. The authors have provided a thorough investigation using a combination of experimental characterization, catalytic performance tests, and theoretical calculations. The findings, particularly concerning the site-specific activation of Co^{2+} and Co^{3+} sites, offer significant insights into the catalytic properties of spinel oxides, which could have broad implications in the field of heterogeneous catalysis. The manuscript is generally well-written and structured, with clear presentations of data and logical flow in the argumentation. However, several aspects of the study need further clarification or additional data to fully substantiate the claims made. Below, I have outlined specific questions and suggestions for the authors to address.

1. You mention changes in surface chemical properties as a factor influencing catalytic activity. Could you provide more detailed data, especially focusing on the chemical environment of different geometrical sites (Co^{2+} and Co^{3+}) and how these might change during the catalytic process?
2. In the DFT calculations, you discuss the band gap and density of states. Can these be compared with experimental techniques such as UV-Vis spectroscopy or other electronic structure characterization methods? How do you ensure the computational models accurately reflect the experimental samples?
3. You use EPR and fluorescence probes to detect ROS. Can you provide more details on how the selectivity and sensitivity of these detection methods were validated? How confident are you in the identification of specific ROS like $\bullet\text{OH}$ and 1O_2 ?
4. The manuscript reports the reusability of the CoGa_2O_4 catalyst. Could you provide more data on the long-term stability of the catalyst, especially after multiple catalytic cycles? What degradation mechanisms, if any, were observed?
5. The manuscript discusses the selective production of $\bullet\text{OH}$ over 1O_2 at different cobalt sites. Can you provide quantitative data or modeling results to support the observed selectivity? How do you propose to control this selectivity in practical applications?
6. How does varying the catalyst loading affect the observed catalytic activity and ROS production? Have you tested different loadings to optimize the catalytic performance, and if so, what trends were observed?
7. How does the performance of your catalysts compare with other established catalysts for ozone activation? Can you provide a more detailed comparison, possibly in a tabular form, that includes metrics like turnover frequency, O_3 utilization efficiency, and ROS production rates?

Reviewer #3

(Remarks to the Author)

The manuscript by Wang et. al., prepared Co-based spinel oxides with specific Co sites (i.e., Co₂+Td, and Co₃+Oh) to investigate the geometrical-site dependent ozone activation performance. By combing experimental results and DFT calculations, they found that highly spin-polarized Co₂+Td sites were the active sites for selectively producing surface-bound hydroxyl radicals (\cdot OH) of Co-based spinel oxides. However, Co-based oxides have been widely used in oxygen-related catalysis, and the geometrical-site dependent relationship have been widely investigated. Moreover, the authors used the same images for different samples, for example CoGa₂O₄ in Figure 1e is the same as Figure S3a for Co₃O₄. I think the authors used the same images to represent different samples by artificially changing the magnification of the image. Therefore, I suggest to reject this manuscript. Some specific comments are listed as follows:

1. In Figure 1e, why is the magnification of different samples different? Please use images with the same magnification for different samples.
2. From Figure 1e, the size of synthesized Co-based spinel oxides is quite uneven. How did the authors ensure the exposed area of (111) plane is the same. If the area is different, the different performance may be attributed to the different number of active sites.
3. The morphological description of the synthesized oxides "In the high-resolution transmission electron microscopy (HRTEM) images of the pristine Co₃O₄ and its substitutes, the lattice spacing of 0.47 nm was observed at the edges, attributing to the (111) facet (Fig. 1f-1g). Additionally, a fixed angle of 60° was observed between the (220), (422), and (202) planes in the corresponding selected area electron diffraction (SAED) images (Supplementary Fig. 8), confirming that the (111) facet was the common exposed facet on their octahedrons", indicating that Co₃O₄, ZnCo₂O₄ and CoGa₂O₄ nanoparticles have similar polyhedron-like morphologies but inevitable diameter changes resulting from the different ion sizes and electronic states of introduced additional elements (i.e., Zn and Ga) with respect to Co." to be not very fortunate. If the mechanisms of the catalytic processes have to be modeled in detail, the surface structure of the main exposed surface should be identified. The determination of the main catalytic crystal surface (111) is not rigorous enough.
4. As mentioned by the authors, replacing Co₂+ by Zn₂+ and Co₃+ by Ga₃+ with larger ionic radii (0.58 vs. 0.60 Å, 0.61 vs. 0.62 Å) gave rise to the increase of interplanar crystal spacing. This phenomenon should be concerned and explained, which may be the key reasons for the different catalytic performances of the three samples.
5. Does the directional substitution of Zn₂+ and Ga₃+ have an impact on the local electronic structure of Co₃+/Co₂+ in other positions? Whether this effect plays a role in catalysis.
6. It is unfavorable to conclude that the Co₃O₄, ZnCo₂O₄ and CoGa₂O₄ displayed different geometric locations of Co sites just based on the XRD and HAADF-STEM image. The authors should provide more atomic structure information to identify the location of Zn and Ga species.
7. In the theoretical calculations section, the authors use density flooding theory for their calculations. The transition metals are a strongly correlated system of d electrons, so the interaction between electrons needs to be considered, and whether the absence of this critical factor has an impact on the catalytic performance and electronic structure, the authors apparently lack this critical evidence.
8. The computational models should be described in detail: lattice parameters, periodic unit size, layers of material, vacuum slab size, modeled crystallographic plane, geometric detail of the complex Co₂+Td/Co₃+Oh, total charge....
9. All calculated energies displayed in figures should be tabulated.
10. Please check the "References" format carefully, there are a few minor errors, such as: Reference 10, 14, 20, 44.

Version 1:

Reviewer comments:

Reviewer #2

(Remarks to the Author)

The author has made the necessary revisions according to my suggestions and has successfully addressed all the concerns I raised. The revised version significantly improves the quality of the paper, and I believe all issues have been properly resolved. Therefore, I support the publication of this paper.

Reviewer #3

(Remarks to the Author)

The manuscript has been revised according to the comments and all my concerns have been suitably addressed. I believe this manuscript now is ready for publication.

Reviewer #4

(Remarks to the Author)

The authors fabricated a series of Co-based spinel oxides with controlled crystal facets/sizes to investigate their geometrical site-dependent catalytic activity for ozone activation. The highly spin-polarized Co₂+ at tetrahedral sites promoted the formation of surface-bound hydroxyl radicals, resulting in enhanced water decontamination efficiency. However, the concerns raised by Reviewer #1 have not been fully addressed in the revised manuscript.

As pointed out by Reviewer #1, the 3DOM-Co₃O₄, with a larger specific surface area than CoGa₂O₄, exhibits higher catalytic activity for ozone activation (Figure R2). Thus, it is still unclear whether the activity difference stems from the coordination geometry of the Co sites. In addition, the Co²⁺ and Co³⁺ species in Co-based spinel structures are usually found in both tetrahedral and octahedral sites. The Co²⁺ species with stronger electron-donating ability than Co³⁺, is well-known to facilitate ·OH generation at both Td and Oh sites, making the claim of negligible Co valence effects between Co²⁺_Td and Co³⁺_Oh inaccurate. Furthermore, while the authors suggest previous research overlooked the uniformity of exposed crystal planes, the high activity of CoGa₂O₄ may also stem from the exposure of [111] crystal facets. Comparative materials with different crystal planes are lacking due to limitations in the synthesis method. Therefore, I do not recommend this work for publication in Nature Communications.

Reviewer #5

(Remarks to the Author)

Version 2:

Reviewer comments:

Reviewer #4

(Remarks to the Author)

The authors have provided sufficient new evidences and literature survey to support the conclusions. My concerns have been addressed and I think now it can be considered for publication.

Reviewer #5

(Remarks to the Author)

Responses to Reviewers' Comments

Manuscript No.: NCOMMS-24-42294

Title: Tailored ozone activation on geometrical-site-dependent cobalt with selective coordination

Reviewer #1:

This manuscript synthesized octahedral Co_3O_4 and doped Ga or Zn to prepare tetrahedral Co or octahedral Co sites only. The authors used these materials for oxalic acid decomposition using O_3 oxidant. Although authors provided massive characterization data, basically CoGa_2O_4 presented slightly better activity than Co_3O_4 (Fig. 2a and Fig. S18a). Considering that their BET surface areas are similar as $0.3 \text{ m}^2/\text{g}$, I doubt if the difference in activity actually results from Co coordination difference. If Co_3O_4 with larger surface area, which can be synthesized easily, is tested for OA decomposition, better activity would be easily obtained. I can't recommend this work for Nat Commun.

Response: Thank you very much for your thoughtful comments. We greatly appreciate your perspective and the important points you have raised. The scope of our study extends beyond simply enhancing the catalytic activity of Co_3O_4 for oxalic acid decomposition using O_3 as an oxidant. Our aim is to uncover the intrinsic active sites in cobalt-based spinel oxides and to explore the spin-polarization effects stemming from geometrical site competition on reaction thermodynamics in transition metal oxide-catalyzed processes. This work highlights the high selectivity of these oxides in producing reactive oxygen species. Moreover, our findings help resolve the conflicting conclusions in previous studies, which were often influenced by uncontrolled crystal facets/sizes and overlooked spin-polarization effects.

Specifically, (i) we fabricated Co_3O_4 and its four types of geometrical substitutes (CoGa_2O_4 and

CoAl₂O₄ for octahedral Co sites substitutions, ZnCo₂O₄, and MgCo₂O₄ for tetrahedral Co sites substitutions) by replacing the undesired sites with catalytically inactive cations in O₃ activation (i.e. Ga, Al, Zn, and Mg). Rigorous examinations revealed that CoGa₂O₄ and ZnCo₂O₄ exhibited the same exposed crystal facet of [111] and similar crystal sizes to those of Co₃O₄ and thus they were selected as the research objects. CoAl₂O₄ and MgCo₂O₄ with distinct morphologies and crystal sizes, possibly arising from the large ionic radius deviations between Co and the substituted cations, were ruled out from further investigations. (ii) For these selected catalysts, a geometrical-site-dependent activity was observed, and Co²⁺_{Td} sites (CoGa₂O₄) demonstrated a significantly higher activity than ZnCo₂O₄ with Co³⁺_{Oh} sites (17.6-fold higher in TOF). (iii) The highly spin-polarized Co²⁺_{Td} sites fine-tuned the reaction thermodynamics by invoking strong orbital interactions with O₃. The decreased energy barriers for O₃ dissociation and the accelerated electronic communications promoted the formation of surface-bonded *OH while preventing the formation of unfavorable singlet oxygen (¹O₂), a competitive side product for O₃ catalytic dissociation. (iv) The abundant surface-bonded *OH on Co²⁺_{Td} sites, via inner-sphere strong complexation, exhibited a high adaptability towards inorganic anions (which are effective quenchers for free radicals) and demonstrated a high treatment efficiency for real petrochemical wastewater.

We believe that the elucidation of intrinsic insights into the geometrical site engineering offers regulating protocols for the reaction thermodynamics in transition metal oxide-induced catalysis and envisages practical application potentials of engineered spinel oxides.

Regarding the concerns on the relationship between surface area and the activity of the Co₃O₄, we employed a variety of synthesis protocols to further synthesize Co₃O₄ with different morphologies (i.e. 3DOM-Co₃O₄, plate-like Co₃O₄, and spherical Co₃O₄, Fig. R1). The synthesis details are as follows.

Figure R1. Scanning electron microscopy (SEM) images of 3DOM Co₃O₄ (a), plate Co₃O₄ (b), and spherical Co₃O₄ (c).

Synthesis of 3DOM-Co₃O₄

Three-dimensionally ordered poly methyl methacrylate (PMMA) microspheres with an average diameter of 350 nm were firstly synthesized according to a previously reported method (Chem. Commun. 2008, 6552-6554, ACS Catal. 2019, 9, 7548-7567). First, 120 mL of deionized water was added to a 500 mL four-neck flask, followed by the introduction of argon gas (0.5 L/h) to purge the air inside. The flask was then heated in an oil bath for 30 min to remove any remaining air. Next, 12 mL of methyl methacrylate was added to 120 mL deionized water, and the solution was maintained at 70 °C with stirring at 350 rpm to stabilize the temperature. Subsequently, potassium persulfate (30 mL, 0.12 g preheating to 70 °C) was added, and after 2 h reaction, a milky white liquid was achieved and transferred to a centrifuge. The centrifugation was conducted at 4000 rpm for 10 h to achieve ordered packing of the microspheres. After centrifugation, the supernatant was decanted, and the solid was dried in an oven at 40 °C for 24 h. The yielded well-ordered PMMA microspheres with a diameter of approximately 350 nm. For synthesis of 3DOM-Co₃O₄, 30 mmol of Co(NO₃)₂·6H₂O and 30 mmol citric acid were completely dissolved in a solution containing 10 mL deionized water and 10 mL methanol under stirring, and then 1.5 g of the as-synthesized ordered PMMA microspheres were soaked into the above solution and kept static for 24 h. After filtration, the solid powders were dried at

50 °C for 24 h, then calcined at 500 °C for 4 h under the air atmosphere. And 3DOM-Co₃O₄ with an average macroporous size of 290 nm was obtained. The yield of 3DOM-Co₃O₄ is around 1.2%.

Synthesis of plate Co₃O₄

In a typical synthesis, 30 mmol of Co(NO₃)₂·6H₂O and 30 mmol citric acid were dissolved in a solution containing 10 mL DI water and 10 mL methanol under stirring, and then the solution was dried at 50 °C for 120 h. After the drying process, the mixed precursors were grinded into fine powders, followed by calcination at 500°C for 4 h under the air atmosphere in a muffle furnace with a heating rate of 5 °C/min. The yield of plate Co₃O₄ is around 8%.

Synthesis of spherical Co₃O₄

To prepare the spherical Co₃O₄, 0.4 mol of Co(NO₃)₂·6H₂O and 0.03 mol of NaOH and were dissolved in 40 mL of DI water with vigorous stirring. After 1 h, the solution was moved into a Teflon-lined autoclave and heated at 180 °C for 5 h. The obtained solution was filtered with DI water and ethyl alcohol and dried overnight at 50 °C. Subsequently, the dried powders were calcined at 500 °C for 3 h in the air atmosphere with a heating rate of 5 °C/min. The yield of spherical Co₃O₄ is around 1.1%.

Figure R2. (a) N_2 sorption isotherms of synthesized Co_3O_4 . (b) Catalytic ozonation for OA degradation with synthesized Co_3O_4 . (c) Reaction rates and BET surface area normalized reaction rates of $\text{Co}_3\text{O}_4/\text{O}_3$ systems. (d) Cyclic experiments for synthesized Co_3O_4 and cobalt leaching concentration. [catalyst] = 0.1 g L^{-1} ; temperature: $25 \text{ }^\circ\text{C}$; initial pH = 3, $[\text{OA}]_0$: 50 mg L^{-1} .

According to the N_2 -sorption isotherms (Fig. R2a), the SSAs of 3DOM- Co_3O_4 , plate Co_3O_4 , and spherical Co_3O_4 are 29.1 , 17.5 , and $78.4 \text{ m}^2/\text{g}$, respectively, significantly exceeding those of the cobalt-based spinel oxides synthesized via the facile sol-gel method in the manuscript. Activity tests revealed that both plate-like Co_3O_4 and spherical Co_3O_4 displayed lower activity than CoGa_2O_4 . 3DOM- Co_3O_4 obtained a similar activity to CoGa_2O_4 , yet the synthesis procedure for 3DOM- Co_3O_4 is considerably more complicated than that of CoGa_2O_4 (Fig. R2b). Furthermore, the intrinsic activity of Co sites in these catalysts were determined by normalizing the kinetic rates by their SSAs. It is found that the Co

sites in CoGa_2O_4 obtained a 59~610-fold higher intrinsic activity than those of the Co_3O_4 with large SSAs (Fig. R2c).

We further evaluated the recyclability of these synthesized Co_3O_4 with large SSAs by a multi-cycle reusability test. Clearly, the newly synthesized Co_3O_4 with high SSAs demonstrated much inferior recyclability compared to CoGa_2O_4 (Fig. R2d). Significant passivation was observed for these Co_3O_4 with large SSAs after the 3rd cycle. Furthermore, high cobalt leaching was observed for both plate Co_3O_4 and spherical Co_3O_4 , as indicated by inductively coupled plasma mass spectrometry (ICP-MS) results (0.54 and 0.66 mg/L), which exceed the national wastewater discharge standard of China (0.5 mg/L) and will lead to secondary pollution. In contrast, the strong bonding in CoGa_2O_4 crystal structure and the fast reaction kinetics at the catalyst surface guarantee the structural integrity of CoGa_2O_4 during the multi-cycle reusability test. As a result, only minor passivation was observed after 4 cycles with a negligible amount of leached Co^{2+} (0.09 mg/L).

Figure R3. Comparison of catalyst fabrication costs (a) and EE/O for different reaction systems (b).

Additionally, we conducted economic analysis to the Co-contained spinels synthesized in this study via the sol-gel method and Co_3O_4 with high SSAs to evaluate the economic feasibility of enhancing

intrinsic activity by regulating the coordination environment around $\text{Co}^{2+}_{\text{Td}}$ sites. The results show that the Co-contained spinels synthesized in this study obtained lower synthesis costs than Co_3O_4 with high SSAs, despite Ga's significantly higher price compared to other metals (Fig. R3a). Moreover, the electrical energy per order (EE/O), measuring the electrical energy consumption, for the Co-contained spinels synthesized in this study are remarkably lower than those of Co_3O_4 with high SSAs (Fig. R3b), suggesting the economic feasibility of the activity enhancement via optimizing the coordination environments around the active sites. For future practical applications, we can replace Ga by other cost-efficient metals such as Al to reduce the catalysts fabrication expenditure while maintaining the high activity. Moreover, further studies can focus on regulating the ambient microenvironments around the active $\text{Co}^{2+}_{\text{Td}}$ sites by facile treatments, such as defect engineering and heteroatom doping, to enhance the activity by inducing more intense electronic communications and stronger complexation with the O_3 molecules and triggering higher production of surface-bound $\cdot\text{OH}$ that resist inorganic anions in real wastewater.

Moreover, our literature survey indicates that simply increasing the SSA of Co_3O_4 does not significantly improve its catalytic activity beyond a certain threshold. Although increasing the SSA facilitates the exposure of the active sites, the thermodynamic-governed intrinsic reactivity of these sites remains unchanged. Furthermore, the enhanced exposure of the active sites may also induce the occurrence of side reactions. We plot the relationships between SSAs and activity based on our experimental results and the reported literature (Fig. R4). In this plot, SSAs of Co_3O_4 are set as the X coordinate, while the target reactant conversion rates per unit time and dosage are served as the Y coordinate. Some Co_3O_4 are not catalytically active even with high SSAs and the activity of the as-synthesized catalysts in this study, despite with small SSAs, is among the highest reported. This

suggests that regulating strategies for the active sites that determine the intrinsic activity of Co_3O_4 are critical in different reaction systems rather than their SSAs.

Figure R4. Relationship between the SSAs of the Co_3O_4 reported in the literature and their activity efficiency. Noted that the activity efficiency is derived by the follow equation.

$$AE = \frac{\text{Pollutant(g)} \times \text{conversion rate}}{\text{catalyst loading (g)} \times \text{time (min)}}$$

Table R1. Comparison of the key reaction parameters and conversion rates of Co_3O_4 -based catalysts in previously reported literature and this study.

Catalysts	Loading	Synthesis method	Pollutant	SSA	Removal rate	Ref.
Co_3O_4 -1	0.1 g L ⁻¹	Co-precipitation	Norfloracin	100.2 m ² g ⁻¹	~26% TOC removal (60 min)	(J. Hazard. Mater. 2021, 403, 123697) (Chem. Eng. J 2019, 373, 1329-1337)
Co_3O_4 -2	0.05 g L ⁻¹	Hydrothermal	Paracetamol	121 m ² g ⁻¹	~33% removal (30 min)	(Adv. Funct. Mater. 2021, 31, 2009000)
Co_3O_4 -3	0.02 g	Hydrothermal	CH ₃ SH	111.2 m ² g ⁻¹	~66% removal (1200 min)	(Adv. Funct. Mater. 2021, 31, 2009000)

						2023, 33, 2301677) (Appl. Catal. B Environ. 2023, 334, 122828) (J. Hazard. Mater. 2023, 452, 131275) (J. Hazard. Mater. 2020, 386, 121957) (J. Catal. 2022, 413, 150-162) (Appl. Catal. B Environ. 2022, 300, 120729) (Environ. Sci. Technol. 2022, 56, 9751- 9761)
Co ₃ O ₄ -4	0.15 g	Hydrothermal	Limonene	4.7 m ² g ⁻¹	52.7 % removal (1200 min)	
Co ₃ O ₄ -5	0.1 g	Hydrothermal	Dichloromethane	21.3 m ² g ⁻¹	~11% removal (120 min)	
Co ₃ O ₄ -6	0.1 g	Co-precipitation	Toluene	16.1 m ² g ⁻¹	~20% conversion (26 h)	
Co ₃ O ₄ -7	0.1 g	Sol-gel	Propane	31 m ² g ⁻¹	~28% conversion (40 h)	
Co ₃ O ₄ -8	0.00103 μmo l	Thermal decomposition	-	17.1 m ² g ⁻¹	TOF 0.24 s ⁻¹	
Co ₃ O ₄ -9	0.05g	Thermal decomposition	O-xylene	35.9 m ² g ⁻¹	~30% conversion (840 min)	
Co ₃ O ₄	0.1g L ⁻¹	Sol-gel	Oxalic acid	0.3 m ² g ⁻¹	~97% removal (60 min)	This work
ZnCo ₂ O ₄	0.1g L ⁻¹	Sol-gel	Oxalic acid	0.4 m ² g ⁻¹	~37% removal	This work

					(60 min)	
CoGa ₂ O ₄	0.1g L ⁻¹	Sol-gel	Oxalic acid	0.3 m ² g ⁻¹	~99% removal (60 min)	This work
3DOM Co ₃ O ₄	0.1g L ⁻¹	Colloidal templating	Oxalic acid	28.1 m ² g ⁻¹	~99% removal (60 min)	This work
Plate Co ₃ O ₄	0.1g L ⁻¹	Molten salt	Oxalic acid	17.5 m ² g ⁻¹	~76% removal (60 min)	This work
Spherical Co ₃ O ₄	0.1g L ⁻¹	Hydrothermal	Oxalic acid	78.4 m ² g ⁻¹	~90% removal (60 min)	This work

In the revised manuscript, we have improved the discussions on the relationships between surface area and the activity of Co₃O₄ based on above results and addressed the significance of accurately identifying and designing Co active sites in catalysis. We hope these clarifications can have your concerns addressed.

Change:

Page 12, Manuscript

Considering the low SSAs of the as-synthesized catalysts (~0.3 m² g⁻¹) which might not facilitate the exposure of the active sites, we subsequently synthesized three dimensional ordered macroporous Co₃O₄ (3DOM-Co₃O₄), plate Co₃O₄, and spherical Co₃O₄, achieving higher SSAs of 29.1, 17.5, and 78.4 m² g⁻¹ accordingly (Supplementary Fig. 20) to investigate the effect of SSAs on catalytic activity. Plate Co₃O₄ and spherical Co₃O₄ display lower activity than that of CoGa₂O₄ (Supplementary Fig. 21), alongside significantly higher cobalt leaching (0.58, 0.66, and 0.09 mg L⁻¹ for plate Co₃O₄, spherical Co₃O₄, and CoGa₂O₄, respectively). 3DOM-Co₃O₄ obtains a similar activity to CoGa₂O₄, yet the synthesis procedure for 3DOM-Co₃O₄ is much more complicated than that of CoGa₂O₄. Furthermore,

the intrinsic activity of Co sites in CoGa_2O_4 is 59~610-fold higher than those of Co_3O_4 with larger SSAs.

Page 13, Manuscript

In addition, CoGa_2O_4 synthesized in this study, despite with small SSAs, are among the highest reported (Fig. 2e; Supplementary Table 10). This can be attributed to the intrinsic activity enhancement by selectively exposing the active $\text{Co}^{2+}_{\text{Td}}$ sites via coordinating environment regulation.

Page 22 and 24, Manuscript

Figure 5h, Comparison of electrical energy per order (EE/O) and fabrication costs for different catalytic ozonation systems.

Economic evaluations on the catalyst fabrication costs and the EE/O (electrical energy per order) suggest that the cobalt-based spinels synthesized in this study have a great advantage on technological economy than the other types of Co_3O_4 -based catalysts (Fig. 5h; Supplementary Fig. 49; Supplementary Tables 13 and 14). For future applications, Ga^{3+} can be replaced by more economical metals. Facile treatments, such as defect engineering and heteroatom doping, can also be applied to the cobalt contained spinels. The optimized ambient microenvironments surrounding the active $\text{Co}^{2+}_{\text{Td}}$

sites can enhance the activity by inducing more intense electronic communications and stronger complexation with the O_3 molecules, which thereby trigger the increased production of surface-bound $\cdot OH$ that are recalcitrant to inorganic anions in wastewater.

Page 42, Supplementary Information

Supplementary Fig. 20. SEM images of 3DOM Co_3O_4 (a), plate-like Co_3O_4 (b), and spherical Co_3O_4 (c). (d) N_2 sorption isotherms of synthesized Co_3O_4 .

Synthesis of 3DOM- Co_3O_4

Three-dimensionally ordered poly methyl methacrylate (PMMA) microspheres with an average diameter of 350 nm were firstly synthesized according to a previously reported method (Chem. Commun. 2008, 6552-6554, ACS Catal. 2019, 9, 7548-7567). First, 120 mL of deionized water was added to a 500 mL four-neck flask, followed by the introduction of argon gas (0.5 L/h) to purge the air inside. The flask was then heated in an oil bath for 30 min to remove any remaining air. Next, 12 mL of methyl methacrylate was added to 120 mL deionized water, and the solution was maintained at 70 °C with stirring at 350 rpm to stabilize the temperature. Subsequently, potassium persulfate (30 mL, 0.12 g preheating to 70 °C) was added, and after 2 h reaction, a milky white liquid was achieved and

transferred to a centrifuge. The centrifugation was conducted at 4000 rpm for 10 h to achieve ordered packing of the microspheres. After centrifugation, the supernatant was decanted, and the solid was dried in an oven at 40 °C for 24 h. The yielded well-ordered PMMA microspheres with a diameter of approximately 350 nm. For synthesis of 3DOM-Co₃O₄, 30 mmol of Co(NO₃)₂·6H₂O and 30 mmol citric acid were completely dissolved in a solution containing 10 mL deionized water and 10 mL methanol under stirring, and then 1.5 g of the as-synthesized ordered PMMA microspheres were soaked into the above solution and kept static for 24 h. After filtration, the solid powders were dried at 50 °C for 24 h, then calcined at 500 °C for 4 h under the air atmosphere. And 3DOM-Co₃O₄ with an average macroporous size of 290 nm was obtained. The yield of 3DOM-Co₃O₄ is around 1.2%.

Synthesis of plate Co₃O₄

In a typical synthesis, 30 mmol of Co(NO₃)₂·6H₂O and 30 mmol citric acid were dissolved in a solution containing 10 mL DI water and 10 mL methanol under stirring, and then the solution was dried at 50 °C for 120 h. After the drying process, the mixed precursors were grinded into fine powders, followed by calcination at 500°C for 4 h under the air atmosphere in a muffle furnace with a heating rate of 5 °C/min. The yield of plate Co₃O₄ is around 8%.

Synthesis of spherical Co₃O₄

To prepare the spherical Co₃O₄, 0.4 mol of Co(NO₃)₂·6H₂O and 0.03 mol of NaOH and were dissolved in 40 mL of DI water with vigorous stirring. After 1 h, the solution was moved into a Teflon-lined autoclave and heated at 180 °C for 5 h. The obtained solution was filtered with DI water and ethyl alcohol and dried overnight at 50 °C. Subsequently, the dried powders were calcined at 500 °C for 3 h in the air atmosphere with a heating rate of 5 °C/min. The yield of spherical Co₃O₄ is around 1.1%.

Supplementary Fig. 21. (a) Catalytic ozonation for OA degradation with synthesized Co_3O_4 samples; (b) Comparison of reaction rates and normalized reaction rates of different $\text{Co}_3\text{O}_4/\text{O}_3$ systems. (c) Cyclic experiments for the as-synthesized Co_3O_4 with high SSAs and the corresponding cobalt leaching concentrations. [catalyst]= 0.1 g L^{-1} ; temperature: $25 \text{ }^\circ\text{C}$; initial pH=3, $[\text{OA}]_0: 50 \text{ mg L}^{-1}$.

The recyclability of these synthesized Co_3O_4 with high SSAs was also evaluated by a multi-cycle reusability test. Clearly, the Co_3O_4 with high SSAs demonstrated much inferior recyclability than the CoGa_2O_4 . Strong passivation was observed after the 3rd cycle. Moreover, elevated cobalt leaching was observed for both the plate-like Co_3O_4 and the spherical Co_3O_4 , as indicated by the inductively coupled plasma mass spectrometry (ICP-MS) results (0.54 and 0.66 mg/L), which exceed the national wastewater discharge standard of China (0.5 mg/L) and contribute to secondary pollution. In contrast, the strong bonding in CoGa_2O_4 crystal structure and the fast reaction kinetics at the catalyst surface guarantee the structural integrity of CoGa_2O_4 during the multi-cycle reusability test. As a result, only

minor passivation was observed after 4 cycles with a negligible amount of leached Co^{2+} (0.09 mg/L).

Page 75, Supplementary Information

Supplementary Fig. 49. Comparison of fabrication costs for different Co-based spinels (a) and EE/O (b) for different reaction systems.

Calculation of fabrication costs

The fabrication costs of catalysts consist of reagents and electricity expense.

$$\text{Cost} = R_{\text{Price}} \cdot \text{Dosage} + P \cdot t \cdot E_{\text{Price}}$$

where, R_{Price} is the unit-price of different reagents (USD/g), Dosage is the reagent dosage (g), P is the input power of instruments (kW), t is the operating time (h), Q is the volume of PCW solution (m^3), E_{Price} is the cost of electricity published by State Grid Corporation of China (0.07 USD/kW·h).

Calculation of expense of electrical energy

The EE/O ($\text{kW}\cdot\text{h}/\text{m}^3$) is defined as the electrical energy ($\text{kW}\cdot\text{h}$) required to reduce the concentration of contaminants by one order of magnitude in one cubic meter of contaminated water. This figure-of-merit serves as a crucial reference for scaling up processes, conducting economic analysis, and

comparing energy efficiency across different treatment technologies. The EE/O was calculated as follows:

$$EE/O = \frac{P \cdot t}{V \cdot \log\left(\frac{C_f}{C_i}\right)}$$

where EE/O is the electrical energy per order ($\text{kW}\cdot\text{h}/\text{m}^3$), P is the rated power (kW) of the catalytic ozonation system, t is the reaction time (min), V is the volume (m^3) of the simulant wastewater solution in the reactor. C_i and C_f are the initial and final states OA concentration, respectively and provided in **Page 19, Supplementary Table 14**.

Supplementary Table 13. Cost assessment of catalyst synthesis.

Catalyst	Item	Unit-price	Dosage	Cost (USD)	Productivity (g)	Price (USD/kg)
Co ₃ O ₄	Citric acid monohydrate	0.002 USD/g	5.674 g	0.0113	1.02	1660
	CoCl ₂ ·6H ₂ O	0.126 USD/g	4.283 g	0.54		
	Magnetic stirrers	0.07 USD/kW·h	0.6 kW·h	0.042		
ZnCo ₂ O ₄	Oven	0.07 USD/kW·h	7.2 kW·h	0.504	1.04	1550
	Muffle furnace	0.07 USD/kW·h	8.5 kW·h	0.595		
	Citric acid monohydrate	0.002 USD/g	5.67 g	0.0113		
CoGa ₂ O ₄	ZnCl ₂	0.117 USD/g	0.818 g	0.096	1.12	6470
	CoCl ₂ ·6H ₂ O	0.126 USD/g	2.855 g	0.36		
	Magnetic stirrers	0.07 USD/kW·h	0.6 kW·h	0.042		
CoGa ₂ O ₄	Oven	0.07 USD/kW·h	7.2 kW·h	0.504	1.12	6470
	Muffle furnace	0.07 USD/kW·h	8.5 kW·h	0.595		
	Citric acid monohydrate	0.002 USD/g	5.67 g	0.0113		

	GaCl ₂	2.8 USD/g	2.113 g	5.92		
	CoCl ₂ ·6H ₂ O	0.126 USD/g	1.43 g	0.18		
	Magnetic stirrers	0.07 USD/kW·h	0.6 kW·h	0.042		
	Oven	0.07 USD/kW·h	7.2 kW·h	0.504		
	Muffle furnace	0.07 USD/kW·h	8.5 kW·h	0.595		
	Citric acid monohydrate	0.002 USD/g	6.3 g	0.0126		
	Co(NO ₃) ₂ ·6H ₂ O	0.062 USD/g	8.73 g	0.541		
	Methanol	0.023 USD/mL	10 mL	0.23		
3DOM Co ₃ O ₄	Polymethyl methacrylate	2.67 USD/g	1.5 g	13.005	0.20	31258
	Magnetic stirrers	0.07 USD/kW·h	0.4 kW·h	0.028		
	Oven	0.07 USD/kW·h	12 kW·h	0.84		
	Muffle furnace	0.07 USD/kW·h	8.5 kW·h	0.595		
	Citric acid monohydrate	0.002 USD/g	6.3 g	0.0126		
	Co(NO ₃) ₂ ·6H ₂ O	0.062 USD/g	8.73 g	0.541		
Plate-like Co ₃ O ₄	Methanol	0.023 USD/mL	10 mL	0.23	1.21	1860
	Magnetic stirrers	0.07 USD/kW·h	0.4 kW·h	0.028		
	Oven	0.07 USD/kW·h	12 kW·h	0.84		
	Muffle furnace	0.07 USD/kW·h	8.5 kW·h	0.595		
	NaOH	0.14 USD/g	1.2 g	0.168		
	Co(NO ₃) ₂ ·6H ₂ O	0.062 USD/g	116.4 g	7.2168		
Spherical Co ₃ O ₄	Magnetic stirrers	0.07 USD/kW·h	0.2 kW·h	0.014	1.32	6250
	Oven	0.07 USD/kW·h	3.6 kW·h	0.252		
	Muffle furnace	0.07 USD/kW·h	8.5kW·h	0.595		

CoAl ₂ O ₄	Citric acid monohydrate	0.002 USD/g	5.67 g	0.0113	1.02	2682
	AlCl ₃ ·6H ₂ O	0.44 USD/g	2.90 g	1.276		
	CoCl ₂ ·6H ₂ O	0.126 USD/g	0.009 g	0.18		
	Magnetic stirrers	0.07 USD/kW·h	0.6 kW·h	0.042		
	Oven	0.07 USD/kW·h	7.2 kW·h	0.504		
	Muffle furnace	0.07 USD/kW·h	9 kW·h	0.72		

Supplementary Table 14. Comparison of energy consumption for simulated wastewater using different spinel oxides based on EE/O concept.

Catalysts	Loading (g L ⁻¹)	Initial OA concentration (mg/L)	System power (kW)	Simulated wastewater volume (mL)	Removal rate (C _f /C _i)
Co ₃ O ₄	0.1	50	0.2	500	0.0286
ZnCo ₂ O ₄	0.1	50			0.61061
CoGa ₂ O ₄	0.1	50			0.07633
3DOM Co ₃ O ₄	0.1	50			0.03553
Plate Co ₃ O ₄	0.1	50			0.23559
Spherical Co ₃ O ₄	0.1	50			0.09414

Reviewer #2

General comment: The manuscript titled "Tailored ozone activation on geometrical-site-dependent cobalt with selective coordination" presents a compelling study on the role of geometrical site-specific cobalt activation in catalytic ozone decomposition. The authors have provided a thorough investigation using a combination of experimental characterization, catalytic performance tests, and theoretical calculations. The findings, particularly concerning the site-specific activation of Co^{2+} and Co^{3+} sites, offer significant insights into the catalytic properties of spinel oxides, which could have broad implications in the field of heterogeneous catalysis. The manuscript is generally well-written and structured, with clear presentations of data and logical flow in the argumentation. However, several aspects of the study need further clarification or additional data to fully substantiate the claims made. Below, I have outlined specific questions and suggestions for the authors to address.

Response:

We very much appreciate your very positive and kind comments on our study and thank you for the valuable suggestions to improve the quality of our manuscript. In the revised manuscript, we added more discussion and provided additional data to clarify the raised concerns according to your suggestions.

Comment 1: You mention changes in surface chemical properties as a factor influencing catalytic activity. Could you provide more detailed data, especially focusing on the chemical environment of different geometrical sites (Co^{2+} and Co^{3+}) and how these might change during the catalytic process?

Response: Thank you for the great question. Indeed, surface chemical properties of transition metal oxides such as surface valence states, Lewis acidic sites and oxygen vacancies can affect catalytic ozonation activity. Our heatmap in Fig. 2f illustrates the correlations of these surface

chemical/electrical properties including average oxidation state, surface acidity, SSA, electrochemical impedance, and surface oxygen vacancy amount to the catalytic activity. We found that these surface properties, except for the exposed geometrical sites, did not act as pivotal factors influencing the activity. To strengthen this viewpoint, in the revised manuscript, *in situ* Raman tests were performed to evaluate the possible variations in the chemical environment of $\text{Co}^{2+}_{\text{Td}}$ and $\text{Co}^{3+}_{\text{Oh}}$ sites during the catalytic ozonation process. Moreover, XPS and O_2 -TPD tests were performed on the used catalysts to investigate the surface chemistry variations.

Change:

Page 14 and 17, Manuscript

Figure 3f, *In situ* Raman spectra of ZnCo_2O_4 and CoGa_2O_4 .

In situ Raman tests reveal a strong signal at $\sim 1020 \text{ cm}^{-1}$ in the $\text{CoGa}_2\text{O}_4/\text{O}_3$ system, which can be ascribed to the produced surface-adsorbed single O atoms ($^*\text{O}$) stemming from the catalytic dissociation of the O_3 on the $\text{Co}^{2+}_{\text{Td}}$ sites (Fig. 3f). Moreover, the $\text{CoGa}_2\text{O}_4/\text{O}_3$ system obtains a higher signal intensity at $\sim 300 \text{ cm}^{-1}$ than the pristine CoGa_2O_4 without a reaction³⁷, suggesting that a large amount of surface-bound $^*\text{OH}$ were formed during the catalytic ozonation process. In contrast, the

lower Raman signals of 1020 and 300 cm^{-1} observed in the $\text{ZnCo}_2\text{O}_4/\text{O}_3$ system than those in the $\text{CoGa}_2\text{O}_4/\text{O}_3$ system reveal that less $\cdot\text{O}$ and surface-bound $\cdot\text{OH}$ are produced because of the weak interaction between the $\text{Co}^{3+}_{\text{Oh}}$ sites and O_3 . Additionally, a peak corresponding to the surface adsorbed O_2 species ($\cdot\text{O}_2$) at $\sim 900 \text{ cm}^{-1}$ emerges in the $\text{ZnCo}_2\text{O}_4/\text{O}_3$ system, indicating that side reactions occur on the $\text{Co}^{3+}_{\text{Oh}}$ sites. We further evaluate the variations of Raman peaks for the stretching vibrations of the $\text{Co}^{2+}_{\text{Td}}$ ($\sim 486 \text{ cm}^{-1}$) and $\text{Co}^{3+}_{\text{Oh}}$ ($\sim 525 \text{ cm}^{-1}$) sites after ozone introduction, which were originated from the stretching of the Co atoms by complexing with O_3 and the subsequent geometrical sites-involved lattice interactions. Clearly, introducing O_3 to CoGa_2O_4 induced less variations in the geometrical sites-involved stretching vibrations than those of ZnCo_2O_4 . This suggests a robust bonding behavior within the lattice frameworks and a minimal change in the surface chemical environment after Ga substituting.

Page 22-23, Manuscript

In contrast, both Co_3O_4 and ZnCo_2O_4 were partially deactivated after multiple runs. XPS deconvolution results on O 1s spectra suggest that ZnCo_2O_4 demonstrated a more significant change in the peaks related to $-\text{OH}/\text{OV}$ for the fresh/used samples than that of the CoGa_2O_4 (Supplementary Fig. 43). In addition, O_2 -TPD profiles further reveal that the area of the OV peak at 200-300 $^\circ\text{C}$ decreases remarkably for the used ZnCo_2O_4 , yet minor changes are observed for the fresh/used CoGa_2O_4 . Therefore, the high surface rigidity with marginal metal leaching (0.3 wt.%) accounts for the superior stability of CoGa_2O_4 .

Page 69, Supplementary Information

Supplementary Fig. 43. High resolution XPS spectra on Co 2p for ZnCo₂O₄ (a) and CoGa₂O₄ (c), High resolution XPS surveys on O1s for ZnCo₂O₄ (b) and CoGa₂O₄ (d). (e) O₂-TPD profiles of ZnCo₂O₄ and CoGa₂O₄.

Comment 2: In the DFT calculations, you discuss the band gap and density of states. Can these be compared with experimental techniques such as UV-Vis spectroscopy or other electronic structure characterization methods? How do you ensure the computational models accurately reflect the experimental samples?

Response: Thanks very much for your insightful comments. In this manuscript, we employed experimental results from various electronic structure characterizations to construct the computational models with high accuracy. Firstly, we employed the EXAFS fitting results from the Co K-edge XANES spectra as the initial parameters to determine the bond lengths of the constructed DFT models.

Secondly, to ensure the spin polarization effects were accurately investigated, we examined the spin states of the as-synthesized spinel oxides by fitting the SQUID results according to Curie – Weiss law ($\chi^{-1} - T$ fitting). And the spin states of the as-synthesized spinel oxides were employed as the crucial parameters for constructing the corresponding models. Thirdly, the accuracy of the constructed computational models was verified by the high consistency between the COHP results of the constructed DFT models and the EXAFS analysis, both suggesting that the unsubstituted cations $\text{Co}^{2+}_{\text{Td}}$ and $\text{Co}^{3+}_{\text{Oh}}$ were the preferentially exposed active sites in CoGa_2O_4 and ZnCo_2O_4 , respectively. In the revised manuscript, we further addressed the accuracy of the constructed computational models.

Change:

Page 17, Manuscript

To illustrate the intrinsic electronic states of the spinel oxides and hybridization strength between Co $3d$ orbitals and O $2p$ orbitals from O_3 molecules and their influences on activity, we constructed Co_3O_4 , CoGa_2O_4 , and ZnCo_2O_4 models with the same exposure of $[111]$ facet, based on the deciphered results from XAS and superconducting quantum interference device (SQUID) magnetization analysis (Supplementary Figs. 31 and 32; Supplementary Table 11), for use in the density functional theory (DFT) calculations. As revealed in the crystal orbital Hamilton population (COHP) analysis, the strong covalency competition arising from the substitution of Ga^{3+} or Zn^{2+} into crystal structure crystal favors the exposure of the unsubstituted cations $\text{Co}^{2+}_{\text{Td}}$ and $\text{Co}^{3+}_{\text{Oh}}$ as the preferentially exposed active sites in CoGa_2O_4 and ZnCo_2O_4 ¹⁴ (Supplementary Fig. 33).

Comment 3. You use EPR and fluorescence probes to detect ROS. Can you provide more details on how the selectivity and sensitivity of these detection methods were validated? How confident are you in the identification of specific ROS like $\cdot\text{OH}$ and $^1\text{O}_2$?

Response: Thank you for the valuable comments. In a heterogeneous catalytic ozonation process, reactive oxygen species such as $\cdot\text{OH}$, $^1\text{O}_2$, and surface-adsorbed oxygen atom/ O_3 can be possibly generated, participating in the oxidation of the organics. In this study, a combined strategy using spin-trapping EPR technique and fluorescence probes was employed to investigate the types of the generated ROS. To verify the accuracy and selectivity of these methods for $\cdot\text{OH}$ and $^1\text{O}_2$ detection, their characteristic signals were tested using the classic homogeneous Fenton system ($\text{Fe}^{2+}/\text{H}_2\text{O}_2$) (Nat. Rev. Chem. 2021, 5, 595-597) (Nat. Rev. Chem. 2021, 5, 595-597) and the homogeneous dye-activated photocatalytic system (methylene blue/solar light) (Coord. Chem. Rev. 2002, 233-234, 351-371), which produce only $\cdot\text{OH}$ and $^1\text{O}_2$, respectively.

Figure R5. (a) EPR spectra with DMPO as the spin trapping agent for $\cdot\text{OH}$ produced in the homogeneous Fenton system and the homogeneous dye-activated photocatalytic system; (b) EPR spectra with DMPO as the spin trapping agent in nonradical species-dominated heterogeneous catalytic ozonation system and effect of methanol as a quenching agent for $\cdot\text{OH}$. Nonradical EPR spectra were

adapted from our previous publication (Appl. Catal. B-Environ. 2019, 254, 283-291). (c) EPR spectra with TEMP as the spin trapping agent for $^1\text{O}_2$ produced in the homogeneous Fenton system and the homogeneous dye-activated photocatalytic system. (d) Visualization of the generated $\cdot\text{OH}$ at the surface of spinel oxides using coumarin as the $\cdot\text{OH}$ fluorescence probe by inverted fluorescence microscopy (IFM) in the homogeneous Fenton system and the homogeneous dye-activated photocatalytic system. (e) Photoluminescence spectra of singlet oxygen sensor green (SOSG) as the $^1\text{O}_2$ probe in the homogeneous Fenton system and the homogeneous dye-activated photocatalytic system.

In the EPR tests, strong signals for DMPO- $\cdot\text{OH}$ adducts, with the hyperfine splitting couplings of $a_{\text{N}} = a_{\text{H}} = 14.9$ G and intensity ratio of 1:2:2:1, were detected in the Fenton system, yet no such signals were observed in dye-activated photocatalytic system (Fig. R5a). These results suggest that DMPO serves as a spin-trapping agent selective for $\cdot\text{OH}$ over $^1\text{O}_2$. Moreover, we compared the EPR signals in this study to our previously reported ones that are dominated by the nonradical species (i.e., surface-adsorbed oxygen atom and O_3) to reinforce the sensitivity of EPR spin-trapping tests (Fig. R5b). In catalytic ozonation dominated by surface-adsorbed nonradical species, no such strong DMPO- $\cdot\text{OH}$ adducts signals were observed and the signal intensity of the DMPO- $\cdot\text{OH}$ adducts was similar to that of the $\text{H}_2\text{O}/\text{O}_3$ system without the presence of a catalyst. For the $\text{CoGa}_2\text{O}_4/\text{O}_3$ system producing massive $\cdot\text{OH}$, the addition of methanol as the $\cdot\text{OH}$ quencher significantly decreased the intensity of DMPO- $\cdot\text{OH}$ adducts, resulting in the formation of carbon-centered radicals, which are the oxidation products of methanol by $\cdot\text{OH}$. In our EPR tests using TEMP as the spin-trapping agent, methanol (1 M) was added to eliminate the generated $\cdot\text{OH}$, as $\cdot\text{OH}$ can also oxidize TEMP to form TEMPO, which may mislead the results. As shown in Fig. R5c, strong triplet signals with an intensity ratio of 1:1:1

were observed in the methylene blue/solar light photocatalytic system; however, no such signals were detected in the Fenton system. Furthermore, a higher intensity of triplet signals in the $\text{ZnCo}_2\text{O}_4/\text{O}_3$ system than that in the $\text{Co}_3\text{O}_4/\text{O}_3$ system suggested a greater production of $^1\text{O}_2$.

In the tests utilizing fluorescence probes, coumarin was selected as the $\cdot\text{OH}$ probe to detect the produced 7-hydroxycoumarin (7-HC) as the selectively oxidation product, whose fluorescence can be observed by an inverted fluorescence microscope. Strong fluorescence signals of 7-HC were observed on the partially dissolved FeCl_2 solid in the Fenton system, whereas no significant fluorescence was detected in the $^1\text{O}_2$ -generating methylene blue/solar light photocatalytic system. The addition of methanol as the $\cdot\text{OH}$ scavenger quenched the blue fluorescence signals of 7-HC generated in the Fenton system. As a highly selective probe for $^1\text{O}_2$, singlet oxygen sensor green (SOSG) can be oxidized into the fluorescence form by $^1\text{O}_2$ with an excitation wavelength of 504 nm and an emission wavelength of 525 nm. Similar observations to those of coumarin-based results were noted in the application of SOSG for $^1\text{O}_2$ detection. According to the fluorescence spectra, marginal signal responses were observed in the Fenton system and $\text{O}_3/\text{H}_2\text{O}$ system (without catalyst), however, a strong fluorescence signal was detected in the methylene blue/solar light photocatalytic system. This strong deviation confirms the high selectivity of SOSG as a fluorescence probe for distinguishing $^1\text{O}_2$ from $\cdot\text{OH}$.

In the revised manuscript, we added these proofs on the selectivity and sensitivity of the detection methods in the Supplementary Information.

Change:

Page 15, Manuscript

The high accuracy and selectivity of these techniques for ROS detection have been verified using the classic homogeneous systems designed to produce single type of ROS (Supplementary Fig. 26).

Supplementary Fig. 26. (a) EPR spectra with DMPO as the spin trapping agent for $\cdot\text{OH}$ produced in the homogeneous Fenton system and the homogeneous dye-activated photocatalytic system; (b) EPR spectra with DMPO as the spin trapping agent in nonradical species-dominated heterogeneous catalytic ozonation system and effect of methanol as a quenching agent for $\cdot\text{OH}$. Nonradical EPR spectra was adapted from our previous publication²⁹. (c) EPR spectra with TEMP as the spin trapping agent for ¹O₂ produced in the homogeneous Fenton system and the homogeneous dye-activated photocatalytic system. (d) Visualization of the generated $\cdot\text{OH}$ on the surface of spinel oxides using coumarin as the $\cdot\text{OH}$ fluorescence probe by inverted fluorescence microscopy (IFM) in the homogeneous Fenton system and the homogeneous dye-activated photocatalytic system. (e) Photoluminescence spectra of singlet oxygen sensor green (SOSG) as the ¹O₂ probe in the homogeneous Fenton system and the homogeneous dye-activated photocatalytic system.

In this study, a combined strategy using the spin-trapping EPR technique and fluorescence probes was

employed to investigate the types of the generated ROS. To verify the selectivity of these methods for $\cdot\text{OH}$ and $^1\text{O}_2$ detection, their characteristic signals were validated using the classic homogeneous Fenton system ($\text{Fe}^{2+}/\text{H}_2\text{O}_2$) and the homogeneous dye-activated photocatalytic system (methylene blue/solar light) that are designed for the production of $\cdot\text{OH}$ and $^1\text{O}_2$ as the single type of ROS, respectively.

In the EPR tests, strong signals for DMPO- $\cdot\text{OH}$ adducts, with the hyperfine splitting couplings of $a_{\text{N}} = a_{\text{H}} = 14.9$ G and intensity ratio of 1:2:2:1, were detected in the Fenton system, yet no such signals were observed in the dye-activated photocatalytic system (**Supplementary Fig. 26a**). These results suggest that DMPO serves as a spin-trapping agent selective for $\cdot\text{OH}$ over $^1\text{O}_2$. Moreover, we compared the EPR signals in this study to our previously reported ones that are dominated by the nonradical species (i.e., surface-adsorbed oxygen atom and O_3) to reinforce the sensitivity of EPR spin-trapping tests (**Supplementary Fig. 26b**). In catalytic ozonation dominated by surface-adsorbed nonradical species, no such strong DMPO- $\cdot\text{OH}$ adducts signals were observed and the signal intensity of the DMPO- $\cdot\text{OH}$ adducts was similar to that of $\text{H}_2\text{O}/\text{O}_3$ system without the presence of a catalyst. For the $\text{CoGa}_2\text{O}_4/\text{O}_3$ system producing massive $\cdot\text{OH}$, the addition of methanol as the $\cdot\text{OH}$ quencher significantly decreased the intensity of DMPO- $\cdot\text{OH}$ adducts, resulting in the formation of carbon-centered radicals, which are the oxidation products of methanol by $\cdot\text{OH}$. In our EPR tests using TEMP as the spin-trapping agent, methanol (1 M) was added to eliminate the generated $\cdot\text{OH}$, as $\cdot\text{OH}$ can also oxidize TEMP to form TEMPO, which may mislead the results. As shown in **Supplementary Fig. 26c**, strong triplet signals with an intensity ratio of 1:1:1 were observed in the methylene blue/solar light photocatalytic system; however, no such signals were detected in the Fenton system. Furthermore, a higher intensity of triplet signals in the $\text{ZnCo}_2\text{O}_4/\text{O}_3$ system than that in the $\text{Co}_3\text{O}_4/\text{O}_3$ system suggested a greater production of

$^1\text{O}_2$.

In the tests utilizing fluorescence probes, coumarin was selected as the $\cdot\text{OH}$ probe to detect the produced 7-hydroxycoumarin (7-HC) as the selectively oxidation product, whose fluorescence can be observed by an inverted fluorescence microscope. Strong fluorescence signals of 7-HC were observed on the partially dissolved FeCl_2 solid in the Fenton system, whereas no significant fluorescence was detected in the $^1\text{O}_2$ -generating methylene blue/solar light photocatalytic system (**Supplementary Fig. 26d**). The addition of methanol as the $\cdot\text{OH}$ scavenger quenched the blue fluorescence signals of 7-HC generated in the Fenton system. As a highly selective probe for $^1\text{O}_2$, singlet oxygen sensor green (SOSG) can be oxidized into the fluorescence form by $^1\text{O}_2$ with an excitation wavelength of 504 nm and an emission wavelength of 525 nm. Similar observations to those of coumarin-based results were noted in the application of SOSG for $^1\text{O}_2$ detection (**Supplementary Fig. 26e**). According to the fluorescence spectra, marginal signal responses were observed in the Fenton system and $\text{O}_3/\text{H}_2\text{O}$ system (without a catalyst), however, a strong fluorescence signal was detected in the methylene blue/UV photocatalytic system. This strong deviation confirms the high selectivity of SOSG as a fluorescence probe for distinguishing $^1\text{O}_2$ from $\cdot\text{OH}$.

Comment 4. The manuscript reports the reusability of the CoGa_2O_4 catalyst. Could you provide more data on the long-term stability of the catalyst, especially after multiple catalytic cycles? What degradation mechanisms, if any, were observed?

Response: Thank you very much for your comments and suggestions. In the revised manuscript, *in situ* Raman tests were performed to evaluate the possible variations in the chemical environment of $\text{Co}^{2+}_{\text{Td}}$ and $\text{Co}^{3+}_{\text{Oh}}$ sites during the catalytic ozonation process. Moreover, XPS and O_2 -TPD tests were performed on the used CoGa_2O_4 and ZnCo_2O_4 after 5th use cycle to investigate the surface chemistry

variations compared to the fresh catalysts and probe the possible degradation mechanisms. For the details, please refer to the Response to Comment 1.

Comment 5: The manuscript discusses the selective production of $\cdot\text{OH}$ over $^1\text{O}_2$ at different cobalt sites. Can you provide quantitative data or modeling results to support the observed selectivity? How do you propose to control this selectivity in practical applications?

Response: Thank you very much for the great questions. In this study, DFT simulation has been employed to investigate the selective production of $\cdot\text{OH}$ over $^1\text{O}_2$ at $\text{Co}^{2+}_{\text{Td}}$ sites. The identification of the $^1\text{O}_2$ production by DFT simulation was achieved by analyzing the magnetic moments of the produced “ O_2 free” species as the O_3 catalytic dissociation product on the active sites. As described in the manuscript, “the magnetic moments for O_2 free generated on $\text{Co}^{2+}_{\text{Td}}$ and $\text{Co}^{3+}_{\text{Oh}}$ sites are calculated as 2 and 0 μ_{B} accordingly, corresponding to the normal O_2 molecule and the $^1\text{O}_2$ molecule, respectively”. Moreover, Bader charge analysis revealed that the dissociation of O_3 into $^1\text{O}_2$ on $\text{Co}^{3+}_{\text{Oh}}$ sites required a greater number of electrons than on $\text{Co}^{2+}_{\text{Td}}$ sites. This finding explained the low OUE and $\cdot\text{OH}$ production rate on $\text{Co}^{3+}_{\text{Oh}}$ sites as well as the high selectivity of $\cdot\text{OH}$ over $^1\text{O}_2$ at $\text{Co}^{2+}_{\text{Td}}$ sites. In the revised manuscript, we modified the figures to address the selective production of $\cdot\text{OH}$ over $^1\text{O}_2$ at $\text{Co}^{2+}_{\text{Td}}$ sites.

We further quantified the production of $\cdot\text{OH}$ in the $\text{CoGa}_2\text{O}_4/\text{O}_3$, $\text{ZnCo}_2\text{O}_4/\text{O}_3$, and $\text{Co}_3\text{O}_4/\text{O}_3$ systems based on their oxalic acid oxidation efficiencies. Oxalic acid is a highly recalcitrant aliphatic acid that exhibits quite slow reaction kinetics with ROS having low oxidation potentials such as $^1\text{O}_2$ and $\text{O}_2\cdot^-$, but can be rapidly oxidized by $\cdot\text{OH}$ (ACS EST Eng. 2020, 1, 32-45; ACS Catal. 2022, 12, 2770-2780). In this study, ROS detection characterizations revealed that $\cdot\text{OH}$ and $^1\text{O}_2$ were the two possible ROS generated in the O_3 activation by cobalt-based spinels. Therefore, oxalic acid can be used as a probe

to quantified the production of $\cdot\text{OH}$. Calculation details were provided in Note 2 in Supplementary Information. Compared to the catalytically inactive ZnCo_2O_4 , the highly active CoGa_2O_4 obtained a 3.0-fold increase in the $\cdot\text{OH}/\text{O}_3$ conversion rate, verifying the high selectivity of $\text{Co}^{2+}_{\text{Td}}$ sites towards $\cdot\text{OH}$ production. Production of $^1\text{O}_2$ as the byproduct ROS might stem from the different O_3 adsorption and intermediate evolution routes on $\text{Co}^{3+}_{\text{Oh}}$ sites, which required a higher electron consumption and a decreased O_3 utilization efficiency. In the revised manuscript, we modified the corresponding discussion to address the quantification results.

In practical applications, the high concentrations of inorganic ions (both cations and anions) in real wastewater increase the ionic strength and affect the outer-sphere complexation of ozone at catalytic active sites. These ions also function as radical scavengers, eliminating free radicals in the reaction solution.

Figure R6. Influences of inorganic anions on the catalytic performance for the $\text{CoGa}_2\text{O}_4/\text{O}_3$ system (a), $\text{Co}_3\text{O}_4/\text{O}_3$ system (b) and ZnCo_2O_4 system (c).

Our research found that the strong interaction between the $\text{Co}^{2+}_{\text{Td}}$ sites in CoGa_2O_4 and O_3 resulted in an inner-sphere complexation, minimizing the interference of solution ionic strength. Additionally, the abundant surface-bound $\cdot\text{OH}$ can counteract the quenching effects of inorganic anions, ensuring the high oxidation efficiency. As a result, CoGa_2O_4 maintained high catalytic activity in the presence of the different inorganic ions (as reflected in Fig. R6a) and exhibited a remarkable performance in

treating real wastewater. In contrast, the weak interaction between $\text{Co}^{3+}_{\text{Oh}}$ sites and O_3 resulted in the outer-sphere complexation, which can be heavily interfered by ionic strength. Additionally, the fewer production of the surface-bound $\cdot\text{OH}$ on $\text{Co}^{3+}_{\text{Oh}}$ sites further reduced the oxidation capability (Fig. R6b and c). Notably, $^1\text{O}_2$, a nonradical species that is resistant to inorganic anions, was generated on $\text{Co}^{3+}_{\text{Oh}}$ sites, yet its low oxidation potential and slow reaction rate were insufficient to achieve deep oxidation of pollutants. Based on the above analysis, the strong interaction with O_3 and the high selectivity for producing $\cdot\text{OH}$ over $^1\text{O}_2$ offers promising future research opportunities to regulate the activity of $\text{Co}^{2+}_{\text{Td}}$ sites. Further optimization of the intrinsic activity of the $\text{Co}^{2+}_{\text{Td}}$ sites can be achieved by regulating their ambient microenvironments, such as defect engineering and high-entropy engineering, to induce more intense electronic communications and stronger complexation with O_3 molecules, and to trigger the higher production of surface-bound $\cdot\text{OH}$ recalcitrant to inorganic ions in the real wastewater. In the revised manuscript, practical application possibilities and the future research opportunities to control the selectivity of producing $\cdot\text{OH}$ over $^1\text{O}_2$ on the $\text{Co}^{2+}_{\text{Td}}$ sites have been addressed.

Change:

Page 19, Manuscript

Figure 4e-f, O_3 adsorption configurations at TS and FS and the corresponding charge density difference

(CDD) plots at the $\text{Co}_3\text{O}_4\text{-Co}^{2+}_{\text{Td}}$ site (e) and the $\text{Co}_3\text{O}_4\text{-Co}^{3+}_{\text{Oh}}$ site (f).

Page 11 and 17, Manuscript

Figure 2d, Comparison of turn over frequency (TOF) and $\cdot\text{OH}$ production rates of different spinels/ O_3 systems.

The selectivity of the $\text{Co}^{2+}_{\text{Td}}$ and $\text{Co}^{3+}_{\text{Oh}}$ sites on $\cdot\text{OH}$ production was assessed by the $\cdot\text{OH}/\text{O}_3$ conversion rates. Compared to the catalytically inactive ZnCo_2O_4 , the highly active CoGa_2O_4 obtained a 3.0-fold increase in the $\cdot\text{OH}/\text{O}_3$ conversion rate (Fig. 2d), confirming the high selectivity of $\text{Co}^{2+}_{\text{Td}}$ sites towards $\cdot\text{OH}$ production. The production of $^1\text{O}_2$ as the byproduct ROS might stem from the distinct O_3 adsorption and intermediate evolution routes on $\text{Co}^{3+}_{\text{Oh}}$ sites, which require higher electron consumption and result in a decreased O_3 utilization efficiency.

Page 22 and 23, Manuscript

Figure 5h, Comparison of electrical energy per order (EE/O) and fabrication costs for different catalytic ozonation systems.

Economic evaluations on the catalyst fabrication costs and the EE/O (electrical energy per order) suggest that the cobalt-based spinels synthesized in this study have a great advantage on technological economy than the other types of Co_3O_4 -based catalysts (Fig. 5h; Supplementary Fig. 49; Supplementary Tables 13 and 14). For future applications, Ga^{3+} can be replaced by more economical metals. Facile treatments, such as defect engineering and heteroatom doping, can be applied to the cobalt contained spinels. The optimized ambient microenvironments surrounding the active $\text{Co}^{2+_{\text{Td}}}$ sites can enhance the activity by inducing more intense electronic communications and stronger complexation with the O_3 molecules, which thereby trigger the increased production of surface-bound $\cdot\text{OH}$ that are recalcitrant to inorganic anions in wastewater.

Comment 6: How does varying the catalyst loading affect the observed catalytic activity and ROS production? Have you tested different loadings to optimize the catalytic performance, and if so, what trends were observed?

Response: Thank you for your valuable questions. We investigated the activity of CoGa_2O_4 catalysts under different catalyst loadings and ozone dosages (Fig. R7). The results indicated that increasing the catalytic loading and the ozone dosage beyond 0.1 g/L and 25 mg/L accordingly resulted in minor improvements of the catalytic activity. Excessive catalyst loading and the ozone dosage beyond those points led to a markedly diminished marginal benefit for improving the catalytic efficiency. Although a higher catalyst loading offers more active sites, the concentration of dissolved ozone in solution limits the reaction rate because the active sites on the catalysts have a high affinity for dissolved ozone but are relatively inert to gaseous O_3 . Therefore, a catalyst loading of 0.1 g/L and an ozone dosage of

25 mg/L were identified as the optimum operating parameters in this study.

Figure R7. Effect of catalytic loading (a) and O₃ dosage (c) on degradation of oxalic acid for CoGa₂O₄ and their corresponding rate constants comparison (b and d) Catalyst loading: 0.1 g L⁻¹; temperature: 25 °C; ozone flow rate: 100 mL min⁻¹; ozone concentration: 25 mg L⁻¹; initial pH was adjusted by adding 0.01 M H₂SO₄/NaOH in OA solution. [OA]₀: 50 mg L⁻¹.

In the revised manuscript, discussion on effect of catalyst loading and ozone dosage on catalytic performance has been added.

Change:

Page 12, Manuscript

In the catalytic O₃ activation, geometrical-site-dependent activities are observed. CoGa₂O₄ with substituted Co³⁺_{Oh} sites displays a higher performance to the pristine Co₃O₄ in OA destruction, achieving over 95% destruction of the initial OA (50 mg L⁻¹) within 45 min (Fig. 2a, details on optimizing the catalytic loading/O₃ dosage and influence of initial solution pH are provided in Supplementary Fig. 18 and 19, respectively).

Supplementary Fig.18. Effect of catalytic loading (a) and O_3 dosage (c) on degradation of oxalic acid for CoGa_2O_4 and their corresponding rate constants comparison (b and d) Catalyst loading: 0.1 g L^{-1} ; temperature: $25 \text{ }^\circ\text{C}$; ozone flow rate: 100 mL min^{-1} ; ozone concentration: 25 mg L^{-1} ; initial pH was adjusted by adding $0.01 \text{ M H}_2\text{SO}_4/\text{NaOH}$ in OA solution. $[\text{OA}]_0$: 50 mg L^{-1} .

We investigated the activity of CoGa_2O_4 catalysts under different catalyst loadings and ozone dosages. The results indicated that increasing the catalytic loading and the ozone dosage beyond 0.1 g/L and 25 mg/L accordingly resulted in minor improvements of the catalytic activity. Excessive catalyst loading and the ozone dosage beyond those points led to a markedly diminished marginal benefit for improving the catalytic efficiency. Although a higher catalyst loading offers more active sites, the concentration of dissolved ozone in solution limits the reaction rate because the active sites on the catalysts have a high affinity for dissolved ozone but are relatively inert to gaseous O_3 . Therefore, a catalyst loading of 0.1 g/L and an ozone dosage of 25 mg/L were identified as the optimum operating parameters in this

study.

Comment 7: How does the performance of your catalysts compare with other established catalysts for ozone activation? Can you provide a more detailed comparison, possibly in a tabular form, that includes metrics like turnover frequency, O₃ utilization efficiency, and ROS production rates?

Response: Thanks very much for your question and we accept your suggestion in the revised manuscript.

We compared the catalytic ozonation performance of various reported metal catalysts, metal-supported catalysts, carbon catalysts, and carbon-composite metal catalysts with the as-synthesized Co-contained spinels. Unfortunately, not all reported studies provided the relevant data on ozone utilization efficiency and turnover frequency (TOF). Therefore, we employed S_{BET} normalized activity efficiency (AE), which can be derived from the following equations, to compare their catalytic performances.

$$AE = \frac{\text{Reaction rate}}{\text{catalyst loading (g)}}$$
$$\text{Normalized AE} = \frac{AE}{S_{\text{BET}} (\text{m}^2 \text{ g}^{-1})}$$

Figure R8. Comparison of the catalytic ozonation performances of the as-synthesized catalysts in this study with the previously report ones.

Displayed in Fig. R8, CoGa_2O_4 synthesized in this study exhibited exceptionally high activity compared to the various types of reported catalysts. This exceptional activity can be attributed to the intrinsic activity enhancement by selectively exposing the active $\text{Co}^{2+}_{\text{Td}}$ sites via coordinating environment regulations.

Change:

Page 11 and 13, Manuscript

Figure 2e. Comparison of the catalytic ozonation performance of our spinel oxides catalysts with values from previous reports.

In addition, CoGa_2O_4 synthesized in this study, despite with small SSAs, are among the highest reported (Fig. 2e and Supplementary Table 10). This can be attributed to the intrinsic activity enhancement by selectively exposing the active $\text{Co}^{2+}_{\text{Td}}$ sites via coordinating environment regulations.

Page 13, Supplementary Information

Supplementary Table 10. Comparison of the catalytic ozonation activities using different types of heterogeneous catalysts.

Catalyst	Pollutants	Conditions	BET SSAs	Main ROS production	Efficiency	Ref.
$\delta\text{-MnO}_2$	p-Chlorophenol	[Cat.] = 0.5 g/L	271.1 m^2/g	$\cdot\text{OH}$, $^1\text{O}_2$, $\text{O}_2\cdot^-$	97% removal	2

Catalyst	Pollutant	[Cat.] (g/L)	[Poll.] (mg/L)	[O ₃] (mg/min)	[Time] (min)	Rate (m ² /g)	Radicals	Efficiency (%)	Removal (min ⁻¹)	Ref.
CaMn ₄ O ₈	p-Nitrophenol	0.1	100	5	30	8.8	¹ O ₂ , O ₂ ^{•-}	99%	0.227	3
								removal	0.125	
								80.5%	0.0845	
MWCNTs	Oxalic acid	0.1	96	9	40	115	•OH	97%	0.60	4
								removal	0.123	
Fe ₃ O ₄ /MWCNTs	Sulfamethazine	0.5	20	25	10	130	•OH	86%	0.031	5
								removal	0.0741	
2% Cu/SBA-15	Reactive Orange 4	0.2	100	5	60	432.3	•OH	98%	0.0917	6
								removal	0.0668	
Mn-CeO _x /γ-Al ₂ O ₃	Bromaminic acid	1.0	50	2.5	120	264.8	•OH	90.6%	0.0741	7
								removal	0.0741	
MgO[111]	Nitrobenzene	1.0	50	0.5	30	187.8	•OH	94%	0.0668	8
								removal	0.0668	
MgFe ₂ O ₄	Acid Orange II	0.1	50	5.0	40	84.2	•OH	99%	0.123	9
								removal	0.123	
MnO ₂ /rGO	p-NP	0.1	50	0.9	60	35.2	¹ O ₂ , O ₂ ^{•-}	99%	0.123	10
								removal	0.123	
MWCNTs/Fe ₃ O ₄	Bisphenol A	0.5	50	0.9		93.0	•OH	90%	0.0579	11
								removal	0.0579	

FeMgO/C NT	phenol	[Cat.] = 1 g/L [Poll.] = 400 mg/L	40 min [O ₃] = 1.5 mg/min [Time] = 60 min	120 m ² /g	•OH	79% removal 0.010 min ⁻¹	12
Fe/KCC	SMT	[Cat.] = 0.3 g/L [Poll.] = 20 mg/L	6.0 mg/min [Time] = 15 min	464.5 m ² /g	•OH	100% removal 0.29 min ⁻¹	13
rGO	p-Hydroxylbenzoic Acid p-HBA	[Cat.] = 0.1 g/L [Poll.] = 20 mg/L	[O ₃] = 2 mg/min [Time] = 60 min	305 m ² /g	¹ O ₂ , O ₂ ^{•-}	95% removal 0.083 min ⁻¹	14

Reviewer #3

General comment: The manuscript by Wang et. al., prepared Co-based spinel oxides with specific Co sites (i.e., $\text{Co}^{2+}_{\text{Td}}$, and $\text{Co}^{3+}_{\text{Oh}}$) to investigate the geometrical-site dependent ozone activation performance. By combing experimental results and DFT calculations, they found that highly spin-polarized $\text{Co}^{2+}_{\text{Td}}$ sites were the active sites for selectively producing surface-bound hydroxyl radicals ($\bullet\text{OH}$) of Co-based spinel oxides. However, Co-based oxides have been widely used in oxygen-related catalysis, and the geometrical-site dependent relationship have been widely investigated. Moreover, the authors used the same images for different samples, for example CoGa_2O_4 in Figure 1e is the same as Figure S3a for Co_3O_4 . I think the authors used the same images to represent different samples by artificially changing the magnification of the image. Therefore, I suggest to reject this manuscript.

Response: We sincerely appreciate your time and effort in reviewing our manuscript and providing valuable feedback. Our study goes beyond merely investigating the geometrical-site-dependent relationships in Co-based oxides-driven catalysis. Its primary objectives are to elucidate the intrinsic active sites in cobalt-based spinel oxides and to address the spin-polarization effects caused by geometrical site competition on reaction thermodynamics in transition metal oxide-catalyzed processes.

Our findings reveal that geometrical site competition not only governs catalytic activity but also modulates reaction thermodynamics, thereby influencing selectivity. Notably, the highly reactive $\text{Co}^{2+}_{\text{Td}}$ sites demonstrate remarkable selectivity in producing $\bullet\text{OH}$ over the less favorable $^1\text{O}_2$ during O_3 activation. These insights help resolve the conflicting conclusions in previous studies, which often stemmed from uncontrolled crystal facets/sizes and neglected spin-polarization effects. For details on this issue, please refer to the Response to Reviewer #1.

In response to your concerns regarding the image manipulation, we must clarify that the TEM images for CoGa_2O_4 (Figure 1e) and Co_3O_4 (Figure S3) are distinctively different, which are compared in Fig. R9. Furthermore, there have been no issues related to image or data manipulation in this manuscript.

Figure R9. Comparison of TEM images of CoGa_2O_4 and Co_3O_4 .

In the revised manuscript, we included additional discussion and provided supplementary data to address the raised concerns in accordance with your suggestions. And we hope the response to your comments and the modifications in the revised manuscript would well resolve your concerns.

Comment 1. In Figure 1e, why is the magnification of different samples different? Please use images with the same magnification for different samples.

Response: Thanks for the comments. In the revised manuscript, the same magnification has been applied to the different samples in TEM images.

Change:

Page 7, Manuscript

Figure 1b, Transmission electron microscopy (TEM) images of Co_3O_4 , ZnCo_2O_4 and CoGa_2O_4 in [111]

orientation.

Comment 2: From Figure 1e, the size of synthesized Co-based spinel oxides is quite uneven. How did the authors ensure the exposed area of [111] plane is the same. If the area is different, the different performance may be attributed to the different number of active sites.

Response: Thanks for the comments. We provided the particle size distributions for Co_3O_4 , ZnCo_2O_4 , and CoGa_2O_4 based on macroscopic SEM observations (Supplementary Fig. 7) and we append this figure as Fig. R10 for ease of reference. These catalysts demonstrated centered particle size distributions with average sizes ranging from 327 to 466 nm, offering a robust platform for evaluating the intrinsic reactivity differences stemming from geometrical sites. In the revised manuscript, we have addressed the centered particle size distributions of these as-synthesized Co-based spinels.

Figure R7. Particle size distributions of (a) Co_3O_4 , (b) ZnCo_2O_4 and (c) CoGa_2O_4 octahedra prepared by sol-gel method obtained from SEM observations; (d) Average particle sizes of the as-prepared three samples.

Change:

Page 7, Manuscript

Scanning electron microscopy (SEM) observations reveal that these spinel oxides demonstrate a similar octahedral morphology and centered particle size distributions with average sizes ranging from 327 to 466 nm (Fig. 1c; Supplementary Figs. 6a-c). This centric particle size range offers a solid basis for evaluating the intrinsic reactivity differences arising from geometrical sites (Supplementary Fig. 7).

Comment 3: The morphological description of the synthesized oxides “In the high-resolution transmission electron microscopy (HRTEM) images of the pristine Co_3O_4 and its substitutes, the lattice spacing of 0.47 nm was observed at the edges, attributing to the [111] facet (Fig. 1f-1g). Additionally, a fixed angle of 60° was observed between the (220), (422), and (202) planes in the corresponding selected area electron diffraction (SAED) images (Supplementary Fig. 8), confirming that the [111] facet was the common exposed facet on their octahedrons”, indicating that Co_3O_4 , ZnCo_2O_4 and CoGa_2O_4 nanoparticles have similar polyhedron-like morphologies but inevitable diameter changes resulting from the different ion sizes and electronic states of introduced additional elements (i.e., Zn and Ga) with respect to Co.” to be not very fortunate. If the mechanisms of the catalytic processes have to be modeled in detail, the surface structure of the main exposed surface should be identified. The determination of the main catalytic crystal surface [111] is not rigorous enough.

Response: Thanks for your comments. In this study, we employed the relative spatial positions of the crystal planes in the reciprocal space (Adv. Mate. 2018, 30, 1802000), which we believe is a relatively rigorous method, to determine the exposed crystal facet (Adv. Mater. 2022, 34, 2203320). The method is schematically illustrated in Supplementary Figure 8, which is also appended below for ease of

reference.

Figure R11. (a) Schematic illustration of octahedral spinel Co_3O_4 with (111) facets. Atomic structure of [111] surface of the top view (b). Standard diffraction pattern of spinel Co_3O_4 in [111] direction in

the reciprocal space (c). SAED images for Co_3O_4 (d), ZnCo_2O_4 (e), and CoGa_2O_4 (f). Atomic-resolved high-resolution TEM (HRTEM) images for Co_3O_4 (g), ZnCo_2O_4 (h), and CoGa_2O_4 (i)

Crystal spacings (norms of the vectors) and angles of the crystal planes measured in the reciprocal space (such as FFT patterns and SAED plots) are of great significance to indexing the diffraction vectors and determining the exposed crystal facet (Adv. Mater. 2022, 34, 2203320). Typically, three crystal planes parallel to the same line (the normal line perpendicular to the diffraction plane) are required to be indexed. And these three crystal planes need to satisfy the parallelogram rule: $V_{h_1k_1l_1} + V_{h_2k_2l_2} = V_{h_3k_3l_3}$. As shown in Fig. R11c, the crystal planes determining the [111] facet are $V_{(220)}$, $V_{(202)}$, and $V_{(422)}$, respectively, with the interplanar angle of 60° between the (220) and (202) crystal plane. The interplanar crystal spacing between these crystal planes is 3.67 nm^{-1} . For the Co_3O_4 , CoGa_2O_4 , and ZnCo_2O_4 synthesized in this study, (220), (202), and (422) crystal planes were identified in their SAED patterns (Fig. R11d-f). Additionally, the interplanar angle between the (220) and (202) planes for these three catalysts was 60° , with the crystal spacing of 3.67 nm^{-1} . These identification results confirm that Co_3O_4 , CoGa_2O_4 , and ZnCo_2O_4 share the same main exposed facet of [111], though the introduced Ga^{3+} and Zn^{2+} within the lattice backbones brought inevitable changes in particle sizes due to the different ion sizes and electronic states. Furthermore, in the revised manuscript, we further determined the exposed facet by atom-resolved surface structure characterization based on HRTEM observations (Fig. R11g-i). All Co_3O_4 , CoGa_2O_4 , and ZnCo_2O_4 samples exhibit the same characteristic atom distributions as those for the theoretical [111] facet.

In the revised manuscript, we have modified Fig. 1 and rewritten the corresponding discussion on the exposed facet determination to provide more key information on the method we employed.

Change:

Figure 1 b, Transmission electron microscopy (TEM) images of Co_3O_4 , ZnCo_2O_4 and CoGa_2O_4 in [111] orientation. **c-d**, Atomic-resolved high-resolution TEM (HRTEM) image (**c**) and the corresponding selected area electron diffraction (SAED) image (**d**) of Co_3O_4 .

In this study, the relative spatial positions of the lattice planes in reciprocal space were measured to determine their main exposed crystal facets by identifying the angles and spacings between the spots in selected area electron diffraction (SAED) images²⁶ (Supplementary Fig. 8). For the as-synthesized Co_3O_4 , ZnCo_2O_4 , and CoGa_2O_4 , the fixed angle of 60° with the crystal spacing of 3.67 nm^{-1} observed between the (220), (422), and (202) planes in their SAED patterns suggests that the [111] facet is the common exposed facet on their octahedrons²⁷ (Figs. 1d and e; Supplementary Figs. 9a and b). Additionally, atom-resolved surface structure characterizations and the lattice spacing measuring further confirm the common exposure of [111] facet for these spinel oxides²⁷ (Fig. 1d; Supplementary Figs. 9c and d).

Supplementary Fig. 8. (a) Schematic illustration of octahedral spinel Co_3O_4 with (111) facets. Atomic structure of [111] surface of the top view (b). Standard diffraction pattern of spinel Co_3O_4 in [111] direction in the reciprocal space (c).

Crystal spacings (norms of the vectors) and angles of the crystal planes measured in the reciprocal space (such as FFT patterns and SAED plots) are of great significance to indexing the diffraction vectors and determining the exposed crystal facet²¹. Typically, three crystal planes parallel to the same line (the normal line perpendicular to the diffraction plane) are required to be indexed. And these three crystal planes need to satisfy the parallelogram rule: $V_{h_1k_1l_1} + V_{h_2k_2l_2} = V_{h_3k_3l_3}$. As shown in Supplementary Fig. 8, the crystal planes determining the [111] facet are $V_{(220)}$, $V_{(202)}$, and $V_{(422)}$, respectively, with the interplanar angle of 60° between the (220) and (202) crystal plane. The interplanar crystal spacing between these crystal planes is 3.67 nm^{-1} .

Supplementary Fig. 9. HRTEM images of ZnCo_2O_4 (a) and CoGa_2O_4 (b) in $[111]$ orientation and their corresponding SAED images and atomic resolved HRTEM images (c-d).

Comment 4: As mentioned by the authors, replacing Co^{2+} by Zn^{2+} and Co^{3+} by Ga^{3+} with larger ionic radii (0.58 vs. 0.60 Å, 0.61 vs. 0.62 Å) gave rise to the increase of interplanar crystal spacing. This phenomenon should be concerned and explained, which may be the key reasons for the different catalytic performances of the three samples.

Response: Thanks very much for your kind comments and pointing out this issue. The claim regarding the “increase in interplanar crystal spacing” was based on the enlarged XRD patterns of the as-synthesized Co-containing spinels at the $[311]$ crystal plane (Supplementary Fig. 2), which does not accurately reflect the variations in interplanar crystal spacing of the primarily exposed $[111]$ facet. In the revise manuscript, we have corrected the enlarged XRD patterns of these as-synthesized Co-

contained spinels to focus on the [111] facet, which is also appended below. The 2θ values for the (111) peaks of Co_3O_4 , ZnCo_2O_4 , and CoGa_2O_4 are identified as 18.94, 19.02, and 18.50 °, respectively. According to Bragg's law, the interplanar spacing for the (111) plane of Co_3O_4 , ZnCo_2O_4 , and CoGa_2O_4 are determined as 4.74, 4.73, and 4.85 Å, respectively. It can be found that substituting Co^{3+} by Ga^{3+} slightly increases the interplanar spacing for the (111) plane, while replacing Co^{2+} by Zn^{2+} trivially decreases the interplanar spacing

Figure R12. XRD patterns (a) and enlarged XRD patterns (b) for Co_3O_4 , ZnCo_2O_4 , and CoGa_2O_4 at [111] facet.

We also applied the following equations to rigorously calculate the interplanar spacing of (111) facet for Co_3O_4 , ZnCo_2O_4 , and CoGa_2O_4 (SN Appl. Sci. 2020, 2, 1-29) ..

$$\begin{aligned}
 |d|^2 &= (\mathbf{h}\mathbf{a}^* + \mathbf{k}\mathbf{b}^* + \mathbf{l}\mathbf{c}^*) \cdot (\mathbf{h}\mathbf{a}^* + \mathbf{k}\mathbf{b}^* + \mathbf{l}\mathbf{c}^*) \\
 &= h^2|\mathbf{a}^*|^2 + k^2|\mathbf{b}^*|^2 + l^2|\mathbf{c}^*|^2 + 2hk|\mathbf{a}^*||\mathbf{b}^*|\cos\gamma^* \\
 &\quad + 2hl|\mathbf{a}^*||\mathbf{c}^*|\cos\beta^* + 2kl|\mathbf{b}^*||\mathbf{c}^*|\cos\alpha^*
 \end{aligned}$$

where h, k, l are the Miller indexes, and $a^*, b^*, c^*, \cos\alpha^*, \cos\beta^*,$ and $\cos\gamma^*$ are constants derived from the unit cell parameters from XRD Rietveld refinement results. The derived interplanar spacings are summarized in the Table R1. Similarly, the interplanar spacings of (111) plane in CoGa_2O_4 was slightly higher than those in Co_3O_4 and ZnCo_2O_4 .

Table R1. Comparison of the interplanar spacing and bond length in Co_3O_4 , ZnCo_2O_4 , and CoGa_2O_4 .

Sample	d (Å, Bragg's law)	d (Å, eq.1)	$\text{M}_{\text{Oh}}\text{-O}$ (Å, DFT)	$\text{M}_{\text{Td}}\text{-O}$ (Å, DFT)
Co_3O_4	4.74	4.72	1.97	1.90
ZnCo_2O_4	4.73	4.70	1.95	1.92
CoGa_2O_4	4.85	4.82	2.01	1.97

In oxygen-involved catalysis, the effects of interplanar crystal spacing of the spinels on the activity might originate from two aspects: (1) variations in bonding behaviors; (2) variations in interatomic distance. Regarding the effects on bonding behaviors, the EXAFS-based DFT modelling results suggest that the variations in interplanar spacings by Ga^{3+} and Zn^{2+} substitutions can be ascribed to the changes in bond lengths between $\text{Zn}_{\text{Td}}\text{-O}$ and $\text{Ga}_{\text{Oh}}\text{-O}$ compared to the pristine $\text{Co}_{\text{Td}}\text{-O}$ and $\text{Co}_{\text{Oh}}\text{-O}$ in the crystal structure of spinels. These variations resulting from Ga^{3+} and Zn^{2+} substitutions led to the covalency competition between the $\text{M}_{\text{Td}}\text{-O}$ (metal-oxygen bond in tetrahedral unit) and $\text{M}_{\text{Oh}}\text{-O}$ (metal-oxygen bond in octahedral unit) in the $\text{M}_{\text{Td}}\text{-O}-\text{M}_{\text{Oh}}$ backbone. Bond breakage occurring at the weaker bond in $\text{M}_{\text{Td}}\text{-O}-\text{M}_{\text{Oh}}$ backbone gives rise to the exposure of the non-oxygen-bonding metal (M-), which is typically considered as the active site in oxygen-involved catalysis. Crystal orbital Hamilton populations (COHPs) analysis revealed that substituting either the $\text{Co}^{2+}_{\text{Td}}$ or $\text{Co}^{3+}_{\text{Oh}}$ site by Zn^{2+} or Ga^{3+} diminished the bonding strength of the $\text{Co}_{\text{Oh}}\text{-O}$ or $\text{Co}_{\text{Td}}\text{-O}$ in ZnCo_2O_4 and CoGa_2O_4 accordingly, and making $\text{Co}^{3+}_{\text{Oh}}$ and $\text{Co}^{2+}_{\text{Td}}$ as the preferentially exposed active sites in ZnCo_2O_4 and CoGa_2O_4 , respectively. D-band center analysis further suggested that the greater spin polarization effect on $\text{Co}^{2+}_{\text{Td}}$ sites induced by the Ga substitution regulated the energy levels for molecular orbitals interactions and accelerated the electronic communications with the adsorbed O_3 , thereby enhancing the intrinsic activity of $\text{Co}^{2+}_{\text{Td}}$ sites.

Regarding the interatomic distance effects, increased interplanar spacing can create a larger reaction

space around $\text{Co}^{2+}_{\text{Td}}$ and facilitate the interactions with O_3 molecules. Noteworthy that O_3 molecules can be adsorbed onto the active sites via two possible models (i.e., bridge and Pauling). For Co_3O_4 , ZnCo_2O_4 , and CoGa_2O_4 , the distance of the adjacent two metal atoms is at least 5.3 Å, however, the total length of an O_3 molecule is 1.41 Å. This makes the O_3 adsorption via the bridge mode impossible. Our *in situ* Raman results further verified that only the Pauling type adsorption mode of O_3 molecules existed for the spinels (Response to Comment 1 from Reviewer #2). Therefore, altering the interplanar crystal spacing results in a negligible influence on reaction thermodynamics. Based on the results above, we posit that the influence of interplanar spacing differences induced by Zn^{2+} and Ga^{3+} substitution on catalytic ozonation activity is primarily due to the changes in bonding behavior within the crystal structure of spinels and the modulation of electronic properties. In the revised manuscript, we modified the corresponding discussion and added more discussion on the influence of alternations in interplanar crystal spacing on catalytic activity.

Change:

Page 6, Manuscript

Although $\text{Zn}^{2+}/\text{Ga}^{3+}$ with larger ionic radii than $\text{Co}^{2+}/\text{Co}^{3+}$ (0.58 vs. 0.60 Å, 0.61 vs. 0.62 Å) slightly altered the interplanar crystal spacing within the spinel structure, the corresponding Rietveld refinement results validate the pure single normal spinel crystal phase of the as-synthesized Co_3O_4 and its substitutes (Supplementary Table 1). The increase in the interplanar crystal spacing can result in the covalency competition between the $\text{M}_{\text{Td}}\text{-O}$ (metal-oxygen bond in tetrahedral unit) and $\text{M}_{\text{Oh}}\text{-O}$ (metal-oxygen bond in octahedral unit) in the $\text{M}_{\text{Td}}\text{-O-M}_{\text{Oh}}$ backbone of the spinel crystal structure, which affects the exposed active sites and the electronic properties that govern the catalytic activity²³.

Page 23, Supplementary Information

Supplementary Fig. 2. XRD patterns (a) and enlarged XRD patterns (b) for Co_3O_4 , ZnCo_2O_4 , CoAl_2O_4 , and CoGa_2O_4 .

For pristine Co_3O_4 , the diffraction peaks located at 19° , 31° , 37° , 45° , 59° , and 65° are attributed to the (111), (220), (311), (400), (511), and (440) planes of its cubic spinel phase, respectively. ZnCo_2O_4 and CoGa_2O_4 obtained similar diffraction patterns to pristine Co_3O_4 , with peaks slightly shifted to lower positions, suggesting that they shared the same cubic spinel crystal phase.

According to Bragg's law, the interplanar spacing for the (111) plane of Co_3O_4 , ZnCo_2O_4 , and CoGa_2O_4 are determined as 4.74, 4.73, and 4.85 Å, respectively. It can be found that substituting Co^{3+} by Ga^{3+} slightly increased the interplanar spacing for the (111) plane, while replacing Co^{2+} by Zn^{2+} trivially decrease the interplanar spacing

We also applied the following equations to rigorously calculate the interplanar spacing of (111) plane for Co_3O_4 , ZnCo_2O_4 , and CoGa_2O_4 ¹⁵.

$$\begin{aligned}
 |d|^2 &= (\mathbf{h}\mathbf{a}^* + \mathbf{k}\mathbf{b}^* + \mathbf{l}\mathbf{c}^*) \cdot (\mathbf{h}\mathbf{a}^* + \mathbf{k}\mathbf{b}^* + \mathbf{l}\mathbf{c}^*) \\
 &= h^2|\mathbf{a}^*|^2 + k^2|\mathbf{b}^*|^2 + l^2|\mathbf{c}^*|^2 + 2hk|\mathbf{a}^*||\mathbf{b}^*|\cos\gamma^* \\
 &\quad + 2hl|\mathbf{a}^*||\mathbf{c}^*|\cos\beta^* + 2kl|\mathbf{b}^*||\mathbf{c}^*|\cos\alpha^*
 \end{aligned}$$

where h, k, l are the Miller indexes, and $a^*, b^*, c^*, \cos\alpha^*, \cos\beta^*,$ and $\cos\gamma^*$ are constants derived from the unit cell parameters from XRD Rietveld refinement results. The derived interplanar spacings are summarized in the **Supplementary Table S1**. Similarly, the interplanar spacings of (111) plane in CoGa_2O_4 was slightly higher than those in Co_3O_4 and ZnCo_2O_4 .

Supplementary Table 1. XRD Rietveld refinement results and comparisons of the interplanar spacing and bond length in Co_3O_4 , ZnCo_2O_4 , and CoGa_2O_4 .

Sample	Rietveld refinement results (Å)			d (Å, Bragg's law)	d (Å, interplanar spacing equation)
	a	b	c		
Co_3O_4	8.10	8.10	8.10	4.74	4.72
ZnCo_2O_4	8.08	8.08	8.08	4.73	4.70
CoGa_2O_4	8.33	8.33	8.33	4.85	4.82

Comment 5. Does the directional substitution of Zn^{2+} and Ga^{3+} have an impact on the local electronic structure of $\text{Co}^{3+}/\text{Co}^{2+}$ in other positions? Whether this effect plays a role in catalysis.

Response: Thanks for your comments. Based on our characterization results, we believe that Zn^{2+} and Ga^{3+} can only replace the Co^{2+} site and Co^{3+} sites, respectively. The followings are the detailed explanations.

Figure R13 (a) Schematic illustration of the occupation of Co sites based on crystal field theory. (b) High resolution XPS spectrum on Zn 2p. (c) High resolution XPS spectrum on Ga 2p. (d) Normalized Ga K-edge XANES spectrum of the as-synthesized CoGa_2O_4 . (e) The corresponding Fourier transformed EXAFS (FT-EXAFS) spectra at R space.

According to the crystal field theory, the Co^{3+} is octahedrally coordinated, while Co^{2+} is tetrahedrally coordinated in a normal spinel oxide of Co_3O_4 (Fig. R13a) (Adv. Mater. 2019, 31, 1902509). XRD patterns and the corresponding Rietveld refinement results suggested that all as-prepared Co_3O_4 , ZnCo_2O_4 , and CoGa_2O_4 belonged to normal spinel oxides. Our XAS analysis results (both Co K-edge XANES spectra and the corresponding fitted EXAFS results) further revealed that $\text{Co}^{2+}_{\text{Td}}$ and $\text{Co}^{3+}_{\text{Oh}}$ sites in ZnCo_2O_4 and CoGa_2O_4 were successfully substituted by Zn^{2+} and Ga^{3+} , respectively. Moreover, the unsubstituted $\text{Co}^{2+}_{\text{Td}}$ and $\text{Co}^{3+}_{\text{Oh}}$ sites remained unchanged. Additionally, given that the 3d orbitals for both Zn and Ga are fully occupied with electrons, only Zn^{2+} and Ga^{3+} can be formed when they are incorporated in the Co_3O_4 crystal scaffolds. The high resolution XPS spectra on Zn 2p and Ga 2p

suggest the formation of Zn^{2+} and Ga^{3+} (Figs. R13b-c). These results suggest that substituting the $\text{Co}^{2+}_{\text{Td}}$ and $\text{Co}^{3+}_{\text{Oh}}$ sites by Zn^{2+} and Ga^{3+} did not alter the normal spinel structure and that $\text{Co}^{2+}_{\text{Td}}$ and $\text{Co}^{3+}_{\text{Oh}}$ were the only forms of the Co^{2+} and Co^{3+} in CoGa_2O_4 and ZnCo_2O_4 , respectively.

To further reinforce this viewpoint, we acquired Ga K-edge XANES spectrum to elucidate its coordination environments (Figs. R13d-e) in the revised manuscript. Not surprisingly, Ga^{3+} substituted the Co^{3+} position and exhibited two bonding modes with the surrounding octahedral ($\text{Ga}^{3+}_{\text{Oh}}-\text{Ga}^{3+}_{\text{Oh}}$, ~ 2.5 Å) and tetrahedral sites ($\text{Ga}^{3+}_{\text{Oh}}-\text{Co}^{2+}_{\text{Td}}$, ~ 3.0 Å). The identification of these bonding modes verifies the positioning of Ga^{3+} at octahedral sites. In the revised manuscript, we have modified the corresponding discussion on confirming the $\text{Co}^{2+}_{\text{Td}}$ and $\text{Co}^{3+}_{\text{Oh}}$ sites are the only forms of the Co^{2+} and Co^{3+} in CoGa_2O_4 and ZnCo_2O_4 , respectively

Change:

Page 10, Manuscript

The coordination environments surrounding the substituted Zn and Ga were also investigated. The fully occupied $3d$ orbitals of Zn and Ga restrict their valence states to +2 and +3 accordingly when incorporated into the Co_3O_4 crystal scaffolds, as revealed by their high resolution XPS spectra (Supplementary Fig. 13). Additionally, the acquired Ga K-edge XANES spectrum and the corresponding fitted EXAFS results indicate that Ga^{3+} substituted the Co^{3+} position and exhibited both octahedral and tetrahedral bonding modes (Supplementary Fig. 14; Supplementary Table 4), similar to those of Co^{3+} in ZnCo_2O_4 . These results suggest that Zn^{2+} and Ga^{3+} substitution did not alter the spinel scaffolds and that $\text{Co}^{2+}_{\text{Td}}$ and $\text{Co}^{3+}_{\text{Oh}}$ were the only forms of the Co^{2+} and Co^{3+} in CoGa_2O_4 and ZnCo_2O_4 , respectively.

Supplementary Fig. 13. High resolution XPS survey on Zn 2p of ZnCo₂O₄ (a) and Ga 2p of CoGa₂O₄

(b).

Supplementary Fig. 14. Ga K-edge XANES spectrum (a) and the corresponding fitted EXAFS results

at R space (b) and at k space (c).

Supplementary Table 4. The EXAFS fitting parameters at the Ga K-edge for CoGa₂O₄.

Sample	Shell	N	R(Å)	$\sigma^2(\text{Å}^2)$	$\Delta E_0(\text{eV})$	S_0^2	R factor
CoGa ₂ O ₄	Ga-O	6	1.95	0.0098(1)	4.959	0.83	0.023
	Ga-Ga	6	2.96	0.0066(1)			
	Ga-Co	6	3.46	0.0029			
	Ga-Ga	6	3.57	0.0018(1)			

Comment 6. It is unfavorable to conclude that the Co₃O₄, ZnCo₂O₄ and CoGa₂O₄ displayed different geometric locations of Co sites just based on the XRD and HAADF-STEM image. The authors should provide more atomic structure information to identify the location of Zn and Ga species.

Response: Thanks for the comments. We conducted Co K-edge XANES analyses on Co₃O₄, ZnCo₂O₄, CoAl₂O₄, and CoGa₂O₄ to elucidate variations in local coordination environment (Figs. 1c and d). Our findings indicated that Co³⁺_{Oh} and Co²⁺_{Td} were the predominant geometric sites in ZnCo₂O₄ and CoAl₂O₄/CoGa₂O₄, respectively. Additionally, theoretical calculations using COHP analysis were performed to demonstrate the alteration of favorable exposed geometrical sites by Ga³⁺/Zn²⁺ substitution (Supplementary Fig. 29). Moreover, the Ga K-edge XANES spectrum was also recorded to further analyze the corresponding atomic structure information, following your insightful suggestion. Please refer to Response to Comment 5 for detailed information.

Comment 7. In the theoretical calculations section, the authors use density flooding theory for their

calculations. The transition metals are a strongly correlated system of d electrons, so the interaction between electrons needs to be considered, and whether the absence of this critical factor has an impact on the catalytic performance and electronic structure, the authors apparently lack this critical evidence.

Response: Thanks for the comments. In this manuscript, we addressed the significance of the d band electrons and their electronic interactions with the O 2p electrons. Firstly, we applied the molecular orbital theory, which was based on the band center analysis of the 3d orbitals of the transition metals and the O 2p orbital, to identify the critical molecular orbitals governing the activity (Figs. 4 a and b; Supplementary Figs. 30 and 31). We concluded that the occupancy of MO*s was crucial to determine the electron transfer resistance and hybridization strength. Moreover, we also performed bond order analysis of ozone adsorption on $\text{Co}^{2+}_{\text{Td}}$ sites and $\text{Co}^{3+}_{\text{Oh}}$ sites (Supplementary Fig. 36). We found that the greater hybridization strength at the $\text{Co}^{2+}_{\text{Td}}$ sites than the $\text{Co}^{3+}_{\text{Oh}}$ sites resulted in a higher bond order between $\text{Co}^{2+}_{\text{Td}}$ 3d orbital and O 2p orbital than that between $\text{Co}^{3+}_{\text{Oh}}$ and O 2p orbital (2 vs. 1.5). As a result, the electronic communications at the $\text{Co}^{2+}_{\text{Td}}$ sites was promoted, accounting for its high catalytic activity.

In the revised manuscript, we modified the corresponding discussion to address the incorporation of the Co 3d electrons throughout the DFT analysis.

Change:

Page 18, Manuscript

To illustrate the intrinsic electronic states of the spinel oxides and hybridization strength between Co 3d orbitals and O 2p orbitals from O_3 molecules and their influences on activity, we constructed Co_3O_4 , CoGa_2O_4 , and ZnCo_2O_4 models with the same exposure of [111] facet, based on the deciphered results from XAS and superconducting quantum interference device (SQUID) magnetization analysis, for use

in the density functional theory (DFT) calculations (Supplementary Figs. 31 and 32; Supplementary Table 11). As revealed in the crystal orbital Hamilton population (COHP) analysis, the strong covalency competition arising from the substitution of Ga^{3+} or Zn^{2+} into crystal structure favors the exposure of the unsubstituted cations $\text{Co}^{2+}_{\text{Td}}$ and $\text{Co}^{3+}_{\text{Oh}}$ as the preferentially exposed active sites in CoGa_2O_4 and ZnCo_2O_4 ¹⁴ (Supplementary Fig. 33). The calculation outcomes well match the XAS analysis results, supporting the validity of the constructed models. Given the electron-rich nature of O_3 molecules, all molecular orbitals (MOs) and partial of anti-bonding molecular orbitals (MO*s) can be occupied when O_3 hybridizes on either $\text{Co}^{2+}_{\text{Td}}$ or $\text{Co}^{3+}_{\text{Oh}}$ sites (Supplementary Fig. 34), making the occupancy of MO*s crucial to determine the electron transfer resistance and hybridization strength³⁸. Projected electronic density of states (PDOS) profiles on Co $3d$ orbitals and the corresponding d -band center analysis reveal that the high spin polarization resulting from the covalency competition positions the band centers of spin-down channels ($\epsilon_{d\downarrow}$) of the spinel oxides at higher energy levels than those of spin-up channels ($\epsilon_{d\uparrow}$), endowing spin-down channels a decisive role in communicating with the O_3 $2p$ orbitals for MO*s formation (Supplementary Fig. 35). Based on the orbital analysis results, orbital overlapping diagrams are plotted (Figs. 4a and b). MO*s formed by hybridization of spin-down channel of $\text{Co}^{2+}_{\text{Td}}$ sites and O_3 $2p$ orbitals (MO*s- $\text{Co}^{2+}_{\text{Td}}$) are situated closer to the Fermi level than those formed by $\text{Co}^{3+}_{\text{Oh}}$ sites. This facilitates the electron transition at $\text{Co}^{2+}_{\text{Td}}$ sites and decreased the energy barriers for O_3 dissociation, which promotes the production of reactive intermediates³⁹.

Comment 8. The computational models should be described in detail: lattice parameters, periodic unit size, layers of material, vacuum slab size, modeled crystallographic plane, geometric detail of the complex $\text{Co}^{2+}_{\text{Td}}/\text{Co}^{3+}_{\text{Oh}}$, total charge....

Response: Thanks for your suggestions. The detailed description of the computational models has

been provided in the revised manuscript.

Change:

Supplementary Table 13. Details of DFT models.

Model- exposed site	Periodic unit size	Layers of model	Vacuum slab size	Crystallographic plane	Total charge	Magmom
Co ₃ O ₄ - Co ²⁺	a = 9.804 Å, b = 9.804 Å, c = 8.35 Å, α = 60°, β = 120°, and γ = 90°.	2	10 Å	(111)	612	Co ²⁺ _{Td=3} , Co ³⁺ _{Oh=2}
Co ₃ O ₄ - Co ³⁺	a = 9.805 Å, b = 9.805 Å, c = 8.345 Å, α = 60°, β = 120°, and γ = 90°.	2	10 Å	(111)	612	Co ²⁺ _{Td=3} , Co ³⁺ _{Oh=2}
ZnCo ₂ O ₄ - Co ³⁺	a = 9.92 Å, b = 9.92 Å, c = 8.426 Å, α = 60°, β = 120°, and γ = 90°.	2	10 Å	(111)	648	Zn ²⁺ _{Td=0} , Co ³⁺ _{Oh=2}
CoGa ₂ O ₄ - Co ²⁺	a = 10.292 Å, b = 10.292 Å, c = 8.937 Å, α = 60°, β = 120°, and γ = 90°.	2	10 Å	(111)	468	Co ²⁺ _{Td=3} , Ga ³⁺ _{Oh=0}

Comment 9. All calculated energies displayed in figures should be tabulated.

Response: Thanks for your kind suggestion. The calculated energies displayed in figures have been

provided and tabulated in the revised Supplementary Information.

Change:

Supplementary Table 14. Computed energy releases in different transition states.

Configuration	O ₃ adsorption	TS ₁	O ₃ dissociation	H ₂ O adsorption	TS ₂	H ₂ O dissociation	•OH desorption
Co ₃ O ₄ -Co ²⁺	0.41 eV	-0.6 eV	2.32 eV	0.35 eV	-1.51 eV	-0.53 eV	-0.44 eV
Co ₃ O ₄ -Co ³⁺	0.32 eV	-0.75 eV	2.2 eV	0.24 eV	-2.36 eV	-0.42 eV	0.77 eV
ZnCo ₂ O ₄ -Co ³⁺	1.27 eV	-1.15 eV	3.05 eV	0.38 eV	-3.69 eV	-0.48 eV	0.62 eV
CoGa ₂ O ₄ -Co ²⁺	1.54 eV	-1.01 eV	4.36 eV	0.22 eV	-3.23 eV	-0.91 eV	-0.97 eV

Comment 10. Please check the "References" format carefully, there are a few minor errors, sun as:

Reference 10, 14, 20, 44.

Response: Thanks for your kind comments. The formatting errors in the References have been corrected in the revised manuscript.

Response to Reviewers' Comments

Ms. Ref. No.: NCOMMS-24-42294A-Z

Title: Tailored ozone activation on geometrical-site-dependent cobalt with selective coordination

Reviewer #2

Comment: The author has made the necessary revisions according to my suggestions and has successfully addressed all the concerns I raised. The revised version significantly improves the quality of the paper, and I believe all issues have been properly resolved. Therefore, I support the publication of this paper.

Response: Thank you very much for your high recognition of this study. The improvement of the overall quality of this manuscript solidly relies on your valuable and kind comments.

Reviewer #3

Comment: The manuscript has been revised according to the comments and all my concerns have been suitably addressed. I believe this manuscript now is ready for publication.

Response: We appreciate your efforts in reviewing our manuscript and your high recognition of this substantial revision.

Reviewer #4

Comment: The authors fabricated a series of Co-based spinel oxides with controlled crystal facets/sizes to investigate their geometrical site-dependent catalytic activity for ozone activation. The highly spin-polarized Co^{2+} at tetrahedral sites promoted the formation of surface-bound hydroxyl radicals, resulting in enhanced water decontamination efficiency. However, the concerns raised by Reviewer #1 have not been fully addressed in the revised manuscript. As pointed out by Reviewer #1, the 3DOM- Co_3O_4 , with a larger specific surface area than CoGa_2O_4 , exhibits higher catalytic activity for ozone activation (Figure R2). Thus, it is still unclear whether the activity difference stems from the coordination geometry of the Co sites. In addition, the Co^{2+} and Co^{3+} species in Co-based spinel structures are usually found in both tetrahedral and octahedral sites. The Co^{2+} species with stronger electron-donating ability than Co^{3+} , is well-known to facilitate $\cdot\text{OH}$ generation at both Td and Oh sites, making the claim of negligible Co valence effects between $\text{Co}^{2+}_{\text{Td}}$ and $\text{Co}^{3+}_{\text{Oh}}$ inaccurate. Furthermore, while the authors suggest previous research overlooked the uniformity of exposed crystal planes, the high activity of CoGa_2O_4 may also stem from the exposure of [111] crystal facets. Comparative materials with different crystal planes are lacking due to limitations in the synthesis method. Therefore, I do not recommend this work for publication in Nature Communications.

Response: Thanks very much for providing a new perspective to further validate the scientific findings in solid mechanistic understanding on the relationships between coordination geometry of the Co sites and the reactivities of the Co-based spinel oxides. The followings are our detailed responses to the raised concerns.

The first concern still relates to the influence of surface area on intrinsic activity, which was raised

by Reviewer #1. In *the last revision*, we addressed this concern from the following points.

(1) First, we re-affirmed the research objectives of this study. Our study's focus extends beyond merely increasing the catalytic ozonation activity of Co_3O_4 for oxalic acid decomposition. The objectives of this study were to unveil the intrinsic active sites in cobalt-based spinel oxides and to address spin-polarization effects from geometrical site competition on reaction thermodynamics in transition metal oxide-induced catalysis, which resulted in a high selectivity for production of reactive oxygen species. We believe that the elucidation of intrinsic insights into the geometrical site engineering offers regulating protocols of the reaction thermodynamics in transition metal oxide-induced catalysis and envisages the practical application potentials of engineered spinel oxides.

Figure R1. SEM images of 3DOM Co_3O_4 (a), plate Co_3O_4 (b), and spherical Co_3O_4 (c). (d) N_2 sorption isotherms of synthesized Co_3O_4 . (e) Catalytic ozonation for OA degradation with synthesized Co_3O_4 . (f) Reaction rates and BET surface area normalized reaction rates of $\text{Co}_3\text{O}_4/\text{O}_3$ systems. (g) Relationship between the SSAs of the Co_3O_4 reported in the literature and their activity efficiency. Noted that the activity efficiency is derived by the follow equation. (h) Cyclic experiments for synthesized Co_3O_4 and cobalt leaching concentration. (i) Comparison of catalyst fabrication costs and electrical energy per order (EE/O) of different reaction systems.

(2) To clarify the influence of surface area on intrinsic activity, we synthesized Co_3O_4 with significantly higher SSAs (i.e., 3DOM Co_3O_4 with SSA of $28.4 \text{ m}^2/\text{g}$, plate Co_3O_4 with SSA of $17.5 \text{ m}^2/\text{g}$, and spherical Co_3O_4 with SSA of $78.4 \text{ m}^2/\text{g}$), Fig. R1a-d) compared to the coordination geometry regulated Co-based spinels ($\sim 0.3 \text{ m}^2/\text{g}$) and evaluated their catalytic ozonation activity. In these additionally synthesized high-SSA Co_3O_4 samples, 3DOM- Co_3O_4 exhibited a slightly higher catalytic ozonation activity than CoGa_2O_4 (Fig. R1e). However, the specific surface area normalized reaction rate of 3DOM- $\text{Co}_3\text{O}_4/\text{O}_3$ system was actually 59-fold less than that of $\text{CoGa}_2\text{O}_4/\text{O}_3$ system (Fig. R1f), suggesting that the CoGa_2O_4 , featuring a finely tuned coordination geometry dominated by spin-polarized Co^{2+} sites, achieves a substantially higher *intrinsic activity* than Co_3O_4 with large specific surface areas and non-regulated coordination geometry. This result clearly demonstrates that the difference in activity originates from the coordination geometry of the Co sites.

(3) Our literature review further suggests that simply increasing the SSA of Co_3O_4 does not significantly enhance its catalytic activity beyond a certain threshold. While increasing the SSA

facilitates the exposure of the active sites, the thermodynamic-governed intrinsic reactivity of these sites remains unchanged (Fig. R1g). To enhance the intrinsic activity of the active sites, fine-manipulations over crystal structures (such as regulation of geometrical coordinations and spin state) and electronic interactions (such as introducing support-metal interactions) are required. Furthermore, enhanced exposure of the active sites may also induce side reactions and adversely affect the catalytic performance. Additionally, some Co_3O_4 samples are not catalytic active even with high SSAs and the activity of as-synthesized catalysts in this study, despite with small SSAs, are among the highest reported. The above literature review results indicate that the strategies for regulating the active sites are critical in determining the intrinsic activity of Co_3O_4 rather than their SSAs.

(4) To address the practical concerns on coordination geometry regulation in enhancing the activity, we evaluated the recyclability of these synthesized Co_3O_4 with high SSAs by a multi-cycle reusability test (Fig. R1h). Clearly, Co_3O_4 catalysts with high SSAs demonstrated much inferior recyclability and less structural stability compared to CoGa_2O_4 with selectively exposed $\text{Co}^{2+}_{\text{Td}}$ sites owing to the strong bonding strength in CoGa_2O_4 crystal structure. Then, we conducted economic evaluations on the coordination-geometry-regulated Co-based spinels and Co_3O_4 samples with high SSAs. Both the comparisons of fabrication costs and the electrical energy per order (EE/O) suggested the economic feasibility of the activity enhancement via optimizing the coordination environments around the active sites, envisaging its practical application possibilities (Fig. R1i).

Based on above analysis, we believed that the concerns from *Reviewer#1* had been adequately

addressed from both experimental and practical views, reinforcing the pivotal role of regulating coordination geometry in enhancing the intrinsic activity and the practical application possibilities in Co-based spinel catalysts.

In this revision, considering the electron transfer nature of the heterogeneous O₃ activation process, we further measured the electrochemical active surface areas (ECSAs) of the as-synthesized catalysts to examine the number of the exposed active sites during the reaction. The intrinsic activity of the active sites with different coordination geometries were further compared by ECSAs normalized reaction kinetics (k_{ECSAs}), which served as a strong complementary of SSAs normalized reaction kinetics (k_{normS}) to exclude the influence of exposed active sites.

Figure R2. Cyclic Voltammetry (CV) curves of Co_3O_4 (a), ZnCo_2O_4 (b), CoGa_2O_4 (c), spherical Co_3O_4 (d), 3DOM Co_3O_4 (e), and plate-like Co_3O_4 (f) at different scan rates, and ECSA measurements of samples (g) and (h). (i) Comparison of the BET surface area normalized reaction rates (k_{norms}) and ECSA normalized reaction rates (k_{ECSAS}) of different spinels/ O_3 systems. The electrolyte contained $100 \text{ mg L}^{-1} \text{ Na}_2\text{SO}_4$ (pH=3) and the 0.1 mg cm^{-2} catalyst was loaded on glassy carbon-rotating disk electrode (GC-RDE).

As shown in Fig. R2a-h, by fitting the non-Faradaic currents from cyclic voltammetry measurements at different scan rates, the ECSAs of Co_3O_4 , CoGa_2O_4 , ZnCo_2O_4 , and 3DOM Co_3O_4 are derived as 0.50, 0.30, 0.44, and 3.3 mF cm^{-2} , respectively. This suggests that 3DOM Co_3O_4 exposes the highest number of active sites during the heterogeneous reactions owing to its periodic structure, which agrees with its highest BET SSA. The k_{ECSAS} can provide a more accurate representation of the intrinsic activity of the exposed active sites than reaction kinetics normalized by BET SSAs (k_{norms}). It is found that the exposed Co sites in CoGa_2O_4 obtained an 8.7-fold higher intrinsic activity than those in 3DOM Co_3O_4 (Fig. R2i). Additionally, the intrinsic activity of the Co sites in CoGa_2O_4 , Co_3O_4 , and ZnCo_2O_4 aligns well with the trends observed in the k_{norms} comparison. These results collectively reinforce the notion that regulating coordination geometry in Co-based spinel outperforms SSA engineering in determining the intrinsic activities of the active sites.

Regarding the second concern, we appreciate the reviewer for providing new perspectives on coordination-geometry-competition of Co sites within Co-based spinels and the Co valence effects, which further validate the geometry-dependent-activity of the Co sites. And it is critical to investigate

the potential coordination-geometry-competition of Co sites in the as-synthesized Co-based spinel and evaluate the catalytic activity of the possibly exist $\text{Co}^{2+}_{\text{Oh}}$ sites.

[Figure Redacted]

Figure R3. (a) Crystal structure of spinel. (b) Empirical octahedral site preference energy (OSPE) of transition metal ions (Adapted from *Science* **1964**, 144, 1001-1003).

The allocation of the geometric coordination for metal cations within the $\text{A}^{2+}\text{B}^{3+}_2\text{O}_4$ spinel oxide structure depends on their octahedral site preference energy (OSPE) (Fig. R3) (*Science* **1964**, 144, 1001-1003). Coordination geometry competition exists when the OSPE of the metal cations are similar, while a significant difference in OSPE decreases this competition, resulting in more definitive coordination geometries for both metal cations (*Adv. Funct. Mater.* **2023**, 33, 2214275). For the as-synthesized Co_3O_4 in this study, Co^{3+} and Co^{2+} preferentially occupy the octahedral and tetrahedral sites, respectively, because of the significantly higher OSPE of Co^{3+} than that of Co^{2+} (19.2 vs. 2.1 kcal/mol, Fig. R3b) (*Nat. Commun.* **2021**, 12, 2608, *J. Am. Chem. Soc.* **2023**, 145, 18992-19004). Ga^{3+} and Zn^{2+} , which have fully occupied 3d orbitals, exhibit not only catalytic inactivity but also a zero OSPE. This excludes the coordination geometry competition in CoGa_2O_4 and ZnCo_2O_4 and enabling $\text{Co}^{2+}_{\text{Td}}$ and $\text{Co}^{3+}_{\text{Oh}}$ to serve as the dominant active sites, respectively.

Our literature review results further reinforce the predominant role of octahedral site preference energy

in determining the allocation of the geometric coordination for the metal ions in spinel structures (Table R1). We also find that most of the reported Co_3O_4 , CoGa_2O_4 , and ZnCo_2O_4 exhibited a normal spinel structure. Noted that, under some specific synthesis conditions, such as extremely high pressures (*J. Phys.: Condens. Matter* **2012**, 24, 435401) and significantly structural defects (*J. Phys. Chem. Lett.* **2022**, 13, 35, 8386–8396), the structure of the Co-based spinels can become inverted.

Figure R4. (a) Rietveld-refined XRD patterns of Co_3O_4 , ZnCo_2O_4 and CoGa_2O_4 . (b) FT-EXAFS spectra at the Co K-edge XANES spectra of spinel oxides. (c) Raman spectra of spinel oxides.

Beyond the considerations from OSPE, our experimental results further revealed $\text{Co}^{2+}_{\text{Td}}$ and $\text{Co}^{3+}_{\text{Oh}}$ as the dominant active sites rather than the structurally inverted ones. *Firstly*, XRD patterns and the corresponding Rietveld refinement results suggested that all as-prepared Co_3O_4 , CoGa_2O_4 , and ZnCo_2O_4 belonged to the normal spinel oxides (Fig. R4a). *Secondly*, Fourier-transform extended X-ray absorption fine structure (FT-EXAFS) spectra indicated that the peak intensity ratio of $\text{Co}_{\text{Oh}}-\text{Co}_{\text{Oh}}$ path (at 2.5 \AA) to $\text{Co}_{\text{Td}}-\text{Co}_{\text{Oh}}$ path (at 3.0 \AA) for Co_3O_4 and CoGa_2O_4 were 1.39 and 0.79, respectively (Fig. R4b). The significant decrease of the peak intensity ratio suggests that most of the Co octahedral sites have been replaced by Ga^{3+} cations, while the considerably decreased $\text{Co}_{\text{Oh}}-\text{Co}_{\text{Oh}}$ path for CoGa_2O_4 might originate from minor presence of $\text{Co}^{2+}_{\text{Oh}}$ sites and the interference signal of Ga d orbital electrons. To investigate the influence of this interference signal from Ga, we recorded the FT-

EXAFS spectrum of CoAl_2O_4 , which was synthesized using the same protocol as these Co-based spinels, and compared it to that of CoGa_2O_4 . Evidently, the absence of d orbital in Al^{3+} diminished the scattering path at $\sim 2.5 \text{ \AA}$. This suggests $\text{Co}^{2+}_{\text{Oh}}$ sites as the minor coordination-geometry-competitor against the dominant $\text{Co}^{2+}_{\text{Td}}$ sites.

Thirdly, Raman spectra (Fig. R4c) indicate that Co_3O_4 displays five Raman peaks: F_{2g}^1 (194 cm^{-1}), E_g (480 cm^{-1}), F_{2g}^2 (521 cm^{-1}), F_{2g}^3 (615 cm^{-1}), and A_{1g} (688 cm^{-1}). Specifically, the A_{1g} and F_{2g}^1 peaks correspond to signals from octahedral and tetrahedra sites, respectively, while E_g , F_{2g}^2 , and F_{2g}^3 are signals of octahedral and/or tetrahedra sites without clear identification (*J. Am. Chem. Soc.* **2024**, 146, 2967-2976). Substituting Co^{3+} by Ga^{3+} in CoGa_2O_4 blue shifted the A_{1g} symmetry, while the peak position of F_{2g}^1 symmetry remained unchanged. The variation of the cation–anion bond length at the octahedral sites and polyhedral distortion occurring in the spinel lattice after Ga^{3+} replacement accounts for the change of A_{1g} symmetry in CoGa_2O_4 . The unchanged coordinative environment of $\text{Co}^{2+}_{\text{Td}}$ sites maintained the F_{2g}^1 symmetry. Similarly, the shift of F_{2g}^1 symmetry peak and the maintaining of A_{1g} symmetry in ZnCo_2O_4 suggest Zn^{2+} occupied the tetrahedral sites while the $\text{Co}^{3+}_{\text{Oh}}$ sites remained unchanged. The above Raman results further solidify the normal spinel structures of the as-synthesized Co-based spinels and the minor presence of inversed $\text{Co}^{2+}_{\text{Oh}}/\text{Co}^{3+}_{\text{Td}}$ sites.

Figure R5. Schematic illustration of the influence of coordination geometry on allocation of the frontier electrons for $\text{Co}^{2+}_{\text{Td}}$, $\text{Co}^{2+}_{\text{Oh}}$, and $\text{Co}^{3+}_{\text{Oh}}$.

For the valence effect, we agree with the Reviewer that Co^{2+} species obtain a higher catalytic activity than Co^{3+} sites due to their higher tendency to donate the frontier electrons. And this effect is determined by the geometry coordination of the Co species within the spinel structure. As schematically illustrated in Fig. R5, the geometric (tetrahedral/octahedral) coordination rules the distribution and allocation of the frontier electrons of the $\text{Co}^{2+}/\text{Co}^{3+}$ species participating the redox reactions according to the crystal field theory. Co^{2+} at both tetrahedral and octahedral sites obtains a higher number of frontier electrons than the Co^{3+} sites.

Figure R6. (a) High resolution XPS surveys on Co 2p for CoO and Co₃O₄. (b) Rietveld-refined XRD patterns of CoO. (c) N₂ sorption isotherms of synthesized CoO. (d) ECSA measurements of CoO with inset of the corresponding CV curves with different scanning rates. (e) Catalytic ozonation activities of CoO and spinel oxides. (f) Comparison of k_{ECSAS} and k_{normS} of CoO and spinel oxides.

To differentiate the intrinsic catalytic activity between $\text{Co}^{2+}_{\text{Td}}$ sites with $\text{Co}^{2+}_{\text{Oh}}$ sites, we synthesized

CoO with $\text{Co}^{2+}_{\text{Oh}}$ as the predominated Co sites according to the previous reported method (*ACS Catal.* **2017**, 7, 1626-1636). XPS survey confirms the absence of Co^{3+} (Fig. R6a), while XRD Rietveld refinement result suggests the successful preparation of the CoO with $\text{Co}^{2+}_{\text{Oh}}$ as the single crystal phase (Fig. R6b). We evaluated the catalytic ozonation activity of the as-synthesized CoO and normalized it by its BET SSA (Fig. R6c) and ECSA (Fig. R6d) to obtain the intrinsic activity. It is found that the intrinsic activity (k_{ECSA}) of $\text{Co}^{2+}_{\text{Oh}}$ sites is 1.5-fold higher than that of $\text{Co}^{3+}_{\text{Oh}}$, yet 10.1-fold lower than $\text{Co}^{2+}_{\text{Td}}$ (Figs. R6e and f). Similar trend is also observed in comparison of k_{normS} . This can be ascribed to the inefficient electron transfer ability of octahedrally coordinated O atoms compared to tetrahedrally coordinated O atoms in $\text{Co}^{2+}_{\text{Td}}$ sites.

In light of the above analysis regarding the structural compositions of $\text{Co}^{2+}_{\text{Td}}/\text{Co}^{2+}_{\text{Oh}}$ sites and the intrinsic activity of the $\text{Co}^{2+}_{\text{Oh}}$, we conclude that the normal spinel structure is the dominate phase in the as-synthesized Co-based spinels, and $\text{Co}^{2+}_{\text{Td}}$ sites still act as the predominant active sites for O_3 activation. In the revised manuscript, we have added the corresponding discussions on the in-depth evaluation of the coordination-geometry-competition of Co sites in the as-synthesized Co-based spinel and the importance of the valence effect of the Co in different coordination geometries on catalytic activity. Moreover, an examination of the catalytic activity of the possibly existing $\text{Co}^{2+}_{\text{Oh}}$ sites have also been included in the revised manuscript.

Table R1. Co-based spinel oxides and Co sites.

	Spinel oxides and Co site			Characterization methods	Ref.
Catalyst	Co_3O_4	CoAl_2O_4	ZnCo_2O_4	XRD, XPS,	J. Am. Chem. Soc. 2016, 138, 36-39
Geometric site	$\text{Co}^{2+}_{\text{Td}}, \text{Co}^{3+}_{\text{Oh}}$	$\text{Co}^{2+}_{\text{Td}}$	$\text{Co}^{3+}_{\text{Oh}}$	XANES and	
Spinel type	normal	normal	normal	EXAFS	

Catalyst	Co ₃ O ₄	CoAl ₂ O ₄	ZnCo ₂ O ₄	XRD, XANES and EXAFS	Angew. Chem. Int. Ed. 2019, 131, 6103-6108
Geometric site	Co ²⁺ _{Td} , Co ³⁺ _{Oh}	Co ²⁺ _{Td}	Co ³⁺ _{Oh}		
Spinel type	normal	normal	normal		
Catalyst	Co ₃ O ₄	CoAl ₂ O ₄	ZnCo ₂ O ₄	XRD, XPS, XANES and EXAFS	J. Am. Chem. Soc. 2022, 144, 19163- 19172
Geometric site	Co ²⁺ _{Td} , Co ³⁺ _{Oh}	Co ²⁺ _{Td}	Co ³⁺ _{Oh}		
Spinel type	normal	normal	normal		
Catalyst	Co ₃ O ₄	CoAl ₂ O ₄	ZnCo ₂ O ₄	XRD, Raman, XPS, XANES and EXAFS	ACS Catal. 2017, 7, 1626-1636
Geometric site	Co ²⁺ _{Td} , Co ³⁺ _{Oh}	Co ²⁺ _{Td}	Co ³⁺ _{Oh}		
Spinel type	normal	normal	normal		
Catalyst	Co ₃ O ₄	CoAl ₂ O ₄	ZnCo ₂ O ₄	XRD, XPS, XANES and EXAFS	ACS Appl. Mater. Interfaces 2019, 11, 12525-12534
Geometric site	Co ²⁺ _{Td} , Co ³⁺ _{Oh}	Co ²⁺ _{Td}	Co ³⁺ _{Oh}		
Spinel type	normal	normal	normal		
Catalyst	Co ₃ O ₄	CoAl ₂ O ₄	MgCo ₂ O ₄	XRD, Raman, XPS, XANES and EXAFS	J. Am. Chem. Soc. 2024, 146, 2967- 2976
Geometric site	Co ²⁺ _{Td} , Co ³⁺ _{Oh}	Co ²⁺ _{Td}	Co ³⁺ _{Oh}		
Spinel type	normal	normal	normal		
Catalyst	Co ₃ O ₄	CoAl ₂ O ₄	ZnCo ₂ O ₄	XRD, XPS, XANES and EXAFS	Angew. Chem. Int. Ed. 2020, 132, 4766-4772
Geometric site	Co ²⁺ _{Td} , Co ³⁺ _{Oh}	Co ²⁺ _{Td}	Co ³⁺ _{Oh}		
Spinel type	normal	normal	normal		
Catalyst	Co ₃ O ₄	CoCr ₂ O ₄	MgCo ₂ O ₄	XRD, Raman, XPS, XANES and EXAFS	Angew. Chem. Int. Ed. 2020, 132, 4766-4772
Geometric site	Co ²⁺ _{Td} , Co ³⁺ _{Oh}	Co ²⁺ _{Td}	Co ³⁺ _{Oh}		
Spinel type	normal	normal	normal		
Catalyst	NiAl ₂ O ₄	CuAl ₂ O ₄	CoAl ₂ O ₄	XRD, Raman, XPS, XANES and EXAFS	J. Am. Chem. Soc. 2025, 147, 988- 997
Geometric site	Al ³⁺ _{Oh} , Ni ²⁺ _{Td}	Cu ²⁺ _{Td}	Co ²⁺ _{Td}		
Spinel type	normal	normal	normal		
Catalyst	Co ₃ O ₄	CoFe ₂ O ₄	FeCo ₂ O ₄	XRD, Raman, XPS, VSM, XANES and EXAFS	J. Am. Chem. Soc. 2023, 145, 18992- 19004
Geometric site	Co ²⁺ _{Td} , Co ³⁺ _{Oh}	Co ²⁺ _{Td}	Co ³⁺ _{Oh}		
Spinel type	normal	normal	normal		
Catalyst	CoFe ₂ O ₄		Co ₂ FeO ₄	XRD, Raman, XPS, XANES and EXAFS	ACS Catal. 2018, 8, 5, 4082–4090
Geometric site	Co ²⁺ _{Td} , Co ²⁺ _{Oh}		Co ²⁺ _{Td} , Co ³⁺ _{Oh}		
Spinel type	partial		partial		
Catalyst	ZnCo ₂ O ₄		LiCoVO ₄	XRD, VSM, XPS, XANES and EXAFS	Adv. Mater. 2020, 32, 1907976.
Geometric site	Co ³⁺ _{Oh}		Co ²⁺ _{Oh}		
Spinel type	normal		inverse		
Catalyst	Co ₃ O ₄		Co ₂ VO ₄	XANES, XRD	Adv.

Geometric site	$\text{Co}^{2+}_{\text{Td}}, \text{Co}^{3+}_{\text{Oh}}$	$\text{Co}^{2+}_{\text{Oh}}$		Mater. 2020, 32, 1907168.
Spinel type	normal	inverse		
Catalyst	Co_3O_4	Co_3O_4		J. Condens. Matter Phys. 2012, 24, 435401
Geometric site	$\text{Co}^{2+}_{\text{Td}}, \text{Co}^{3+}_{\text{Oh}} (<17.7$ GPa)	$\text{Co}^{2+}_{\text{Td}}, \text{Co}^{2+}_{\text{Oh}}, \text{Co}^{3+}_{\text{Oh}}$ (>17.7 GPa)	XRD, Raman	
Spinel type	normal	partial		
Catalyst	Co_3O_4	D- Co_3O_4		J. Phys. Chem. Lett. 2022, 13, 35, 8386–8396
Geometric site	$\text{Co}^{2+}_{\text{Td}}, \text{Co}^{3+}_{\text{Oh}}$	$\text{Co}^{2+}_{\text{Td}}, \text{Co}^{2+}_{\text{Oh}}, \text{Co}^{3+}_{\text{Oh}}$	XRD, XANES and EXAFS	
Spinel type	normal	partial		

In response to the third concern regarding the influence of the exposed crystal facet on catalytic activity, we thank the Reviewer for the insightful comments. And we agree with the Reviewer that different exposed crystal facets on the spinels may substantially affect both reaction kinetics and thermodynamics (*Angew. Chem. Int. Ed.* **2011**, 50, 12294-12298, *ACS Catal.* **2017**, 7, 1626-1636, *Nat. Nanotechnol.* **2020**, 15, 848-853, *Nat. Commun.* **2024**, 15, 3923). Therefore, in this study, we regulated the exposed crystal facets of the as-synthesized Co-based spinel oxides, i.e., Co_3O_4 , CoGa_2O_4 , and ZnCo_2O_4 by making them share the same [111] facet via a carefully controlled synthesis protocol. Only in this way, should the coordination geometry effects on catalytic activity be rigorously investigated. Additionally, optimizing the catalytic activity by tuning the exposed crystal facet is not the primary focus of this study. We focused on elucidating the geometrical-dependent catalytic behaviors in cobalt-based spinel oxides and addressing the spin-polarization effects from geometrical site competition on reaction thermodynamics by ruling out the influence of crystal facets.

Figure R7. (a) Rietveld-refined XRD patterns of cubic-shaped Co_3O_4 . (b) EDX mapping images of cubic-shaped- Co_3O_4 . Atomic-resolved high-resolution TEM (HRTEM) image (c) and the corresponding SAED image (d) of cubic-shaped- Co_3O_4 . (e) N_2 sorption isotherms of synthesized cubic-shaped- Co_3O_4 . (f) ECSA measurements of cubic-shaped- Co_3O_4 . (g) O_2 -TPD profiles of cubic-shaped- Co_3O_4 and octahedral-shaped Co_3O_4 . (h) Catalytic ozonation activities of cubic-shaped- Co_3O_4 and octahedral-shaped Co_3O_4 . (i) Comparison of the k_{norms} and k_{ECSAS} for cubic-shaped- Co_3O_4 and octahedral-shaped Co_3O_4 .

In the revised manuscript, we addressed the significance of the exposed crystal facet on catalytic activity by synthesizing cubic-shaped Co_3O_4 with the exposed crystal facet of [100] as the comparative

material and evaluated its catalytic ozonation activity (*J. Environ. Manage.* **2020**, 267: 110582). The XRD with Rietveld refinement results (Fig. R7a) and the EDX mappings (Fig. R7b) suggest the successful synthesis of Co_3O_4 spinel oxide. We further determined the exposed crystal facet of cubic-shaped Co_3O_4 (Fig. R7c) using the same protocol as the other Co-based spinels via measuring interplanar crystal spacing and angle of the crystal planes measured in the SAED plot (Fig. R7d). The interplanar crystal spacing of 0.283 nm and a fixed interplanar angle of 90° between the (0 2 2), (2 0 2), and (0 2 2) crystal plane suggest the [100] crystal plane as the dominated exposed facet. By normalizing their reaction kinetics with BET SSAs/ECSA (Fig. R7h&i), we found that cubic-shaped Co_3O_4 exhibited a 2.0-fold lower intrinsic activity than octahedral-shaped Co_3O_4 with the exposed crystal facet of [111], revealing that the [111] facet indeed accelerated the reaction kinetics.

The underlying mechanisms by which exposed crystal facets influence catalytic activity are complex and result from a combination of spatial effects, the compositional proportions of active geometric sites, and surface defect structures. The spatial effect governs the number of exposed active atoms on the crystal facet and the reaction kinetics. In the [111] crystal facet, the angle between the resided Co atoms is 60° , which is smaller than the 90° angle of the resided Co atoms in the [100] crystal facet (Fig. R8a&b). This enables [111] facet to reside a higher number of Co atoms within the same reaction area than [100] facet.

Figure R8. Theoretical models of [111] (a) and [100] (b) crystal facets of Co_3O_4 and the spatial distribution of Co atoms on each facet. (c) Reported distribution of cobalt valence states across different crystal facets of Co_3O_4 .

We then differentiated the proportions of geometric sites on these crystal facets. By constructing theoretical models of the [111] and [100] crystal facets, we find that although $\text{Co}^{3+}_{\text{Oh}}$ and $\text{Co}^{2+}_{\text{Td}}$ are the exposed geometric sites on [100] and [111] facets, respectively, the interlayer spacing of $\text{Co}^{2+}_{\text{Td}}$ and $\text{Co}^{3+}_{\text{Oh}}$ sites in both facets is only $\sim 0.1 \text{ \AA}$, which is significantly lower than the inter-atom spacing of the Co sites ($> 2 \text{ \AA}$). This tiny interlayer spacing suggests that the dependence of exposed geometric sites on crystal facet is only theoretically applicable. In actual characterizations and reactions, both $\text{Co}^{2+}_{\text{Td}}$ and $\text{Co}^{3+}_{\text{Oh}}$ sites are present on the different crystal facets and the compositional proportions of $\text{Co}^{2+}_{\text{Td}}$ and $\text{Co}^{3+}_{\text{Oh}}$ sites among the exposed atoms on [100] and [111] facets are similar (Figure R8c and Table R2). This study follows this trend as well. Given the similar proportion of $\text{Co}^{2+}_{\text{Td}}$ and $\text{Co}^{3+}_{\text{Oh}}$ sites among the exposed atoms on [100] and [111] facets, the [111] facet with a higher atomic density

obtains a greater number of the active $\text{Co}^{2+}_{\text{Td}}$ sites, which accounted for its high activity in O_3 activation. It is noteworthy that the sparse distribution of Co atoms in [100] facet can promote local structural reconfiguration and facilitate the formation of surface structural defects such as oxygen vacancies and strain defects (*Angew. Chem.* **2024**, 136, e202404834). As revealed in O_2 -TPD profiles in Fig. R7g, a greater amount surface oxygen vacancy is observed on cubic-shaped Co_3O_4 with the [100] facet than that of octahedral-shaped Co_3O_4 with the [111] facet. These structural defects can also act as the active sites for O_3 activation (*J. Hazard. Mater.* **2022**, 437, 129235). In our study, the higher activity of the [111] facet than the [100] facet suggests that the contribution of exposed number of active sites to activity outperforms that of surface defective structures. This further highlights the governing role of geometric coordination sites in catalytic activity.

Based on these insights, we delicately controlled the synthesis protocol and regulated the exposed crystal facets of the as-synthesized Co-based spinel oxides by making them share the same [111] facet, which obtains a high atom density, high stability, and low structural defects, to better investigate the coordination geometry-dependent activity in this study. The elucidation of the crystal geometry coordination on catalytic activity helps us investigate the influence of different exposed crystal facets on activity, which is a promising future research opportunity to further enhance the both the catalytic activity and the thermodynamic selectivity of the reaction. We would like to thank the Reviewer again for guiding the rational design of our future work.

Table R2. Proportion of Co^{3+} and Co^{2+} on different exposed facets of Co_3O_4 in previous reports

No.	Catalysts	Co^{2+} (%)	Co^{3+} (%)	Ref.
1	[100]- Co_3O_4 -1	44.99	55.01	ACS Nanosci. Au 2024, 4, 409-415
	[111]- Co_3O_4 -1	57.05	42.95	
2	[100]- Co_3O_4 -2	33.44	66.56	J. Hazard. Mater. 2020, 392, 122358
	[111]- Co_3O_4 -2	33.47	66.53	
3	[100]- Co_3O_4 -3	33.56	66.44	Appl. Catal. B Environ. 2025, 364, 124854
	[111]- Co_3O_4 -3	36.50	63.50	
4	[100]- Co_3O_4 -4	51.22	48.78	Electrochim. Acta 2025, 513, 145588
	[111]- Co_3O_4 -4	52.15	47.85	
5	[100]- Co_3O_4 -5	66.23	33.77	Appl. Catal. B Environ. 2021, 283, 119657
	[111]- Co_3O_4 -5	67.11	32.89	
6	[100]- Co_3O_4 -6	35.6	64.4	Chem. Eng. J. 2018, 337, 488-498
	[111]- Co_3O_4 -6	42.8	57.2	
7	[100]- Co_3O_4 -7	36.22	63.67	Appl. Catal. B Environ. 2020, 270, 118819
	[111]- Co_3O_4 -7	42.19	57.81	
8	[100]- Co_3O_4 -9	31.7	68.3	This work
	[111]- Co_3O_4 -9	33.1	66.9	

Changes made in this Revision

Line 62, Page 5, Manuscript

The significantly larger octahedral site preference energy (OSPE) of Co^{3+} compared to those of Co^{2+} and Zn^{2+} enables Co^{3+} to preferentially occupy the octahedral sites while resides the Co^{2+} and Zn^{2+} in tetrahedral sites²³. The fully occupied 3d orbitals in Ga^{3+} exclude the coordination geometry competition in CoGa_2O_4 , making $\text{Co}^{2+}_{\text{Td}}$ as the dominant active sites. Additionally, the A_{1g} symmetry and F_{2g}^1 symmetry in CoGa_2O_4 and ZnCo_2O_4 shifted slightly compared to the Co_3O_4 , respectively. This observation reinforces the normal spinel structures of the as-synthesized Co-based spinels and the minor presence of inversed $\text{Co}^{2+}_{\text{Oh}}/\text{Co}^{3+}_{\text{Td}}$ sites²⁴ (Supplementary Fig. 3).

Line 84, Page 7, Manuscript

$\text{Zn}^{2+}/\text{Ga}^{3+}$ with larger ionic radii than $\text{Co}^{2+}/\text{Co}^{3+}$ (0.58 vs. 0.60 Å, 0.61 vs. 0.62 Å) slightly altered the interplanar crystal spacing within the spinel structure (Supplementary Fig. 2; Supplementary Table 1). This variation in interplanar crystal spacing induces covalency competition between $M_{\text{Td}}\text{-O}$ (metal-oxygen bond in tetrahedral unit) and $M_{\text{Oh}}\text{-O}$ (metal-oxygen bond in octahedral unit) in the $M_{\text{Td}}\text{-O-M}_{\text{Oh}}$ backbone of the spinel crystal structure, affecting the exposed active sites and the electronic properties that govern the catalytic activity²⁵.

Line 145, Page 10, Manuscript

In addition, the significant variations in peak intensity ratios of $\text{Co}_{\text{Oh}}\text{-Co}_{\text{Oh}}$ path to $\text{Co}_{\text{Oh}}\text{-Co}_{\text{Td}}$ path for CoGa_2O_4 and ZnCo_2O_4 compared to that of Co_3O_4 suggested that the inversed $\text{Co}^{2+}_{\text{Oh}}/\text{Co}^{3+}_{\text{Td}}$ sites act as the minor coordination-geometry-competitors against the dominant $\text{Co}^{2+}_{\text{Td}}$ and $\text{Co}^{3+}_{\text{Oh}}$ sites (Supplementary Fig. 12).

Line 208, Page 13, Manuscript

3DOM-Co₃O₄ obtains a similar activity to CoGa₂O₄, yet simply increasing exposure amount of the active sites did not enhance their intrinsic activity. Both the k_{norm} and the electrochemical surface area (ECSA) normalized reaction kinetic (k_{ECSA}) of 3DOM-Co₃O₄ are significantly lower than those of CoGa₂O₄ (59-fold and 8.7-fold, respectively) (Supplementary Fig. 25). Similarly, the intrinsic activity (k_{ECSAS}) of Co sites in plate Co₃O₄ and spherical Co₃O₄ are 13 and 39-fold less than that of CoGa₂O₄. These suggest the great significance of regulating coordination geometry in Co-based spinel outperforms SSA engineering in determining the intrinsic activity of the active sites.

Line 225, Page 13, Manuscript

This high intrinsic activity of CoGa₂O₄ can be attributed to the selectively exposure of the active Co²⁺_{Td} sites via coordinating environment regulation. Co²⁺_{Td} sites with a greater tendency to donate the frontier electrons facilitate the electrophilic activation of O₃ molecules than Co³⁺_{Oh} sites^{16, 21}.

Line 229, Page 14, Manuscript

Influence of the exposed crystal facet on catalytic activity was also investigated by synthesizing cubic-shaped Co₃O₄ exposing the [100] crystal facet as the reference catalyst (Supplementary Fig. 27). Results suggest that octahedral-shaped Co₃O₄ with the [111] crystal facet exhibits a higher intrinsic activity than the cubic-shaped Co₃O₄ with the [100] crystal facet. Although the sparse distribution of Co atoms in the [100] facet results in a greater surface defective structure than the [111] facet, the larger number of the active Co²⁺_{Td} sites on the [111] facet governs the catalytic activity (Supplementary Fig. 28)^{35, 36}. This further discloses the decisive role of geometric coordination sites in catalytic activity. Furthermore, the inversed Co²⁺_{Oh} sites as the minor coordination-geometry-competitors against the

dominant $\text{Co}^{2+}_{\text{Td}}$ sites demonstrate a markedly lower intrinsic catalytic activity than the $\text{Co}^{2+}_{\text{Td}}$ sites, which can be ascribed to the inefficient electron transfer ability of octahedrally coordinated O atoms $\text{Co}^{2+}_{\text{Oh}}$ sites compared to tetrahedrally coordinated O atoms in $\text{Co}^{2+}_{\text{Td}}$ sites (Supplementary Fig. 29)²¹.

Page 26, Supplementary Information

Supplementary Fig. 3. Raman spectra of spinel oxides.

Raman spectra indicate that Co_3O_4 displays five Raman peaks: F_{2g}^1 (194 cm^{-1}), E_g (480 cm^{-1}), F_{2g}^2 (521 cm^{-1}), F_{2g}^3 (615 cm^{-1}), and A_{1g} (688 cm^{-1}). Specifically, the A_{1g} and F_{2g}^1 peaks correspond to signals from octahedral and tetrahedra sites, respectively, while E_g , F_{2g}^2 , and F_{2g}^3 are signals of octahedral and/or tetrahedra sites without clear identification. Substituting Co^{3+} by Ga^{3+} in CoGa_2O_4 blue shifted the A_{1g} symmetry, while the peak position of F_{2g}^1 symmetry remained unchanged. The variation of the cation–anion bond length at the octahedral sites and polyhedral distortion occurring in the spinel lattice after Ga^{3+} replacement accounts for the change of A_{1g} symmetry in CoGa_2O_4 . The unchanged coordinative environment of $\text{Co}^{2+}_{\text{Td}}$ sites maintained the F_{2g}^1 symmetry. Similarly, the shift of F_{2g}^1 symmetry peak and the maintaining of A_{1g} symmetry in ZnCo_2O_4 suggest that Zn^{2+} occupied the tetrahedral sites while the $\text{Co}^{3+}_{\text{Oh}}$ sites remained unchanged. The above Raman results further solidify the normal spinel structures of the as-synthesized Co-based spinels and the minor presence of

inversed $\text{Co}^{2+}_{\text{Oh}}/\text{Co}^{3+}_{\text{Td}}$ sites.

Page 36, Supplementary Information

Supplementary Fig. 13. The corresponding Fourier transformed EXAFS (FT-EXAFS) spectra at R space.

FT-EXAFS spectra indicated that the peak intensity ratio of $\text{Co}_{\text{Oh}}-\text{Co}_{\text{Oh}}$ path (at 2.5 Å) to $\text{Co}_{\text{Td}}-\text{Co}_{\text{Oh}}$ path (at 3.0 Å) for Co_3O_4 and CoGa_2O_4 were 1.39 and 0.79, respectively. The significant decrease of the peak intensity ratio suggests that most of the Co octahedral sites have been replaced by Ga^{3+} cations, while the considerably decreased $\text{Co}_{\text{Oh}}-\text{Co}_{\text{Oh}}$ path for CoGa_2O_4 might originate from minor presence of $\text{Co}^{2+}_{\text{Oh}}$ sites and the interference signal of Ga d orbital electrons. To investigate the influence of this interference signal from Ga, we recorded the FT-EXAFS spectrum of CoAl_2O_4 , which was synthesized using the same protocol as these Co-based spinels, and compared it to that of CoGa_2O_4 . Evidently, the absence of d orbital in Al^{3+} diminished the scattering path at ~2.5 Å. This suggests $\text{Co}^{2+}_{\text{Oh}}$ sites as the minor coordination-geometry-competitor against the dominant $\text{Co}^{2+}_{\text{Td}}$ sites.

Page 49, Supplementary Information

Supplementary Fig. 25. Cyclic Voltammetry (CV) curves of Co_3O_4 (a), ZnCo_2O_4 (b), CoGa_2O_4 (c), spherical Co_3O_4 (d), 3DOM Co_3O_4 (e), and plate-like Co_3O_4 (f) at different scan rates, and ECSA measurements of samples (g) and (h). (i) Comparison of the BET surface area normalized reaction rates (k_{norm}) and ECSA normalized reaction rates (k_{ECSA}) of different spinels/ O_3 systems. The electrolyte contained $100 \text{ mg L}^{-1} \text{ Na}_2\text{SO}_4$ (pH=3) and the 0.1 mg cm^{-2} catalyst was loaded on glassy carbon-rotating disk electrode (GC-RDE).

Considering the electron transfer nature of the heterogeneous O_3 activation process, we further measured the electrochemical active surface areas (ECSAs) of the as-synthesized catalysts to examine the number of the exposed active sites during the reaction (Supplementary Figs. 23a-f). The intrinsic activity of the active sites with different coordination geometries were further compared by ECSAs

normalized reaction kinetics (k_{ECSAS}), which served as a strong complementary of SSAs normalized reaction kinetics (k_{normS}) to exclude the influence of exposed active sites⁸. By fitting the non-Faradaic currents from cyclic voltammetry measurements at different scan rates, the ECSAs of Co_3O_4 , CoGa_2O_4 , ZnCo_2O_4 , and 3DOM Co_3O_4 are derived as 0.50, 0.30, 0.44, and 3.3 mF cm^{-2} , respectively (Supplementary Figs. 23e and g). This suggests that 3DOM Co_3O_4 exposes the highest number of active sites during the heterogeneous reactions owing to its periodic structure, which agrees with its highest BET SSA. The k_{ECSAS} can provide a more accurate representation of the intrinsic activity of the exposed active sites than reaction kinetics normalized by BET SSAs (k_{normS}). It is found that the exposed Co sites in CoGa_2O_4 obtained an 8.7-fold higher intrinsic activity than those in 3DOM Co_3O_4 (Supplementary Fig. 23i). Additionally, the intrinsic activity of the Co sites in CoGa_2O_4 , Co_3O_4 , and ZnCo_2O_4 align well with the trends observed in the k_{normS} comparison. These results collectively reinforce the notion that regulating coordination geometry in Co-based spinel outperforms SSA engineering in determining the intrinsic activity of the active sites.

Supplementary Fig. 27 (a) Rietveld-refined XRD patterns of cubic-shaped Co_3O_4 . (b) EDX mapping images of cubic-shaped- Co_3O_4 . Atomic-resolved high-resolution TEM (HRTEM) image (c) and the corresponding SAED image (d) of cubic-shaped- Co_3O_4 . (e) N_2 sorption isotherms of synthesized cubic-shaped- Co_3O_4 . (f) ECSA measurements of cubic-shaped- Co_3O_4 . (g) O_2 -TPD profiles of cubic-shaped- Co_3O_4 and octahedral-shaped Co_3O_4 . (h) Catalytic ozonation activity of cubic-shaped- Co_3O_4 and octahedral-shaped Co_3O_4 . (i) Comparison of the k_{norms} and k_{ECSAS} for cubic-shaped- Co_3O_4 and octahedral-shaped Co_3O_4 .

We further synthesized cubic-shaped Co_3O_4 with the exposed crystal facet of [100] as the comparative material and evaluated its catalytic ozonation activity⁹. The synthesis method is described as below.

Synthesis of cubic-shaped Co_3O_4

In a typical synthesis, 0.04 mol of cobalt nitrate hexahydrate and 0.01 mol of sodium hydroxide were dissolved in 40 mL of deionized water under vigorous stirring. Once the mixture was thoroughly dissolved, it was transferred to a Teflon-lined autoclave and subjected to hydrothermal treatment at 180 °C for 5 h. After the heating process, the solid product was collected by filtering and washing with deionized water and ethyl alcohol. The obtained product was then dried at 80 °C overnight.

The XRD with Rietveld refinement results (Supplementary Fig. 27a) and the EDX mappings (Supplementary Fig. 27b) suggest the successful synthesis of Co_3O_4 spinel oxide. We further determined the exposed crystal facet of cubic-shaped Co_3O_4 (Supplementary Fig. 29c) using the same protocol as the other Co-based spinels via measuring interplanar crystal spacing and angle of the crystal planes measured in the SAED plot⁹ (Supplementary Fig. 27d). The interplanar crystal spacing of 0.283 nm and a fixed interplanar angle of 90 ° between the (0 2 2), (2 0 2), and (0 2 2) crystal plane suggest the [100] crystal plane as the dominated exposed facet. By normalizing their reaction kinetics with BET SSAs/ECSA (Supplementary Figs. 27h&i), we found that cubic-shaped Co_3O_4 exhibited a 2.0-fold lower intrinsic activity than octahedral-shaped Co_3O_4 with the exposed crystal facet of [111], revealing that the [111] facet indeed accelerated the reaction kinetics.

Supplementary Fig. 28 Theoretical models of [111] (a) and [100] (b) crystal facets of Co₃O₄ and the spatial distribution of Co atoms on each facet. (c) Reported distribution of cobalt valence states across different crystal facets of Co₃O₄.

In the [111] crystal facet, the angle between the resided Co atoms is 60 °, which is smaller than the 90 ° angle of the resided Co atoms in the [100] crystal facet. This enables [111] facet to reside a higher number of Co atoms within the same reaction area than [100] facet (Supplementary Figs. 30a and b).

We then differentiated the proportions of geometric sites on these crystal facets. By constructing theoretical models of the [111] and [100] crystal facets, we find that although Co³⁺_{Oh} and Co²⁺_{Td} are the exposed geometric sites on [100] and [111] facets, respectively, the interlayer spacing of Co²⁺_{Td} and Co³⁺_{Oh} sites in both facets is only ~0.1 Å, which is significantly lower than the inter-atom spacing of the Co sites (>2 Å). This tiny interlayer spacing suggests that the dependence of exposed geometric sites on crystal facet is only theoretically applicable. In actual characterizations and reactions, both

$\text{Co}^{2+}_{\text{Td}}$ and $\text{Co}^{3+}_{\text{Oh}}$ sites are present on the different crystal facets and the compositional proportions of $\text{Co}^{2+}_{\text{Td}}$ and $\text{Co}^{3+}_{\text{Oh}}$ sites among the exposed atoms on [100] and [111] facets are similar (Supplementary Fig. 30c and Supplementary Table 11). This study follows this trend as well. Given the similar proportion of $\text{Co}^{2+}_{\text{Td}}$ and $\text{Co}^{3+}_{\text{Oh}}$ sites among the exposed atoms on [100] and [111] facets, the [111] facet with a higher atomic density obtains a greater number of the active $\text{Co}^{2+}_{\text{Td}}$ sites, which accounted for its high activity in O_3 activation.

It is noteworthy that the sparse distribution of Co atoms in [100] facet can promote local structural reconfiguration and facilitate the formation of surface structural defects such as oxygen vacancies and strain defects. As revealed in O_2 -TPD profiles in Supplementary Fig. 29g, a greater amount of surface oxygen vacancies are observed on cubic-shaped Co_3O_4 with the [100] facet than that of octahedral-shaped Co_3O_4 with the [111] facet. These structural defects can also act as the active sites for O_3 activation. In our study, the higher activity of the [111] facet than the [100] facet suggests that the contribution of exposed number of active sites to activity outperforms that of surface defective structures. This further highlights the governing role of geometric coordination sites in catalytic activity.

Supplementary Fig. 29. (a) High resolution XPS surveys on Co 2p for CoO and Co₃O₄. (b) Rietveld-refined XRD patterns of CoO. (c) N₂ sorption isotherms of synthesized CoO. (d) ECSA measurements of CoO with inset of the corresponding CV curves with different scanning rates. (e) Catalytic ozonation activity of CoO and spinel oxides. (f) Comparison of k_{ECAS} and k_{norms} of CoO and spinel oxides.

To differentiate the intrinsic catalytic activity between Co²⁺_{Td} sites with Co²⁺_{Oh} sites, we synthesized CoO with Co²⁺_{Oh} as the predominated Co sites according to the previous reported method by annealing of Co₃O₄ in Ar atmosphere at 900 °C for 4 h¹⁰. XPS survey confirms the absence of Co³⁺ (Supplementary Fig. 29a), while XRD Rietveld refinement result suggests the successful preparation of the CoO with Co²⁺_{Oh} as the single crystal phase (Supplementary Fig. 29b). We evaluated the catalytic ozonation activity of the as-synthesized CoO and normalized it by its BET SSA (Supplementary Fig. 29c) and ECSA (Supplementary Fig. 29d) to obtain the intrinsic activity. It is found that the intrinsic activity (k_{ECAS}) of Co²⁺_{Oh} sites is 1.5-fold higher than that of Co³⁺_{Oh}, yet 10.1-fold lower than Co²⁺_{Td}

(Supplementary Figs. 29e and f). Similar trend is also observed in comparison of k_{norm} s. This can be ascribed to the inefficient electron transfer ability of octahedrally coordinated O atoms compared to tetrahedrally coordinated O atoms in Co^{2+}_{Td} sites.

Supplementary Table 11. Proportion of Co^{3+} and Co^{2+} on different exposed facets of Co_3O_4 in previous reports

No.	Catalyst	Co^{2+} (%)	Co^{3+} (%)	Ref.
1	[100]- Co_3O_4 -1	44.99	55.01	11
	[111]- Co_3O_4 -1	57.05	42.95	
2	[100]- Co_3O_4 -2	33.44	66.56	12
	[111]- Co_3O_4 -2	33.47	66.53	
3	[100]- Co_3O_4 -3	33.56	66.44	13
	[111]- Co_3O_4 -3	36.50	63.50	
4	[100]- Co_3O_4 -4	51.22	48.78	14
	[111]- Co_3O_4 -4	52.15	47.85	
5	[100]- Co_3O_4 -5	66.23	33.77	15
	[111]- Co_3O_4 -5	67.11	32.89	
6	[100]- Co_3O_4 -6	35.6	64.4	16
	[111]- Co_3O_4 -6	42.8	57.2	
7	[100]- Co_3O_4 -7	36.22	63.67	17
	[111]- Co_3O_4 -7	42.19	57.81	
	[100]- Co_3O_4 -9	31.7	68.3	This work
	[111]- Co_3O_4 -9	33.1	66.9	

Reviewer #5

Comment: I co-reviewed this manuscript with one of the reviewers who provided the listed reports.

This is part of the Nature Communications initiative to facilitate training in peer review and to provide appropriate recognition for Early Career Researchers who co-review manuscripts.

Response: Thanks very much for kindly providing training and mentoring opportunities to ECR(s) for involving peer review process for top journals.

Responses to Editorial Requests and Reviewers' Comments

Reviewer #4

Comments: The authors have provided sufficient new evidences and literature survey to support the conclusions. My concerns have been addressed and I think now it can be considered for publication.

Response: We sincerely appreciate your assistance in enhancing the quality of our manuscript and thank you for your support in its publication.

Reviewer #5

Comments: I co-reviewed this manuscript with one of the reviewers who provided the listed reports. This is part of the Nature Communications initiative to facilitate training in peer review and to provide appropriate recognition for Early Career Researchers who co-review manuscripts.

Response: Thanks very much for kindly providing training and mentoring opportunities to ECR(s) for involving peer review process for top journals.